# ADEPT: Continual Pretraining via Adaptive Expansion and Dynamic Decoupled Tuning

**Jinyang Zhang**[1,2,*], **Yue Fang**[1,2,*], **Hongxin Ding**[1,2,*], **Weibin Liao**[1,2,*], **Muyang Ye**[3],
**Junfeng Zhao**[†,1,2,4], **Yasha Wang**[†,2,5,6], **Xu Chu**[†,1,2,7]

[1]School of Computer Science, Peking University
[2]Key Laboratory of High Confidence Software Technologies, Ministry of Education
[3]Zhejiang University, Hangzhou, China
[4]Big Data Technology Research Center, Nanhu Laboratory
[5]National Engineering Research Center For Software Engineering, Peking University
[6]Peking University Information Technology Institute (Tianjin Binhai)
[7]Center on Frontiers of Computing Studies, Peking University

✉ jinyangzhang25@stu.pku.edu.cn
{chuxu, zhaojf, wangyasha}@pku.edu.cn

https://github.com/PuppyKnightUniversity/ADEPT.git

## ABSTRACT

Conventional continual pretraining (CPT) for large language model (LLM) domain adaptation often suffers from catastrophic forgetting and limited domain capacity. Existing strategies adopt layer expansion, introducing additional trainable parameters to accommodate new knowledge. However, the uniform expansion and updates still entangle general and domain learning, undermining its effectiveness. Our pilot studies reveal that LLMs exhibit functional specialization, where layers and units differentially encode general-critical capabilities, suggesting that parameter expansion and optimization should be function-aware. We then propose ADEPT, **A**daptive Expansion and **D**ynamic D**e**coupled Tuning for continual **p**re**t**raining, a two-stage framework for domain-adaptive CPT. ADEPT first performs *General-Competence Guided Selective Layer Expansion*, duplicating layers least critical for the general domain to increase representational capacity while minimizing interference with general knowledge. It then applies *Adaptive Unit-Wise Decoupled Tuning*, disentangling parameter units within expanded layers according to their general-domain importance and assigning asymmetric learning rates to balance knowledge injection and retention. Experiments on mathematical and medical domains show that ADEPT outperforms full-parameter CPT by up to 5.76% on the general benchmarks and 5.58% on the target domain benchmarks with only 15% of parameters tuned and less than 50% training time. Ablation studies, theoretical analysis, and extended investigations further demonstrate the necessity of targeted expansion and decoupled optimization, providing new principles for efficient and robust domain-adaptive CPT.

## 1 INTRODUCTION

Large language models (LLMs) have demonstrated remarkable performance across a wide range of general-domain tasks (OpenAI, 2023; Dubey et al., 2024c). However, their deployment in specialized domains, such as mathematics or healthcare, requires targeted adaptation (Ding et al., 2024; Chen et al., 2024; Ahn et al., 2024). Continual pretraining (CPT), which conducts post-pretraining on domain-specific corpora, has emerged as a crucial paradigm for injecting domain knowledge and capabilities into pretrained LLMs (Wu et al., 2024a; Ibrahim et al., 2024; Yıldız et al., 2024).

Despite its promise, CPT faces a persistent challenge: catastrophic forgetting. After pretraining, LLMs already encode substantial general knowledge, leaving limited parameter capacity for integrating new domain-specific information. While domain signals can be forcefully fitted through

---

[*]Equal contribution.
[†]Corresponding author.

gradient-based optimization, the aggressive updates on the existing parameters come at the cost of overfitting to the target corpora, which in turn disrupts general abilities and triggers catastrophic forgetting (Liu et al., 2024a; Luo et al., 2025). This tension between new knowledge injection and previous knowledge retention poses a central obstacle to reliable and stable domain adaptation.

To address catastrophic forgetting, some approaches attempt through data-centric strategies, such as data replay or rehearsal (Huang et al., 2024; Zhang et al., 2025). While replay partially preserves prior knowledge, it fails to expand model capacity, leaving the conflict between knowledge injection and retention unresolved. Others focus on increasing capacity via transformer-layer extension (Wu et al., 2024b), yet typically insert new layers uniformly and update all parameters indiscriminately. This expansion strategy neglects the functional specialization within LLMs, where different layers and neurons serve distinct functional roles. Our pilot studies reveal that general-critical layers in LLMs are mainly located in early depths, and functional units within layers contribute unequally to general-domain performance, highlighting functional specialization similar to that found in the human brain (Xu et al., 2025; Zheng et al., 2024; Dai et al., 2022c). Consequently, indiscriminate expansion and optimization may overwrite general-critical regions with new knowledge, compromising general competency preservation and leaving forgetting unresolved.

Inspired by the functional specialization perspective, we propose our core insight: **effective CPT should expand and update the model adaptively, preserving the regions responsible for the general domain and targeting more adaptable parameters.** Specifically, we argue that capacity allocation must be importance-guided, and optimization must be function-decoupled to minimize interference with general competencies. As illustrated in Figure 1, domain-specific extension should be allocated to the regions less constrained by general-domain knowledge and skills, and parameters within these regions should be decoupled and tuned accordingly, preserving general-critical parameters and allowing the rest to be more adaptable to absorb new domain-specific information.

Building on this insight, we propose **A**daptive Expansion and **D**ynamic D**e**coupled Tuning for continual **p**re-**t**raining (ADEPT), a framework for domain-adaptive continual pretraining. ADEPT comprises two stages: *General-Competence Guided Selective Layer Expansion*, which identifies and duplicates layers least critical for the general domain, allocating additional capacity precisely where interference with general capabilities is minimized, thereby preventing catastrophic forgetting. *Adaptive Unit-Wise Decoupled Tuning*, which disentangles the parameters within the expanded layers based on their importance to the general domain. Asymmetric learning rates are then applied on their subsets, ensuring that general-critical parameters are preserved while more adaptable parameters can fully absorb domain-specific knowledge. Extensive experiments on mathematical and medicine domains demonstrate that ADEPT enables efficient and robust domain knowledge injection, while substantially alleviating catastrophic forgetting. Specifically, compared to full-parameter CPT, ADEPT achieves up to **5.58%** accuracy gain on target-domain benchmarks, and

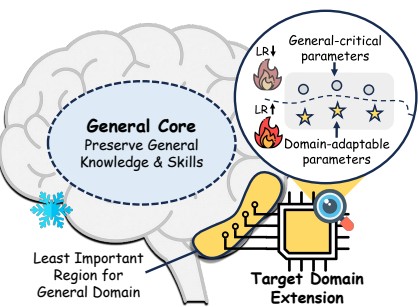

Figure 1: Illustration of the core idea of ADEPT. Target domain extension are applied on the least important region for general domain, minimizing catastrophic forgetting. Asymmetric learning rates are applied to parameter subsets for targeted knowledge injection.

up to **5.76%** gain on the general domain, confirming both effective knowledge acquisition and strong retention of general competencies. Furthermore, ADEPT attains these improvements with only 15% of parameters tuned, and reduces training time relative to other baselines greatly, highlighting its efficiency. Ablation studies and theoretical analysis further validate the designs of ADEPT.

To summarize, our contributions are threefold:

1. **Insightfully**, we highlight the importance of considering functional specialization in LLMs for continual pretraining through empirical experiments and theoretical analysis, advocating for targeted layer expansion and decoupled training as a principled solution to domain adaptation.

2. **Technically**, we propose ADEPT, a framework that consists of General-Competence Guided Selective Layer Expansion and Adaptive Unit-Wise Decoupled Tuning, enabling adaptive and effective domain knowledge integration while minimizing catastrophic forgetting.

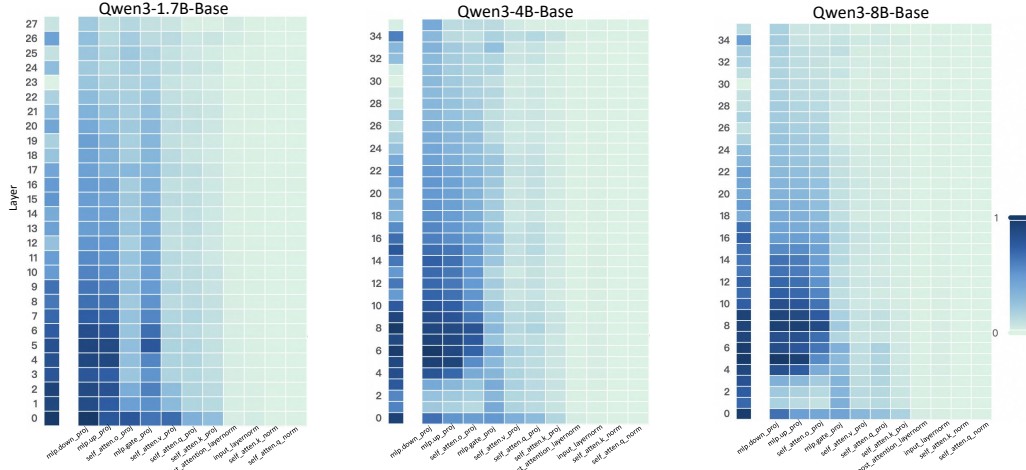

Figure 2: Layer- and unit-level importance distribution of the Qwen3 family. The vertical axis corresponds to different layers, while the horizontal axis denotes parameter units within each layer. Deeper blue indicates higher importance for preserving general-domain competencies.

3. **Empirically**, we conduct extensive experiments on both mathematical and medical domains, demonstrating that ADEPT consistently outperforms baselines in domain performance while preserving general competencies.

## 2 PILOT STUDY: PROBING PARAMETER IMPORTANCE

### 2.1 EXPERIMENTAL SETUP FOR IMPORTANCE PROBING

To investigate the functional specialization of LLMs and understand how different parameters contribute to preserving general-domain knowledge during CPT, we conduct importance probing on multiple backbone models, including *Qwen3-Base* (1.7B, 4B, 8B) (Yang et al., 2025) and *LLaMA3-8B* (Dubey et al., 2024b). Our analyses focus on probing **general-knowledge-critical parameters rather than domain-specific ones**. The rationale is that successful CPT must inject new, domain-specific knowledge without inducing catastrophic forgetting. This necessitates identifying and preserving the model's core parameters that are crucial for its general-domain competencies. By contrast, domain knowledge can then be effectively allocated to less critical parameters, without risking the erosion of pre-existing knowledge and skills. To support this analysis, we construct a *General Competence Detection Corpus* containing broad world knowledge and instruction-following tasks in both English and Chinese, which serves as the probing ground to reflect a model's general competencies. Details of its construction are provided in Appendix B.3.

### 2.2 LAYER-LEVEL IMPORTANCE PROBING

Our first research question is: **How do different layers contribute to preserving general knowledge?** To answer this, we measure the importance of each transformer layer by the model's degradation in general-domain performance when that layer is ablated. Formally, given the *General Competence Detection Corpus* $\mathcal{D}_{\text{probe}}$, we first compute the baseline next-token prediction loss of the pretrained LLM $M_0$:

$$\mathcal{L}_{\text{base}} = \frac{1}{|\mathcal{D}_{\text{probe}}|} \sum_{x \in \mathcal{D}_{\text{probe}}} \ell\big(M_0(x), x\big), \tag{1}$$

where $\ell(\cdot)$ denotes the standard next-token prediction loss in CPT. For each transformer layer $l \in \{1, \ldots, L\}$, we mask its output via a residual bypass and recompute the loss:

$$\hat{\mathcal{L}}^{(l)} = \frac{1}{|\mathcal{D}_{\text{probe}}|} \sum_{x \in \mathcal{D}_{\text{probe}}} \ell\big(M_0^{(-l)}(x), x\big), \tag{2}$$

where $M_0^{(-l)}$ denotes the model with the $l$-th layer masked. The importance of layer $l$ is defined as the loss increase relative to the baseline:

$$I_{\text{layer}}^{(l)} = \hat{\mathcal{L}}^{(l)} - \mathcal{L}_{\text{base}}. \tag{3}$$

A larger $I_{\text{layer}}^{(l)}$ indicates that layer $l$ plays a more critical role in preserving general knowledge. Figure 2 (left-hand bars) reports the layer-level importance distributions of the *Qwen3 family* (results for *LLaMA3-8B* provided in Appendix D). We find that general-knowledge-critical layers are concentrated in the early layers, with importance gradually decreasing toward later layers. This uneven distribution suggests that uniformly expanding layers across the entire depth would be suboptimal. Since some layers are tightly coupled with general knowledge while others are more flexible, uniform expansion not only risks representational interference in critical layers but also allocates parametric budget where it is too constrained to be leveraged for domain learning. In contrast, identifying more adaptable layers with minimal impact on general knowledge and allocating expansion there for knowledge injection is a superior strategy. This leads to our first key observation:

***Observation I:*** *Layers exhibit heterogeneous importance for preserving general competencies, which motivates a selective expansion strategy that targets layers less constrained by general abilities yet more adaptable for domain adaptation.*

### 2.3 UNIT-LEVEL IMPORTANCE PROBING

Building on the layer-level exploration, our next research question is: **How do parameter units within each layer contribute to preserving general knowledge?** To answer this, we partition each transformer layer into functional units (e.g., attention projections, MLP components, and normalization) and assess their relative contributions to preserving general competencies. The detailed partitioning scheme is provided in Appendix C. This granularity provides a more fine-grained perspective than layer-level probing, while avoiding the prohibitive cost of neuron-level analysis. Formally, for each parameter $\theta_j$ in a unit $U$, we estimate its importance using a first-order Taylor approximation:

$$I_j = \theta_j \cdot \nabla_{\theta_j} \mathcal{L}, \tag{4}$$

where $\mathcal{L}$ is the autoregressive training loss. The importance of unit $U$ is then defined as the average importance of its parameters:

$$I_{\text{unit}} = \frac{1}{|U|} \sum_{j \in U} I_j. \tag{5}$$

A higher $I_{\text{unit}}$ indicates that the unit plays a more critical role in preserving general competencies. Figure 2 (right-hand heatmaps) illustrates the unit-level importance distributions of the Qwen3 family (results for LLaMA3-8B provided in Appendix D). We observe that importance is unevenly distributed across modules within a layer, with some units contributing more to general competencies and others more flexible. This finding suggests that treating all parameter units equally would be suboptimal, as a single update rule cannot simultaneously protect critical units and fully train adaptable ones, risking either damaging previous knowledge or failing to sufficiently learn new knowledge. This motivates us to pursue unit-level decoupling, where training can selectively protect critical units while enabling less general-relevant units to absorb new knowledge without constraint. This leads to our second key observation:

***Observation II:*** *Parameter units within each layer exhibit heterogeneous importance, which motivates unit-level decoupling that selectively protects critical units while enabling more adaptable ones to sufficiently absorb domain knowledge.*

**Summary.** Building on the above observations, we propose ADEPT, a continual pretraining framework designed to enable effective domain knowledge injection while preserving general competencies. Inspired by the uneven importance distribution of layers (***Observation I***), ADEPT adopts a *selective expansion* strategy that targets layers less constrained by general abilities yet more receptive to domain adaptation—departing from uniform expansion approaches like LLaMA-Pro (Wu et al., 2024b). Guided by the heterogeneous importance of parameter units within layers (***Observation II***), ADEPT further introduces *unit-level decoupling* on the expanded layers, enabling critical units to be protected while allowing adaptable units to specialize in domain knowledge.

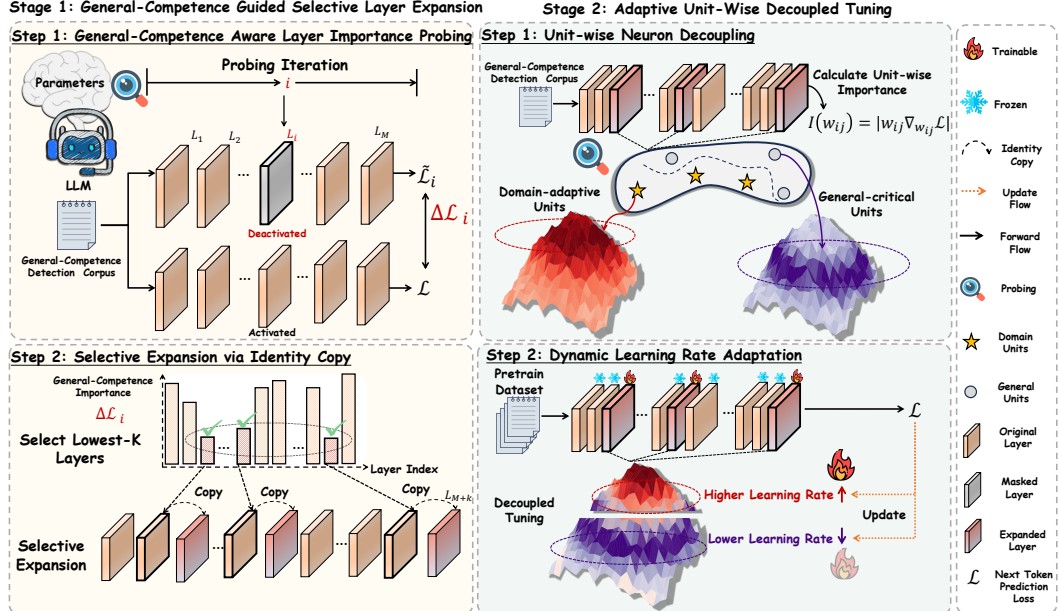

Figure 3: Illustration of ADEPT.

## 3 METHODOLOGY

As illustrated in Figure 3, ADEPT includes two stages:

- **# Stage 1: General-Competence Guided Selective Layer Expansion.** adaptively selects and duplicates layers that minimally affect general competencies while being more adaptable to domain-specific knowledge, thereby introducing fresh representational capacity for domain adaptation.
- **# Stage 2: Adaptive Unit-Wise Decoupled Tuning.** further decouples units within the expanded layers and apply learning-rate–driven adaptive tuning according to their importance to the general domain, ensuring knowledge injection while preserving general competencies.

### 3.1 GENERAL-COMPETENCE GUIDED SELECTIVE LAYER EXPANSION

This stage aims to selectively expand model parameters in a way that introduces fresh representational capacity for domain adaptation while preserving general-domain competencies. To this end, we first estimate the contribution of each transformer layer to preserving general knowledge through *General-Competence Aware Layer Importance Probing*, and then perform *Selective Parameter Expansion via Identity Copy* to duplicate layers that are least critical for general abilities yet more adaptable to domain-specific knowledge.

**General-Competence Aware Layer Importance Probing.** To guide selective expansion, we leverage the layer importance scores $I_{\text{layer}}^{(l)}$ defined as Eq.3. Intuitively, $I_{\text{layer}}^{(l)}$ quantifies how much the $l$-th layer contributes to preserving general-domain knowledge. Layers with lower scores are deemed less critical for general competencies and are thus selected for expansion, as they can accommodate domain-specific adaptation with minimal risk of catastrophic forgetting.

**Selective Parameter Expansion via Identity Copy.** Based on the importance scores $I_{\text{layer}}^{(l)}$, we sort layers by ascending importance and select the *k least-important* ones for general competence:

$$\mathcal{S}_k = \underset{\substack{\mathcal{S} \subseteq \{1,\dots,L\} \\ |\mathcal{S}| = k}}{\arg\min} \sum_{l \in \mathcal{S}} I_{\text{layer}}^{(l)}. \tag{6}$$

We denote the selected set $\mathcal{S}_k$ as the *Domain-Adaptable Layers*. For each selected layer $l \in \mathcal{S}_k$, we create a parallel copy by directly duplicating its parameters without re-initialization ($\tilde{\Theta}^{(l)} = \Theta^{(l)}$).

To preserve stability, we follow the *Function Preserving Initialization* (FPI) principle (Chen et al., 2015), ensuring that the expanded model $M_1$ produces identical outputs as the original model $M_0$ at initialization. Concretely, in the duplicated branch, we set the output projections of both attention and feed-forward sublayers to zero ($W_{\text{MHSA}}^{\text{out}} = 0$, $W_{\text{FFN}}^{\text{out}} = 0$), so the forward computation remains unchanged ($M_1(x) = M_0(x)$, $\forall x$). The duplicated layers thus provide *fresh representational capacity* that can specialize for domain signals with minimal risk of eroding general-knowledge-critical parameters in the original pathway. As formally established in Appendix F.1, expanding the layers with the lowest general-competence importance provably minimizes the risk of forgetting. Intuitively, this strategy ensures that new capacity is added where interference with general abilities is weakest, yielding the most favorable trade-off between domain adaptation and knowledge retention.

## 3.2 ADAPTIVE UNIT-WISE DECOUPLED TUNING

This stage aims to further reduce catastrophic forgetting and enable fine-grained control over parameters within the expanded layers. To achieve this, we first decouple each expanded layer into semantic *units* and evaluate their importance using gradient-based estimation (*Unit-wise Neuron Decoupling*), and then dynamically adjust learning rates for different units according to their importance scores during training (*Dynamic Learning Rate Adaptation*).

**Unit-wise Neuron Decoupling.** Guided by the heterogeneous importance of parameter units within layers, we performs *unit-level decoupling* on the expanded layers. Following the probing analysis in Section 2.3, we quantify unit importance $I_{\text{unit}}$ using gradient sensitivity signals (cf. Eq.5), which aggregate the first-order contributions of parameters $\theta_j$ to the training loss $\mathcal{L}$ via $\nabla_{\theta_j}\mathcal{L}$. A higher $I_{\text{unit}}$ indicates greater contribution to general competencies and thus warrants more conservative updates, whereas less important units are encouraged to adapt more aggressively to domain-specific signals.

**Dynamic Learning Rate Adaptation.** Based on the unit importance $I_{\text{unit}}$ in Eq.5, we assign adaptive learning rates to different units within the expanded layers:

$$\text{lr}_U = 2 \cdot (1 - I_{\text{unit}}) \cdot \text{lr}_{\text{base}}, \tag{7}$$

where $\text{lr}_{\text{base}}$ is the base learning rate, and the coefficient 2 normalizes the global scale to keep the effective average approximately unchanged. Units more important for general knowledge (higher $I_{\text{unit}}$) receive smaller learning rates to reduce overwriting, while less important units are encouraged to adapt more aggressively to domain-specific data. Training proceeds with the standard autoregressive objective: $\mathcal{L} = -\sum_{t=1}^{T} \log P(x_t \mid x_{<t}; \Theta)$. Since the importance of units may change as training progresses, we periodically recompute $I_{\text{unit}}$ and update learning rates accordingly, ensuring dynamic adaptation throughout learning. The full training procedure is provided in Appendix L. Appendix F.2 further shows that allocating learning rates inversely to unit importance minimizes an upper bound on general-domain forgetting. In essence, this design formalizes the intuition that highly general-critical units should be preserved via conservative updates, while less critical yet more adaptable ones can update more aggressively to absorb domain-specific information.

## 4 EXPERIMENT

### 4.1 EXPERIMENTAL SETUP

**Datasets.** We evaluate ADEPT across two domains, *Mathematics* and *Medicine*. For the mathematical domain, we use *OpenWebMath* (Paster et al., 2023), together with *AceReason-Math* (Chen et al., 2025), concatenated into the continual pretraining corpora. For the medical domain, we adopt the multilingual *MMedC* corpus (Qiu et al., 2024), together with *IndustryIns* and *MMedBench*, forming the medical pretraining corpora. Dataset statistics are provided in Appendix B.1 and Appendix B.2. In addition, for detecting general-knowledge-critical regions, we construct a *General Competence Detection Corpus*, following the same setting as in Section 2 and described in Appendix B.3.

**Baselines.** We compare ADEPT with a broad range of baselines from four perspectives:

- **Full-parameter tuning.** *PT-Full* directly updates all model parameters on the target corpora.
- **Replay-based tuning.** *Replay* mitigates catastrophic forgetting by mixing general-domain data into the training process (Que et al., 2024).

Table 1: Performance comparison across *Mathematical* and *Medical* domains. **Bold** numbers indicate the best performance, and underlined numbers denote the second best.

| Method | Mathematics | | | | | Medical | | | | |
| | General | | Domain | | | General | | Domain | | |
| | MMLU | CMMLU | GSM8K | ARC-Easy | ARC-Challenge | MMLU | CMMLU | MedQA | MMCU-Medical | CMB |
|---|---|---|---|---|---|---|---|---|---|---|
| *Qwen3-1.7B-Base* | | | | | | | | | | |
| Vanilla | 62.57 | 66.86 | 57.62 | 81.44 | 51.19 | 62.57 | 66.86 | 48.39 | 69.17 | 63.67 |
| PT-Full | 60.07 | 62.84 | 51.86 | 81.24 | 49.65 | 59.44 | 62.84 | 48.45 | 67.45 | 62.77 |
| Replay | 60.69 | 63.52 | 54.74 | 81.01 | 49.73 | 60.52 | 63.85 | 49.00 | 67.32 | 62.20 |
| Llama-Pro | 61.54 | 63.40 | 60.03 | 81.08 | 49.80 | 59.80 | 65.51 | 50.43 | 66.51 | 63.54 |
| PT-LoRA | 60.07 | 62.69 | 59.50 | 80.22 | 49.34 | 57.31 | 59.68 | 47.29 | 61.55 | 57.60 |
| TaSL | 60.34 | 62.95 | 59.07 | 79.76 | 48.89 | 62.48 | 66.14 | 47.06 | 67.62 | 61.15 |
| **ADEPT** | **62.62** | **67.06** | **70.51** | **82.48** | **52.62** | **62.80** | **66.89** | **50.75** | **71.98** | **65.43** |
| *Qwen3-4B-Base* | | | | | | | | | | |
| Vanilla | 73.19 | 77.92 | 69.07 | 85.52 | 59.13 | **73.19** | 77.92 | 62.77 | 82.44 | 78.92 |
| PT-Full | 70.33 | 73.07 | 60.96 | 85.31 | 57.59 | 69.48 | 72.77 | 62.84 | 81.34 | 76.88 |
| Replay | 70.46 | 73.72 | 63.91 | 85.06 | 57.68 | 70.74 | 73.81 | 63.55 | 80.60 | 76.74 |
| Llama-Pro | 72.42 | 77.39 | 73.16 | 85.14 | 57.76 | 72.28 | 77.28 | 62.53 | 81.20 | 78.12 |
| PT-LoRA | 70.20 | 72.90 | 71.34 | 84.18 | 57.25 | 72.73 | 76.78 | 61.59 | 80.49 | 76.92 |
| TaSL | 70.50 | 73.20 | 70.84 | 83.68 | 56.75 | 73.03 | 77.08 | 60.99 | 79.20 | 77.08 |
| **ADEPT** | **73.21** | **78.30** | **76.19** | **88.44** | **60.98** | 72.95 | **78.77** | **64.49** | **84.58** | **79.87** |
| *Qwen3-8B-Base* | | | | | | | | | | |
| Vanilla | **76.94** | 82.09 | 69.98 | 87.12 | 64.25 | **76.94** | 82.09 | 66.30 | 86.45 | 81.67 |
| PT-Full | 74.90 | 78.49 | 80.21 | 85.90 | 61.77 | 74.06 | 78.82 | 67.24 | 87.69 | 85.27 |
| Replay | 75.19 | 78.92 | 81.12 | 85.98 | 62.37 | 74.51 | 78.86 | 68.89 | 86.66 | 84.73 |
| Llama-Pro | 76.16 | 81.42 | 80.97 | 86.62 | 63.91 | 76.58 | 81.69 | 66.77 | 87.19 | 83.76 |
| PT-LoRA | 75.66 | 80.81 | 82.87 | 86.36 | 62.46 | 76.60 | 81.57 | 67.01 | 86.70 | 83.04 |
| TaSL | 76.63 | 80.37 | 80.54 | 84.81 | 59.09 | 76.42 | 81.86 | 66.51 | 86.20 | 82.54 |
| **ADEPT** | 76.80 | **82.11** | **83.87** | **89.29** | **64.51** | 76.77 | **82.11** | **69.24** | **89.84** | **85.80** |
| *Llama3-8B-Base* | | | | | | | | | | |
| Vanilla | 65.33 | 50.83 | 36.84 | 84.18 | 54.01 | **65.33** | 50.83 | 58.91 | 46.29 | 35.61 |
| PT-Full | 61.62 | 46.21 | 49.73 | 84.01 | 53.52 | 59.15 | 51.39 | 59.23 | 66.58 | 61.65 |
| Replay | 62.00 | **53.31** | 49.51 | 82.49 | 54.18 | 59.98 | **54.52** | 59.07 | 65.84 | 61.71 |
| Llama-Pro | 64.53 | 50.26 | 48.29 | 83.29 | 53.07 | 64.19 | 50.59 | 59.94 | 53.96 | 47.05 |
| PT-LoRA | 64.86 | 49.82 | 48.82 | 83.80 | 54.01 | 64.34 | 50.13 | 58.84 | 56.05 | 48.22 |
| TaSL | 65.16 | 50.11 | 35.43 | 83.29 | 53.51 | 64.64 | 50.43 | 55.55 | 58.34 | 47.69 |
| **ADEPT** | **65.35** | 51.90 | **50.57** | **84.96** | **55.52** | 65.17 | 51.92 | **61.17** | **67.03** | **61.78** |

- **Architecture expansion.** *LLaMA-Pro* (Wu et al., 2024b) expands the model by uniformly inserting new layers across the model, placing each new layer at fixed periodic intervals, while freezing the original weights. Only the newly introduced parameters are trained, enabling structural growth while preserving prior knowledge.
- **Parameter-efficient tuning.** *PT-LoRA* performs CPT using Low-Rank Adaptation (Hu et al., 2022), updating only a small set of task-adaptive parameters. *TaSL* (Feng et al., 2024a) extends PT-LoRA to a multi-task regime by decoupling LoRA matrices across transformer layers, allowing different subsets of parameters to specialize for different tasks.

See Appendix B.6 for implementation details of all baselines.

**Backbone Models.** To assess the generality of our method, we instantiate ADEPT on multiple backbone models, including *Qwen3-Base* (1.7B, 4B, 8B) (Yang et al., 2025) and *LLaMA3.1-8B-Base* (Dubey et al., 2024b), covering a wide range of parameter scales and architectural variants.

**Evaluation Metrics and Strategy.** We adopt multiple-choice question answering accuracy as the primary evaluation metric across all tasks (see Appendix B.9 for further details). For the **Mathematics** domain, we evaluate on *GSM8K* (Cobbe et al., 2021), *ARC-Easy* (Clark et al., 2018), and *ARC-Challenge* (Clark et al., 2018), which collectively span a wide range of reasoning difficulties. For the **Medical** domain, we use *MedQA* (Jin et al., 2021), *MMCU-Medical* (Zeng, 2023), and *CMB* (Wang et al., 2023b), covering diverse medical subjects and varying levels of complexity. Among them, *MedQA* is an English benchmark, while *MMCU-Medical* and *CMB* are in Chinese.

Table 2: Ablation study on ADEPT in *Medical* domain. **Bold** numbers indicate the best performance, and underlined numbers denote the second best.

| Method | Qwen3-1.7B-Base | | | | | Llama3-8B-Base | | | | |
|---|---|---|---|---|---|---|---|---|---|---|
| | MMLU | CMMLU | MedQA | MMCU-Medical | CMB | MMLU | CMMLU | MedQA | MMCU-Medical | CMB |
| **ADEPT** | **62.80** | **66.89** | **50.75** | **70.98** | **65.43** | **65.17** | **51.92** | **61.17** | **61.78** | **67.03** |
| *w/o Stage-1* | 57.31 | 59.68 | 47.29 | 61.55 | 57.60 | 57.88 | 50.76 | 58.32 | 53.32 | 60.32 |
| *w/o Stage-2* | 61.56 | 64.33 | 49.23 | 66.19 | 64.36 | 64.34 | 50.74 | 59.60 | 50.68 | 57.36 |
| *Uniform Expansion* | 59.80 | 65.51 | 50.43 | 66.51 | 63.54 | 64.19 | 50.59 | 59.94 | 47.05 | 53.96 |

To assess the model's ability to retain general-domain knowledge during continual pretraining, we additionally evaluate on *MMLU* (Hendrycks et al., 2020) and *CMMLU* (Li et al., 2023), two broad-coverage benchmarks for general knowledge and reasoning in English and Chinese, respectively.

## 4.2 EXPERIMENTAL RESULTS

**Performance Comparison.** As shown in Table 1, ADEPT consistently outperforms all CPT baselines across both mathematical and medical domains, confirming its effectiveness in domain-specific knowledge acquisition while substantially alleviating catastrophic forgetting. Concretely, **ADEPT achieves substantial domain-specific improvements.** Across all backbones and domain benchmarks, ADEPT consistently surpasses baselines, achieving the strongest performance. For instance, on *Qwen3-1.7B-Base*, ADEPT boosts *GSM8K* accuracy from 57.62% to 70.51%↑, bringing a large gain that highlights its advantage on enhancing LLMs' complex reasoning. Similarly, on *LLaMA3-8B-Base*, it drastically improves *CMB* accuracy improves from 35.61% to 61.78%↑, underscoring the strong enhancement of medical-domain capabilities. On average, ADEPT achieves up to **5.58%** gains over full-parameter CPT on target-domain benchmarks, confirming its advantage in domain knowledge acquisition. Furthermore, **ADEPT demonstrates clear advantages in mitigating catastrophic forgetting.** Whereas most baselines suffer noticeable degradation on general benchmarks such as *MMLU* and *CMMLU*, ADEPT preserves the pretrained LLMs' general-domain competencies, and in some cases even surpasses the vanilla backbone. Notably, with *Qwen3-4B* under medical CPT, ADEPT improves *CMMLU* accuracy from 77.92% to 78.77%↑. It also results in an average performance increase of **5.76%** on general benchmarks over full-parameter CPT. We attribute this to the disentanglement of domain-specific and general parameters, which prevents harmful representational interference during adaptation, ensuring that learning specialized knowledge does not corrupt the model's foundational abilities. Instead, this focused learning process appears to refine the model's overall competencies, leading to synergistic improvements on general-domain tasks. In summary, ADEPT offers a robust solution for CPT achieving superior domain adaptation while effectively preserving general knowledge.

**Ablation Study.** To investigate the effectiveness of each component in ADEPT, we conduct ablation experiments in the medical domain using two representative backbones, *Qwen3-1.7B* and *Llama3-8B*. In *w/o Stage-1*, we remove the *General-Competence Guided Selective Layer Expansion* and directly apply *Adaptive Unit-Wise Decoupled Tuning* on the $k$ *Domain-Adaptable Layers* without introducing any new parameters. In *w/o Stage-2*, we discard the dynamic decoupled tuning stage and instead directly fine-tune the expanded layers from *Stage-1*. In *Uniform Expansion*, we replace importance-guided expansion with uniformly inserting layers at fixed periodic intervals followed by fine-tuning, which is equivalent to the strategy adopted in LLaMA-Pro. As shown in Table 2, removing either *Stage-1* or *Stage-2* leads to clear degradation in both general and domain-specific performance, confirming that **both adaptive expansion and decoupled tuning are indispensable**. In particular, eliminating *Stage-1* results in the largest performance drop, suggesting that adaptive capacity allocation is crucial for enabling effective domain adaptation without sacrificing general-domain competencies. Meanwhile, replacing importance-guided expansion with uniform expansion yields inferior results, underscoring the advantage of expanding only the most domain-adaptable layers.

**Decoupling Effectiveness on Expanded Parameters.** We visualize cross-domain activations using Kernel Density Estimation (KDE) (Silverman, 2018), sampling 500 instances from both *Medical* and *General* corpora. For the original *Qwen3-8B-Base* (left in Figure 4), the most domain-adaptable layer (lowest $I_{layer}$) still shows heavy overlap between *general* and *medical* activations, evidencing

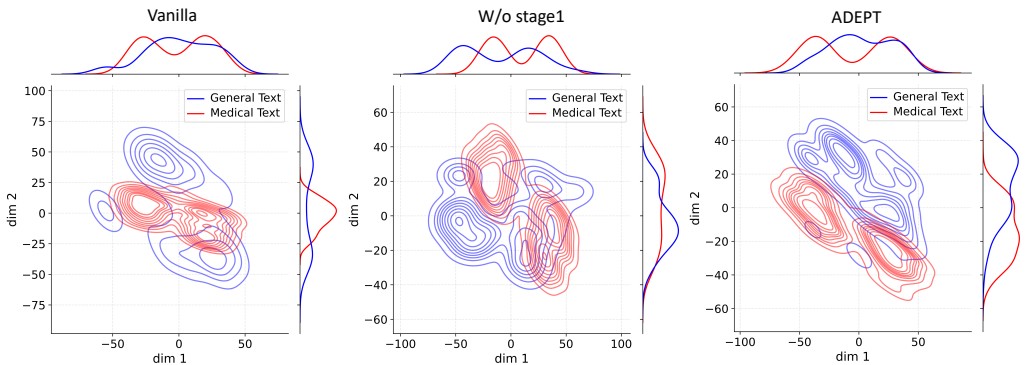

Figure 4: Activation distribution analysis of Qwen3-8B.

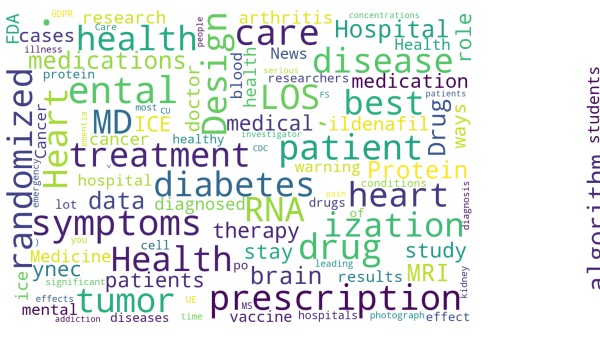

(a) Token distributions shift in *Medical*

(b) Token distributions shift in *Mathematical*

Figure 5: Token distribution shifts across domains. Word cloud visualizations of shifted tokens reveal that ADEPT achieves highly focused alignment, with most changes concentrated on domain-specific terminology.

strong parameter coupling. Direct decoupling without expansion (w/o Stage-1, middle) on the same layer fails to reduce this entanglement, confirming that pretrained parameters are inherently difficult to separate. In contrast, after expansion (right), the duplicated layers serve as a "blank slate," yielding clearly separated activations across domains. Additional analyses on more backbones are provided in Appendix C.1, where we observe that this trend consistently holds across nearly all evaluated LLMs, further validating the generality of our approach.

**Token Distribution Shift Analysis.** To assess how ADEPT injects domain knowledge while preserving general competencies, we analyze token-level shifts between the base and continually pretrained models. Following Lin et al. (2024), tokens are categorized as *unshifted*, *marginal*, or *shifted*. Only a small proportion of tokens shift, while most remain unchanged, indicating stable adaptation. In the medical domain, merely 2.18% shift (vs. 5.61% under full pretraining), largely medical terms such as "prescription," "diagnosis," and "therapy" (Figure 5a). In the mathematical domain, only 1.24% shift, mainly scientific terms such as "theorem" and "equation" (Figure 5b). Further details and analyses are provided in Appendix I. These results demonstrate that ADEPT achieves precise and economical domain knowledge injection while minimizing perturbation to general competence.

**Extended Investigations and Key Insights.** We further investigate several design choices of ADEPT in appendix: In Appendix E, we investigate alternative strategies for probing layer importance and observe the consistency of different measurement methods, offering insight into how importance estimation affects adaptation outcomes. Appendix G explores the effect of expanding different numbers of layers and reveals how the number of expansion layers should be selected under different circumstances and the potential reasons behind this. Appendix H shows that even with relatively low-quality importance detection corpus from pretrain data, our approach maintains strong generalization across domains, suggesting the robustness of ADEPT. Appendix J demonstrates our insights into the potential for merging expanded layers that are independently trained on

different domains, offering an intriguing direction for achieving multi-domain adaptation with minimal catastrophic forgetting. In addition, Appendix B.8 analyzes the training efficiency of ADEPT, showing that our selective updating design substantially accelerates convergence compared to baselines. In addition to the core evaluation, we conduct a comprehensive set of extended analyses to further validate the robustness, generality, and adaptability of ADEPT. In Appendix M, we present a sensitivity analysis of the importance-score update intervals, demonstrating that ADEPT is stable across a wide range of update frequencies, with only marginal performance variation. Appendix N investigates the applicability of ADEPT to supervised fine-tuning settings, showing consistent gains over standard fine-tuning baselines without requiring architectural changes. To assess generalization beyond our primary benchmarks, Appendix O includes extended evaluations on additional domains and datasets, where ADEPT continues to outperform strong baselines. In Appendix P, we evaluate ADEPT specifically on code-domain tasks, confirming its effectiveness in structured, logic-intensive environments. Appendix Q further extends our evaluation to multilingual medical benchmarks, highlighting ADEPT's cross-lingual transfer capability in more challenging domain adaption setting. Finally, Appendix R addresses a key design question: whether to expand domain-critical layers or general-noncritical ones. Appendix S investigated the impact of different zero-initialization strategies. Our analysis reveals that selectively expanding domain-critical layers yields significantly higher domain performance but more forgetting, providing actionable guidance for layer selection in domain adaptation scenarios.

## 5 CONCLUSIONS AND FUTURE WORKS

We present ADEPT, a framework for LLM continual pretraining for domain adaptation that effectively tackles catastrophic forgetting, leveraging functional specialization in LLMs. By selectively expanding layers less critical to the general domain and adaptively updating decoupled parameter units, ADEPT minimizes catastrophic forgetting while efficiently incorporating domain-specific expertise. Our experiments show significant improvements in both domain performance and general knowledge retention compared to baselines. Future work could focus on refining the decoupled tuning mechanism, designing more sophisticated learning rate strategies beyond linear mapping to allow for more precise adjustments. Another direction is to explore better dynamic and real-time methods for measuring parameter importance during training.

## 6 ETHICS STATEMENT

All datasets used for training and evaluation in this study are publicly available versions obtained from the Hugging Face platform. The datasets have been curated, cleaned, and de-identified by their respective data providers prior to release. No patient personal information or identifiable medical data is present. Consequently, the research does not involve human subjects, and there are no related concerns regarding privacy, confidentiality, or legal liability. And for full transparency, we report all aspects of large language model (LLM) involvement in the Appendix K.

We strictly adhered to the usage and redistribution licenses provided by the original dataset authors and hosting platforms. Our research poses no risk of harm to individuals or groups and does not contain any potentially harmful insights, models, or applications. Additionally, there are no conflicts of interest or sponsorship concerns associated with this work. We are committed to research integrity and ethical standards consistent with the ICLR Code of Ethics.

## 7 REPRODUCIBILITY STATEMENT

We actively support the spirit of openness and reproducibility advocated by ICLR. To ensure the reproducibility of our research, we have taken the following measures:

1. Disclosure of Base Models: All base models used in our experiments are explicitly identified and described in the main text. This allows readers to directly reference and obtain these models.

2. Datasets and Experimental Details: All experiments are conducted on publicly available datasets from the Hugging Face platform. In Appendix B, we provide a comprehensive description of our experimental implementation, including dataset sources, browser links, and detailed data processing procedures. We also detail the experimental setup, such as training duration, hardware envi-

ronment (e.g., GPU type), and configuration of hyperparameters, including LoRA_rank, number of extended layers, batch_size, and max_length. These details facilitate transparent verification and replication of our results.

3. Open-Source Code Release: To further support reproducibility, we release all training and evaluation code in a public repository (https://github.com/PuppyKnightUniversity/ADEPT). The repository contains clear instructions on installation, data downloading, preprocessing, and experimentation, allowing interested researchers to replicate our results with minimal effort.

We believe that these actions align with the open science principles championed by the ICLR community, and we are committed to supporting the reproducibility and transparency of our work.

## ACKNOWLEDGMENTS

This research was funded by the Independent Research Project of the National Key Laboratory of Big Data and Decision.

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

## A  RELATED WORK

### A.1  CONTINUAL PRETRAINING FOR LLMS

Continual pretraining updates pretrained LLMs with new corpora to equip them with new knowledge and capabilities. Data-centric approaches adopt data replay to mitigate catastrophic forgetting (Huang et al., 2024; Zhang et al., 2025; Xiong et al., 2023; Song et al., 2023), or utilize data construction strategies to synthesize training corpora (Yang et al., 2024; Arbel et al., 2024). However, these methods make no changes to the model or training procedure, failing to effective inject new knowledge due to capacity saturation and only partially alleviating forgetting. Another line of works focus on adjusting model architecture and training strategy. LoRA (Hu et al., 2022) improve efficiency for fine-tuning by adapting low-rank updates on top of frozen backbones, but their limited adjustments to LLMs can not effectively address continual pretraining for deep domain adaptation. LLaMA-Pro (Wu et al., 2024b) expands model blocks and tunes the added parameters on new corpora, improving knowledge injection and mitigating forgetting compared to vanilla CPT. Yet existing expansion policies insert layers uniformly across depths and treat all expanded parameters indiscriminately during optimization, leaving open how to place capacity where domain signals concentrate and update it without disturbing general knowledge. Classical continual-learning regularizers (Kirkpatrick et al., 2017) constrain updates on weights deemed important to previous tasks, but they do not guide where capacity allocation nor how to target LLM domain adaptation learning.

### A.2  FUNCTIONAL SPECIALIZATION IN LLMS

Growing evidence indicates that, akin to human brains, LLMs exhibit functional specialization, where different regions such as layers, attention heads and neurons play distinct roles. A series of causal and studies show that factual knowledge are predominantly stored in FFN layers (Dai et al., 2022c), and attention heads usually play specialized roles for certain functions (Zheng et al., 2024),

suggesting that knowledge and skills are unevenly distributed in LLMs. Inspired by this specialization, several methods have tried to decouple functional modules during training. For instance, Parenting (Xu et al., 2025) separates the subspaces responsible for evidence-following and noise-robustness in retrieval-augmented generation, and optimizes them with tailored objectives to improve performance under noisy retrieval. Similarly, TaSL (Feng et al., 2024a) addresses multi-task adaptation by disentangling LoRA parameters from different tasks and merging them in a weighted manner, which helps reduce interference. Other works on orthogonal (Wang et al., 2023a) or decomposed LoRA (Liu et al., 2024b) further reflects the idea that training different parameter subspaces separately improves robustness and transfer. Despite these advances, prior work does not address CPT, where the tension between knowledge injection and retention needs to be tackled. To our knowledge, our work is the first to explicitly leverage functional specialization during CPT to simultaneously improve domain performance and alleviate catastrophic forgetting.

## B  DATA RECIPE AND EXPERIMENT SETTINGS

To demonstrate the applicability and generalizability of our approach, we conducted domain-adaptive continual pretraining experiments on two distinct and highly significant domains: Mathematics and Medicine, both of which play crucial roles in the advancement of artificial intelligence and the applications of LLM. The mathematical domain often poses challenges that emphasize a model's reasoning and computational abilities, while the medical domain predominantly requires a deep understanding and memorization of medical concepts. From a cognitive perspective, we believe that the capabilities that need to be infused into the model differ significantly between these two domains, which further demonstrates the generalisability of our approach.

The continual pretraining process leverages both pretraining datasets for foundational knowledge and supervised fine-tuning (SFT) datasets for task-specific optimization (Cheng et al., 2023). Below, we detail the data composition and processing details. **All data used will be processed into the format of pre-training data.**

### B.1  MEDICAL PRETRAIN DATA SOURCE

Our medicine datasets are divided into pre-training data, designed to provide extensive general knowledge, and supervised fine-tuning (SFT) data, which refine the model's understanding for specific instructions in the medicine domain (will be converted to pretrain data format when training).

- Pre-training data: we utilize English and Chinese portions of MMedC dataset, a multilingual medical dataset, furnishing a total of 14.3 billion tokens.
- Instrution tuning data: we incorporate two supervised datasets:
  1. IndustryIns, contributing 1.6 billion tokens from instruction-based examples
  2. MMedBench, with 18 million tokens focused on medical reasoning tasks.

Table 3: Overview of medicine Datasets. This table summarizes medicine-specific pre-training and SFT datasets, including their language coverage, dataset links, and used token counts. For MMedC, we only use the English and Chinese parts and we only use the *Health-Medicine* subset.

| Dataset Name | Dataset Type | Language | Dataset Link | #Token Used |
|---|---|---|---|---|
| MMedC | Pre-training | Multilingual | Henrychur/MMedC | 14.3B |
| IndustryIns | SFT | Chinese and English | BAAI/IndustryInstruction | 1.6B |
| MMedBench | SFT | Chinese and English | Henrychur/MMedBench | 18M |

### B.2  MATHEMATICS PRETRAIN DATA SOURCE

Mathematics pretrain datasets include both pre-training and fine-tuning data (will be converted to pretrain data format when training), structured similarly to the medicine datasets.

- Pre-training data: we use the Open-Web-Math (Paster et al., 2023) dataset, containing a diverse set of general mathematics knowledge amounting to 14.7 billion tokens.
- For Instruction-tuning data: we use the AceReason-Math (Chen et al., 2025), contributing 102 million tokens, with a strong emphasis on chain-of-thought reasoning and problem-solving.

Table 4: Overview of Mathematics Datasets. This table includes the pre-training and SFT datasets for mathematical reasoning, highlighting their contents, links, and used token counts.

| Dataset Name | Dataset Type | Language | Dataset Link | Used Token |
|---|---|---|---|---|
| Open-Web-Math | Pre-training | English | open-web-math/open-web-math | 14.7B |
| AceReason-Math | SFT | English | nvidia/AceReason-Math | 102M |

### B.3 GENERAL COMPETENCE DETECTION CORPUS

To accurately probe which parameters are critical for preserving general knowledge during continual pretraining, we construct a *General Importance Detection Corpus*. This corpus is designed to capture both broad world knowledge and instruction-following capability in English and Chinese. Specifically, we include the development splits of two widely recognized multi-task benchmarks, `MMLU_dev` and `CMMLU_dev` to capture general knowledge without data leakage.

MMLU and CMMLU are formatted as multiple-choice question answering tasks with explicit prompts and ground-truth answers. For these, we compute gradient-based importance only on the target answer tokens to avoid biases from prompt formatting, thereby capturing each parameter group's contribution to accuracy.

To clarify how gradient signals are obtained, we illustrate two examples. In SFT-style corpora (e.g., MMLU, CMMLU), only the ground-truth answer token contributes to gradient computation, ensuring clean signals for decision-making importance. In PT-style corpora (e.g., FineWeb_Edu), all tokens contribute under the causal LM objective, providing dense gradients that reflect general modeling capacity. Examples are shown in Example 1 and Example 2.

Table 5: General Competence Detection Corpus. #Examples means the number of examples we used.

| Dataset | Language | #Examples | Hugging Face Link |
|---|---|---|---|
| MMLU_dev | English | 285 | cais/mmlu |
| CMMLU_dev | Chinese | 295 | haonan-li/cmmlu |

The statistics of the selected datasets are summarized in Table 5.

---

**Example 1**

**Gradient Flow in SFT Data for Importance Estimation**
**Input Prompt:**
Question: Find all $c \in \mathbb{Z}_3$ such that $\mathbb{Z}_3[x]/(x^2 + c)$ is a field.
A. 0    B. 1    C. 2    D. 3

Answer:
B

**Explanation:**
*In this SFT setup, only the **target answer token** (e.g., B) is used to compute gradients for parameter importance. The input question and options are excluded from gradient computation to avoid encoding biases from instruction formatting. By focusing gradient signals solely on the correct answer token, we measure how each parameter contributes to decision-making accuracy under structured knowledge tasks, while preventing overfitting to input patterns and ensuring clean separation between training and probing data.*

---

---

**Example 2**

---

**Gradient Flow in PT Data for Importance Estimation**
**Context (Compute Gradient):**
The heart is a muscular organ responsible for pumping blood throughout the body. It consists of four chambers: the left and right atria, and the left and right ventricles. Oxygen-poor blood enters the right atrium, then flows to the right ventricle, which pumps it to the lungs. After oxygenation, blood returns to the left atrium, moves to the left ventricle, and is finally pumped into the aorta for systemic circulation. This process is regulated by electrical signals originating in the sinoatrial node. These signals ensure synchronized contraction and efficient blood flow.

**Explanation:**
*In PT-style training, parameter importance is computed using causal language modeling loss across the entire sequence. Every token — both context and continuation — contributes to the gradient signal. This captures how parameters support general language modeling over natural text distributions. Unlike SFT, there is no explicit input/output separation; instead, each token is predicted from its prefix, making the gradient flow dense and continuous. This allows us to assess parameter sensitivity in open-ended, domain-relevant pre-training scenarios such as those provided by* `FineWeb_Edu`.

---

### B.4 DATA PROCESSING

To generate training corpus in pretrain format, SFT data is structured by concatenating questions, chain-of-thought (CoT) reasoning, and final answers for each instance. This ensures that the model is optimized for multi-step reasoning tasks common in medicine applications. We take Example 3 as an example.

---

**Example 3**

---

**Problem:** On Liar Island, half the people lie only on Wednesday, Friday, and Saturday, while the other half lie only on Tuesday, Thursday, and Sunday. One day, everyone on the island says: "I will tell the truth tomorrow." What day is it? (2021 Xin Xiwang Bei Competition, Grade 2, Preliminary Math Exam)

**Analysis:** We examine the truth-telling patterns over the week:

- First group (lies on Wed, Fri, Sat): Truth pattern across 7 days: True, True, False, True, False, False, True.

- Second group (lies on Tue, Thu, Sun): Truth pattern: True, False, True, False, True, True, False.

Now evaluate each option:

**Option A (Tuesday):** If today is Tuesday, the first group tells the truth today, so their statement "I will tell the truth tomorrow" implies they should tell the truth on Wednesday. But they lie on Wednesday — contradiction. The second group lies today, so their statement is false, meaning they will *not* tell the truth tomorrow (i.e., lie on Wednesday). But they actually tell the truth on Wednesday — also a contradiction. So A is invalid.

**Option B (Wednesday):** First group lies today; their statement is false → they will *not* tell the truth tomorrow (i.e., lie on Thursday). But they tell the truth on Thursday — contradiction. Second group tells the truth today → they should tell the truth on Thursday. But they lie on Thursday — contradiction. So B is invalid.

**Option C (Friday):** First group lies today → statement is false → they will *not* tell the truth tomorrow (i.e., lie on Saturday). They do lie on Saturday — consistent. Second group tells the truth today → they will tell the truth on Saturday. They do tell the truth on Saturday — consistent. So C is correct.

**Option D (Saturday):** First group lies today → should lie on Sunday. But they tell the truth on Sunday — contradiction. Second group tells the truth today → should tell the truth on Sunday. But they lie on Sunday — contradiction. So D is invalid.

---

> **Option E (Sunday):** First group tells the truth today $\rightarrow$ should tell the truth on Monday. They do — consistent. Second group lies today $\rightarrow$ their statement is false $\rightarrow$ they will *not* tell the truth on Monday (i.e., lie). But they tell the truth on Monday — contradiction. So E is invalid.
> Therefore, the correct answer is **C** (Friday).
>
> *[This example demonstrates how structured SFT data — consisting of a standalone* `problem` *(in blue), detailed step-by-step* `analysis` *(in green) and a short answer (in red) — is concatenated into a single coherent narrative. In PT-style training, such concatenation enables models to learn implicit reasoning patterns from natural language flow, bridging supervised fine-tuning signals with pre-training objectives.]*

To handle input sequences that exceed the maximum context length of 4096 tokens imposed by transformer-based models, we apply a sliding window segmentation strategy with overlap, following the approach used in DATAMAN (Peng et al., 2025). For any sequence longer than 4096 tokens, we split it into multiple segments, each of length at most 4096, using a sliding window with a stride of 3072 tokens and an overlap of 1024 tokens (i.e., 1/4 of the window size). This ensures that consecutive segments share contextual information when training in the same or adjacent batches, preserving semantic continuity and high data utilization rate across boundaries.

Formally, given a token sequence $D = [t_1, t_2, \ldots, t_L]$ of length $L > 4096$, we generate $K = \lceil \frac{L-1024}{3072} \rceil$ segments. The $k$-th segment is defined as $S_k = D[\ell_k : r_k]$, where $\ell_k = (k-1) \cdot 3072 + 1$ and $r_k = \min(\ell_k + 4097, L)$. The overlapping region between $S_k$ and $S_{k+1}$ consists of the last 1024 tokens of $S_k$, which are identical to the first 1024 tokens of $S_{k+1}$.

This method prevents information loss due to truncation and allows the model to learn from continuous context during training. The 1024-token overlap helps maintain coherence at segment boundaries, which is crucial for tasks requiring long-range understanding, while keeping computational overhead manageable.

## B.5 FINAL DATA ORGANIZATION SCHEME

Our final training data is organized as follows:

1. English pre-training corpus
2. Chinese pre-training corpus (if have)
3. English supervised fine-tuning (SFT) corpus
4. Chinese SFT corpus (if have)

This organization is motivated by several key points in Qwen3 Technical Report (Yang et al., 2025) and Llama3 Technical Report (Dubey et al., 2024a). First, we follow the principle that high-quality data (SFT data in our work) should be used after extensive pre-training on large-scale general corpora, allowing the model to first acquire broad knowledge and language structure, and then specialize on more curated tasks and instructions.

What's more, according to the technical reports, it is further beneficial to place the same language's data together during training—this maximizes the coherence within each mini-batch and reduces unintended cross-lingual transfer until later stages. Most LLMs are dominated by English corpora in their pre-training phase, supporting the choice of placing English data first. Finally, during later training stages, continued training and decay are performed on SFT examples, which aligns with established recipes for improving supervised task performance.

## B.6 COMPARED METHODS.

- **Full-parameter tuning.** *PT-Full* directly updates all model parameters on the target corpus, serving as the most straightforward yet commonly used baseline for continual pretraining.
- **Replay-based tuning.** *Replay* mitigates catastrophic forgetting by mixing general-domain data into the continual pretraining process (Que et al., 2024), thereby preserving part of the original knowledge distribution while adapting to the new domain. Following (Zhang et al., 2025), based

on the data from Data Recipe, we randomly sampled totally 1.91B data from FinewebEdu and FinewebEdu-Chinese at a ratio of 7:3, and randomly shuffled them into the domain-specific data, helping the model better recall general domain knowledge.

- **Architecture expansion.** *LLaMA-Pro* (Wu et al., 2024b) expands the model by uniformly inserting new layers into each transformer block while freezing the original weights. Only the newly introduced parameters are trained, enabling structural growth while preserving prior knowledge.
- **Parameter-efficient tuning.** *PT-LoRA* performs continual pretraining using Low-Rank Adaptation (Hu et al., 2022), updating only a small set of task-adaptive parameters. *TaSL* (Feng et al., 2024a) extends PT-LoRA to a multi-task regime by decoupling LoRA matrices across transformer layers, allowing different subsets of parameters to specialize for different tasks. This enables more fine-grained adaptation to domain-specific signals. We used the DEV sets of MMLU and CMMLU to assess general capabilities, and their mathematics and medical subsets to specifically evaluate mathematical and medical competencies, respectively. Taking the medical domain as an example, we treat the original model as one equipped with a LoRA module initialized to all zeros. The final LoRA module is then obtained by merging the domain-specific LoRA with the original (empty) LoRA using TaSL.

### B.7 EXPERIMENTAL IMPLEMENTATION.

We conduct our all pre-training experiments on the Qwen3-1.7B-Base/Qwen3-4B-Base/Qwen3-8B-Base/Llama3-8B model with the following hyperparameter configuration. Training is performed for 3 epochs using a batch size of 512 (8 NVIDIA H800 GPUs) and a maximum sequence length of 4096 tokens. We utilize a cosine learning rate scheduler with an initial learning rate of 3.0e-5 and a warmup ratio of 0.03. Optimization is performed in bf16 precision.

For methods requiring block expansion, we expand 4 layers; for methods based on LoRA, we set the LoRA rank to 256 to ensure the number of trainable parameters is roughly comparable between the two approaches. For the medicine injection into Llama models, which have poor Chinese support, we expand 8 layers for block expansion methods and set the LoRA rank to 512 for LoRA-based methods.

For our ADEPT, we calculate the importance score and update learning rate per 500 iterations. (It does not affect the impact of warmup, decay scheduler on the learning rate, but only performs a reallocation.)

### B.8 EFFICIENCY ANALYSIS OF ADEPT FOR MEDICAL APPLICATIONS

Table 6: Training Time Comparison in the Medical Domain. We select representative baselines including full-parameter (PT-Full) training, PT-Lora, and Llama Pro to validate the effectiveness of our method. The **bold** entries denote the optimal results.

|  | Qwen3-1.7B | Qwen3-4B | Qwen3-8B | Llama3-8B |
|---|---|---|---|---|
| PT-Full | 2 days, 17h | 5 days, 14h | 8 days, 9h | 7 days, 22h |
| ADEPT | **1 day, 9h** | **2 days, 11h** | **3 days, 15h** | **3 days, 19h** |
| PT-Lora | 3 days, 0h | 6 days, 4h | 8 days, 23h | 8 days, 2h |
| Llama Pro | 2 days, 1h | 3 days, 14h | 5 days, 8h | 4 days, 21h |

As shown in the Table 6, our ADEPT approach achieves the fastest training time across all tested model sizes, with Llama Pro being the next most efficient competitor. The substantial efficiency gain of our method is mainly attributed to its design: ADEPT only updates a small subset of parameters, primarily located in the deeper layers of the network. This structure allows the backward computation graph to terminate earlier, significantly reducing the overall training time.

We further analyze two aspects that explain and quantify the practical efficiency of ADEPT: (1) the runtime overhead of the importance-probing steps (layer masking and unit-level gradient probing) under single- and multi-GPU execution; and (2) the scaling behavior of training time when varying the number of expanded layers. These measurements complement Table 6 and clarify why ADEPT achieves shorter end-to-end training times (including both probing and training) in practice.

**Probing overhead.**   The importance-probing in ADEPT comprises: a one-time layer-importance pass (layer masking) and periodic unit-level gradient probing. Both operations are lightweight relative to full training: layer masking is computed once before the main CPT loop and is fully parallelizable, and gradient probing requires only a single backward pass per probe interval (every 500 steps). Table 7 reports wall-clock times for these two components on single-GPU and 8-GPU setups, along with a representative total backpropagation time during training. All measurements use the same hardware configuration and identical probing data (CMMLU + MMLU dev subsets, ≈580 examples). The results demonstrate that (1) layer masking is trivially parallelizable across devices and thus benefits nearly linearly from multi-GPU execution; and (2) gradient probing is a small fraction of the overall training backpropagation time.

Table 7: Wall-clock time for layer masking and unit-level gradient probing on single-GPU and 8-GPU settings. 'Total Backprop (Train)' reports the total backward probing time during training.

| Model | Layer Mask (1 GPU) | Layer Mask (8 GPUs) | Grad Probe (1 GPU) | Grad Probe (8 GPUs) | Total Backprop (Train) |
|---|---|---|---|---|---|
| Qwen3-1.7B-Base | 36m30s | 8m16s | 2m22s | 1m50s | 16m20s |
| Qwen3-4B-Base | 1h10m | 17m39s | 3m14s | 2m10s | 25m43s |
| Qwen3-8B-Base | 1h32m | 23m08s | 5m24s | 2m33s | 29m03s |
| Llama3-8B | 1h24m | 21m47s | 6m26s | 3m19s | 40m42s |

Combined probing time is small relative to the total training backpropagation time, so probing does not meaningfully affect end-to-end efficiency, as the time saved due to ADEPT's reduced backpropagation significantly outweighs the probing overhead.

**Scaling analysis.** We next quantify how training time scales as we increase the number of expanded layers. Because each expanded Transformer layer contributes a known parameter and compute footprint, wall-clock training time increases approximately linearly with the number of expanded layers in our implementation. This near-linear behavior enables straightforward time-constrained auto-tuning: given a time budget, one can estimate an upper bound on the number of layers that may be expanded. Table 8 provides empirical training times for expanding different numbers of layers in Qwen3 base models. These times were measured under the same training configuration used for Table 6, isolating the effect of expansion count.

Table 8: Training time when expanding different numbers of layers in Qwen3 models.

| Model | 1 Layer | 2 Layers | 4 Layers | 8 Layers | 16 Layers |
|---|---|---|---|---|---|
| Qwen3-1.7B-Base | 24h | 1d 2h | 1d 9h | 2d 1h | 2d 20h |
| Qwen3-4B-Base | 1d 17h | 2d 4h | 2d 11h | 3d 4h | 4d 10h |
| Qwen3-8B-Base | 2d 18h | 3d 2h | 3d 15h | 4d 6h | 6d 22h |

The empirical timings show near-linear increases in total training time with expanded-layer count for all evaluated model sizes. Notably, Qwen3-8B-Base shows a sharp time increase beyond 8 expanded layers, not due to algorithmic nonlinearity but GPU memory limits that force smaller batch sizes and thus longer training. Time estimates should therefore account for compute resources. Still, our measured times offer practical guidance for layer scaling and selecting expansion size under a fixed time budget.

**Summary.**   The additional measurements above demonstrate that (1) the runtime overhead of our importance-probing is small and highly parallelizable, and (2) the training-time cost of expanding more layers grows predictably and near-linearly. Together with the fact that ADEPT updates only a fraction of parameters, these behaviors explain the consistent end-to-end time savings reported in Table 6.

## B.9    EVALUATION SETTING

We evaluate the performance of large language models on multiple-choice question answering tasks using accuracy as the primary metric. For a given question with $N$ candidate options (typically $N = 4$, labeled A, B, C, D), the model's prediction is determined by computing the sequence-level likelihood of each option when appended to the question stem.

Specifically, let $Q$ denote the input question and $O_i$ represent the $i$-th answer option (e.g., `A. True`, `B. False`). The model computes the conditional probability of the full sequence $Q \parallel O_i$ (i.e., the concatenation of the question and the $i$-th option) under the causal language modeling objective. We calculate the average negative log-likelihood (or perplexity, PPL) of the tokens in $O_i$ given $Q$:

$$\text{PPL}(O_i \mid Q) = \exp\left(-\frac{1}{|O_i|} \sum_{t=1}^{|O_i|} \log P(o_t \mid Q, o_1, \ldots, o_{t-1})\right) \quad (8)$$

The model selects the option with the lowest perplexity as its predicted answer:

$$\hat{y} = \arg\min_{O_i \in \{A,B,C,D\}} \text{PPL}(O_i \mid Q) \quad (9)$$

This method, often referred to as *perplexity-based decoding*, does not require fine-tuning or additional parameters and is widely used for evaluation of base models. It leverages the pre-training objective directly by predicting the next token, making it particularly suitable for evaluating general knowledge in base LLMs.

Finally, accuracy is defined as the percentage of questions for which the model's predicted answer matches the ground-truth label:

$$\text{Accuracy} = \frac{1}{M} \sum_{j=1}^{M} \mathbb{I}(\hat{y}_j = y_j) \quad (10)$$

where $M$ is the total number of test questions, $\hat{y}_j$ is the model's prediction on the $j$-th question, $y_j$ is the true label, and $\mathbb{I}(\cdot)$ is the indicator function.

For our experiments, we evaluate all model checkpoints using the lm_harness[1] framework. For the **Mathematics** domain, we adopt the default configurations of `GSM8K_cot`, `ARC-Easy`, and `ARC-Challenge`. For the **Medical** domain, we design custom configuration files for `MedQA`, `MMCU-Medical`, and `CMB`, following the official evaluation protocols of `MMLU` and `CMMLU`. For the **General** domain, we directly evaluate on `MMLU` and `CMMLU`. In all cases, we use 5-shot prompts and greedy decoding (temperature = 0) for inference. This standardized evaluation protocol ensures fair comparison across models and tasks.

## C  MODEL PARAMETER GROUP

To enable efficient and semantically meaningful parameter decoupling during fine-tuning, we partition the model parameters into modular units based on their functional roles within the transformer architecture. Given the substantial number of model parameters, extremely fine-grained control at the neuron level—as used in methods like DAS (Ke et al., 2023)—is computationally prohibitive and contradicts the goal of parameter-efficient adaptation. Moreover, such fine granularity often leads to training instability due to noisy importance estimation.

On the other hand, treating an entire layer as a single unit (e.g., standard LoRA) is too coarse and lacks semantic discrimination. While TaSL (Feng et al., 2024b) proposes decomposing LoRA into `LoRA_A` and `LoRA_B`, this approach is specific to low-rank adapters and does not generalize well to full-layer decomposition.

To strike a balance between granularity and efficiency, we introduce a **semantic-aware module partitioning strategy**, which divides each transformer layer into multiple functional units according to their architectural semantics. This design allows us to manipulate parameters at a meaningful intermediate level—finer than whole layers, but coarser than individual neurons—achieving a practical trade-off between controllability and computational feasibility.

Table 9 presents the detailed parameter grouping scheme used in this work, exemplified on the LLaMA architecture.

---

[1] https://github.com/EleutherAI/lm-evaluation-harness

Table 9: Model Parameter Grouping Scheme

| Parameter Type | Parameter Name | Description |
|---|---|---|
| Attention | self_attn.q_proj.weight | Query projection weight; maps input to query space |
| | self_attn.k_proj.weight | Key projection weight; maps input to key space |
| | self_attn.v_proj.weight | Value projection weight; maps input to value space |
| | self_attn.o_proj.weight | Output projection weight; projects attention output back to target dimension |
| MLP | mlp.gate_proj.weight | Gating projection weight; controls information flow in SwiGLU activation |
| | mlp.up_proj.weight | Up-projection weight; maps features to higher-dimensional intermediate space |
| | mlp.down_proj.weight | Down-projection weight; projects features back to original dimension |
| LayerNorm | input_layernorm.weight | Input layer normalization weight; normalizes input before attention |
| | post_attention_layernorm.weight | Normalization weight after attention; stabilizes post-attention outputs |

As shown in Table 9, each transformer layer is decomposed into three primary functional modules: *Attention*, *MLP*, and *LayerNorm*. Within each module, parameters are grouped by their semantic role:

- The **Attention** module includes all four linear projections ($Q$, $K$, $V$, $O$), which collectively handle context modeling through self-attention.
- The **MLP** module contains the up, gate, and down projection layers, responsible for non-linear feature transformation.
- The **LayerNorm** components are kept separate due to their distinct role in stabilizing activations and gradient flow.

This grouping enables targeted manipulation of specific sub-functions (e.g., disabling attention outputs or freezing normalization statistics) while maintaining training stability and interpretability.

### C.1 COMPATIBILITY BETWEEN LAYER EXPANSION AND DECOUPLING

First, we would like to share our understanding of the Compatibility between Layer Expansion and Decoupling:

1. Although layer expansion can minimize changes to the original parameter space, this alone makes it difficult to fully prevent model drift during long-term pre-training. Parameter decoupling offers a more fine-grained means of controlling this phenomenon.

2. Since our models are pre-trained on a large corpus, their parameter space is inherently uncontrollable, making thorough decoupling of the original model parameters challenging. In contrast, the newly expanded parameters initially contribute nothing to the model's output. As we continue domain-specific training in the medical field, gradually decoupling these new parameters is more conducive to achieving complete decoupling.

To examine the effectiveness of our layer extension strategy, we conduct activation distribution analysis across multiple backbones. For each model, we first identify the most domain-adaptable layer (i.e., the layer with the lowest $I_{\text{layer}}$). We then randomly sample 500 instances from both the *Medical* and *General* corpora, compute activations at the selected layer, and visualize their distributions using Kernel Density Estimation (KDE). The following three configurations are compared: (1) the origi-

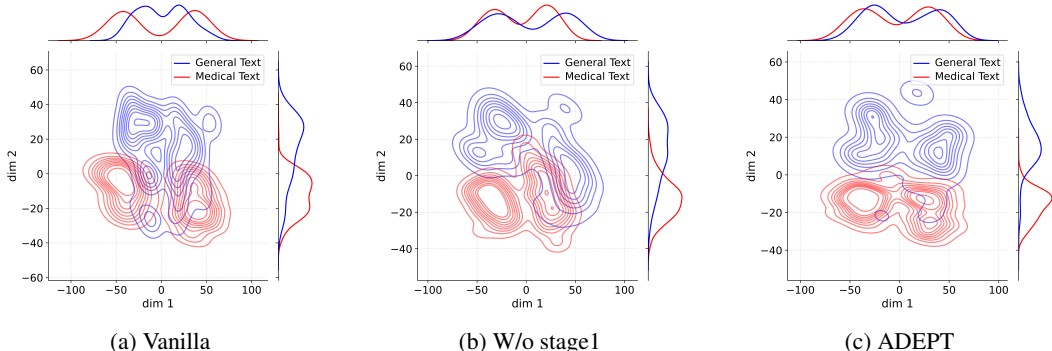

Figure 6: Kernel Density Estimation of activations for Qwen3-1.7B-Base under different configurations. Our layer extension strategy enables effective parameter decoupling. Expanded layers: 22, 23, 25, and 27.

nal base model, where we visualize the most domain-adaptable layer; (2) direct decoupling without expansion (w/o Stage-1), where we visualize the same most domain-adaptable layer; (3) our method with expanded layers, where we visualize the newly created expanded layer (copied from the most domain-adaptable layer).

Figure 6 presents the results from three different model configurations, providing compelling evidence for the advantages of our proposed approach.

Figure 6 a) shows the activation distribution in layer 27 of the original Qwen3-1.7B-Base model. The substantial overlap between general and medical text distributions indicates strong parameter coupling, which is an expected consequence of mixed-domain pretraining. This coupling makes it challenging to achieve clean separation of domain-specific functionalities through conventional fine-tuning approaches. However, the divergence between the peak values in the general domain and the medical domain also indicates the potential for decoupling.

This coupling phenomenon persisted in our ablation studies with only the decoupling method in Figure 6 b). Despite our attempts to decouple the medical and general modules when training, the model's activation distributions remained largely entangled (the graph still shows substantial overlap), failing to achieve distinct separation between domains. This observation further supports our argument that pre-existing parameter coupling from mixed-domain pretraining creates inherent challenges for direct decoupling approaches.

In contrast, Figure 6 c) demonstrates the activation distribution in layer 31 of our extended model, where we first expanded the model by copying parameters from layer 27 and then applied decoupling training. The clear separation between general and medical text distributions suggests successful parameter decoupling. This superior decoupling effect can be attributed to our "blank slate" approach: the extended layers, while initialized with copied parameters, provide a fresh parameter space that hasn't been constrained by mixed-domain pretraining. During decoupling training, these extended layers can adapt more freely to domain-specific patterns through gradient descent and importance-based learning rate adjustments.

To validate our hypothesis, we also examine the effect of applying in Qwen3-4B-Base (Figure 7), Qwen3-8B-Base (Figure 8), Llama3-8B (Figure 9). The results indicate limited separation between domains, which supports our argument that the entangled parameters from mixed-domain pretraining are challenging to decouple through training alone.

These findings demonstrate that our layer extension strategy provides a more effective pathway for parameter decoupling compared to direct decoupling training. By creating a new parameter space through layer extension, we avoid the constraints of pre-existing parameter coupling, allowing for cleaner separation of domain-specific functionalities during subsequent training. This approach offers a promising direction for developing more modular and domain-adaptable language models.

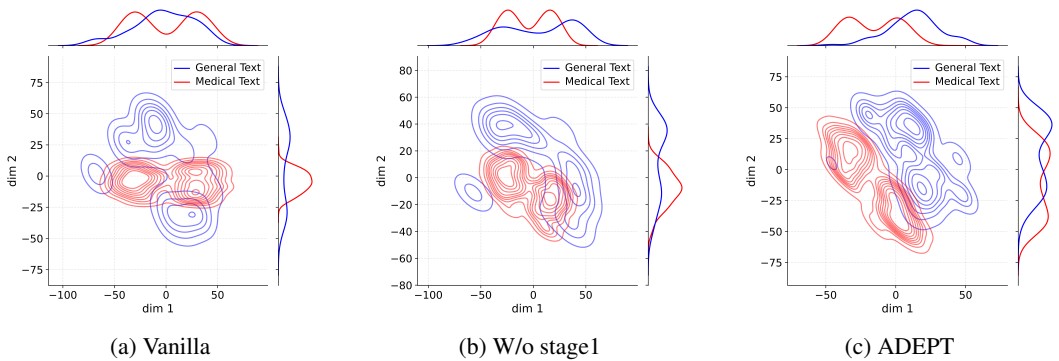

Figure 7: Visualization of activation distributions for Qwen3-4B-Base model configurations showing the effectiveness of our layer extension strategy for parameter decoupling. We expand the layer 28, 30, 31, 35 of Qwen3-4B-Base.

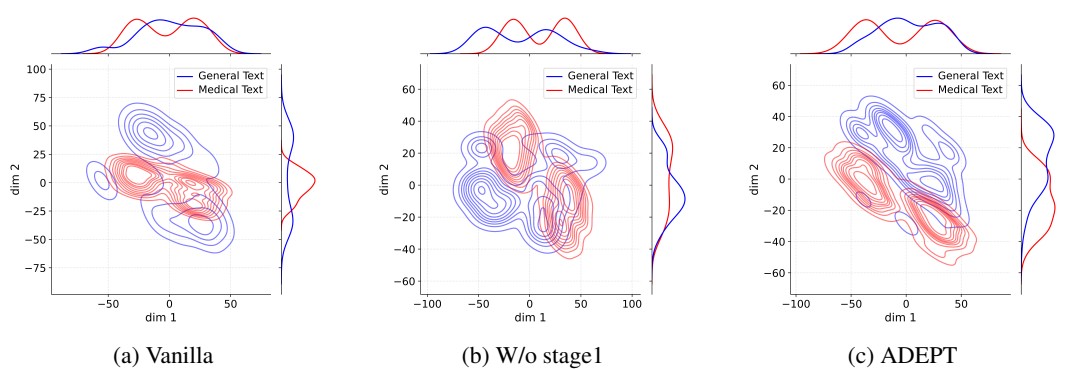

Figure 8: Kernel Density Estimation of activations for Qwen3-8B-Base, showing that our layer extension strategy enables clear parameter decoupling. We expand layers 26, 28, 29, and 30.

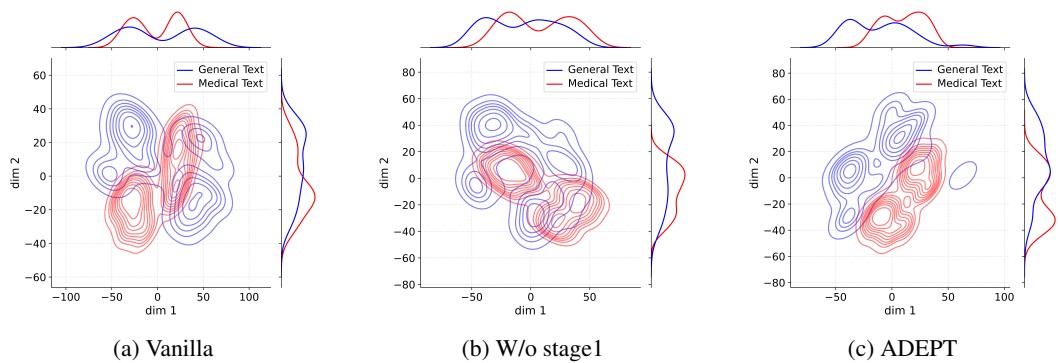

Figure 9: Kernel Density Estimation of activations for Llama3-8B, showing that our layer extension strategy enables clear parameter decoupling. We expand layers 22, 23, 24, and 28.

# D  DETAILED IMPORTANCE DISTRIBUTION

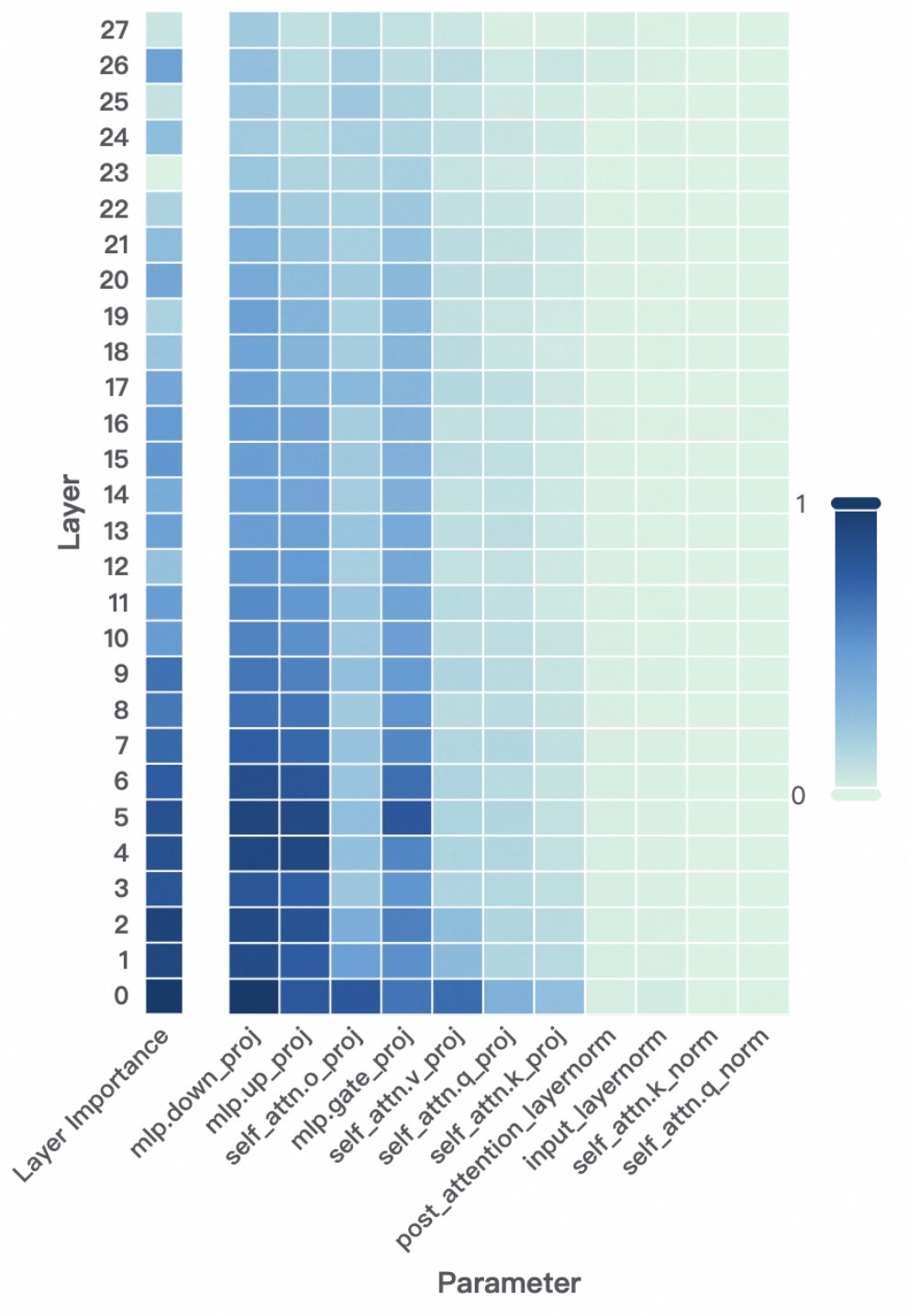

Figure 10: Layer-wise and parameter-wise importance distribution of Qwen3-1.7B-Base model

To investigate which layers should be expanded, we conduct a comprehensive importance analysis at both the layer and parameter levels. Specifically, we compute the importance scores for each layer and parameter across multiple models, and visualize their detailed distributions (see Figure 10, Figure 11, Figure 12, and Figure 13). Our analysis yields the following key observations:

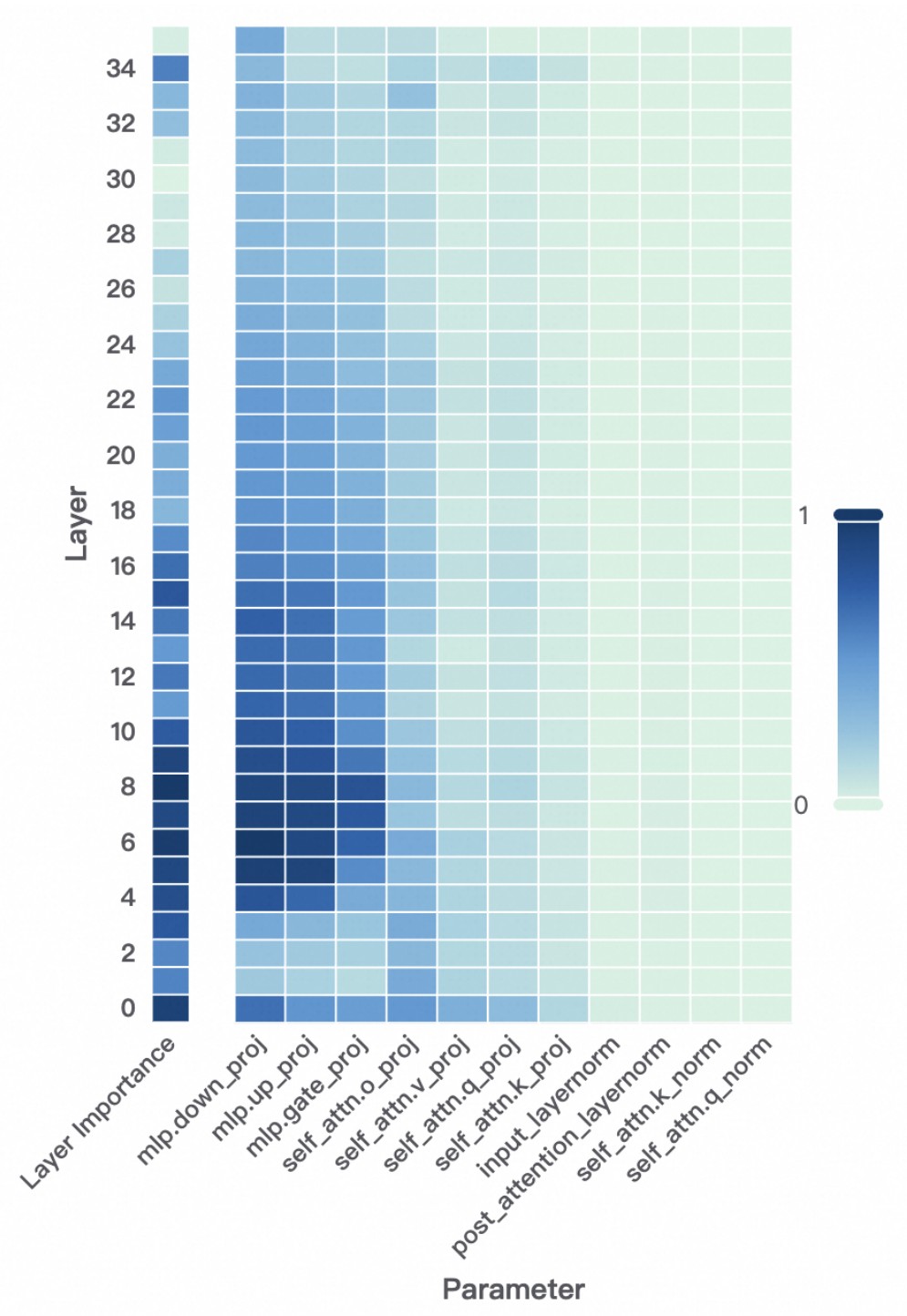

Figure 11: Layer-wise and parameter-wise importance distribution of the Qwen3-4B-Base model

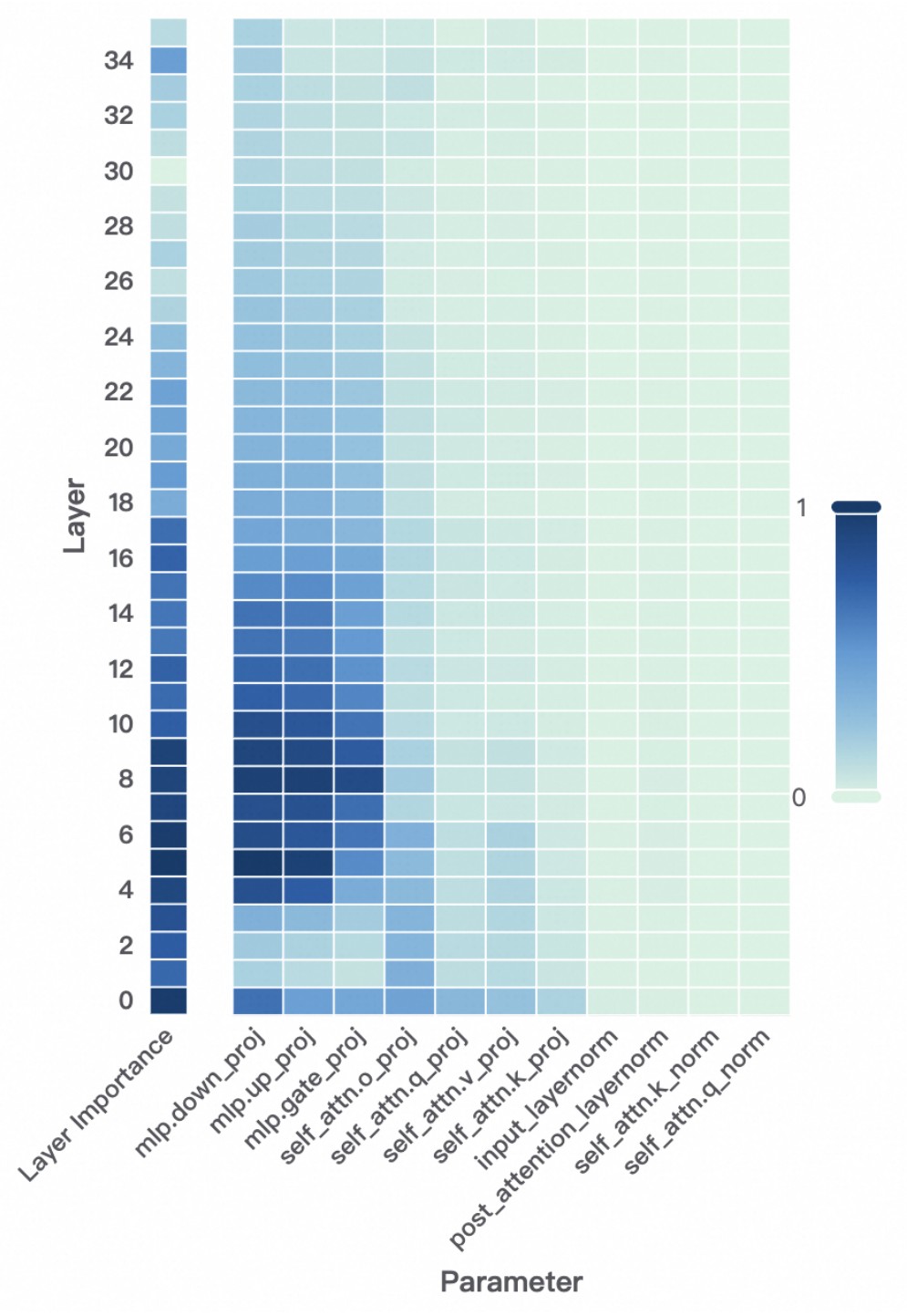

Figure 12: Layer-wise and parameter-wise importance distribution of the Qwen3-8B-Base model

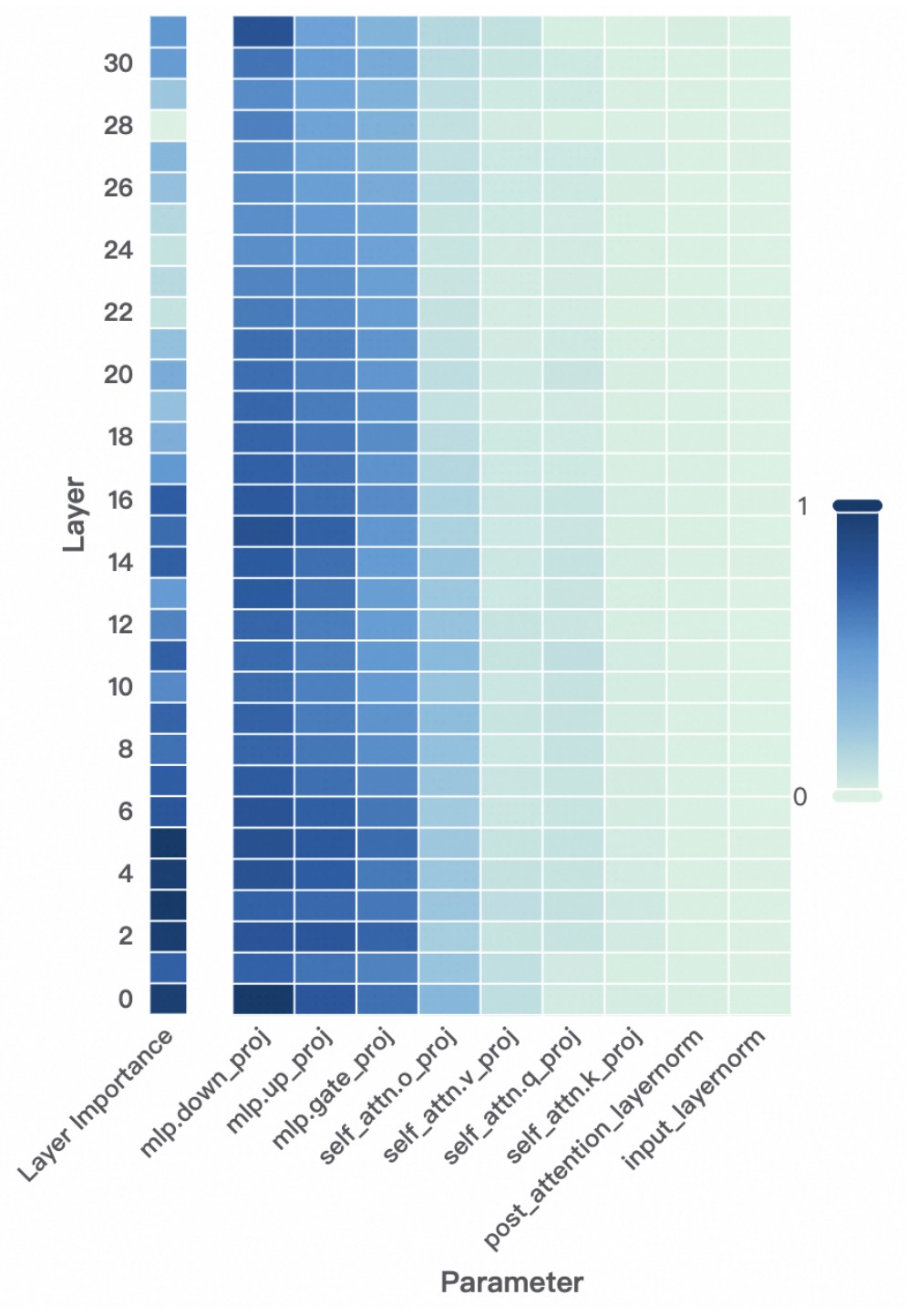

Figure 13: Layer-wise and parameter-wise importance distribution of the Llama3-8B model

**1. Layer and parameter importance alignment.** Overall, the distributions of layer-wise importance and parameter-wise importance are highly aligned across all four models. This alignment is expected, as both metrics are fundamentally computed under the same principle—estimating the impact of masking out (setting to zero) a given layer or parameter on model performance. Since parameter importance essentially decomposes the contribution at the layer level, this consistency reflects the intrinsic, nested relationship between the two. It also indicates that layer-level and parameter-level interventions affect the model's predictive capability in a coherent manner.

**2. High importance in lower layers and the penultimate layer exception.** A notable pattern across all models is that the most important layers tend to be concentrated in the lower (early to middle) layers of the network, with importance values generally decreasing towards higher layers. This pattern suggests that the early layers play a critical role in the overall function of the model.

One plausible explanation, is that lower layers are responsible for capturing general syntactic properties and foundational compositionality (Clark et al., 2019; Hewitt & Manning, 2019), such as basic grammar and phrase structure. In contrast, deeper layers are typically responsible for integrating more task- or context-specific semantic information. This division of labor (earlier layers for generic linguistic structure, deeper layers for task semantics) naturally results in higher sensitivity to interventions at the bottom layers. This also provides a theoretical basis for layer expansion in deep layers.

An interesting exception observed in all models is that the penultimate layer does not follow this general trend: its importance appears elevated relative to immediately adjacent layers. This may stem from the model's need to consolidate high-level semantic features just before producing the output prediction. The penultimate layer may act as a "bottleneck" for aggregating information necessary for the final decision or token generation—potentially as a final representation refinement step. Similar phenomena have been observed in works such as *Intrinsic Dimensionality Explains the Effectiveness of Language Model Pruning* (Aghajanyan et al., 2021), which highlight the special role of upper- and penultimate layers in output formation.

**3. Intra- and inter-family patterns: Qwen vs. Llama models.**

*Qwen family:* Across the Qwen models (Qwen3-1.7B, 4B, 8B), the overall trends are similar:

- Importance is strongly concentrated in the lower and middle layers, particularly within the first 10 layers, regardless of total model depth.
- Among parameters, `mlp.down_proj` and `mlp.up_proj` typically dominate in the most important layers, suggesting that feed-forward (MLP) components contribute substantially to the information processing in the Qwen series.
- With increasing model size (from 1.7B to 8B), the importance distribution appears to spread out slightly, showing less sharpness at the very bottom—possibly reflecting increased capacity and redundancy in larger networks.

*Cross-family:* Comparing Qwen models to Llama3-8B, we observe both notable similarities and differences:

- Both model families consistently exhibit high importance in MLP-related parameters (`mlp.down_proj`, `mlp.up_proj`, and `mlp.gate_proj`), especially in the most important layers. This underscores the universal role of the feed-forward network in transforming and integrating information beyond the capabilities of self-attention alone.
- Llama3-8B shows a broader distribution of importance across layers, with non-negligible values extending further into the middle and upper layers, suggesting a more distributed processing pipeline. In contrast, Qwen models tend to concentrate importance more in the lower layers.
- The dominance of MLP components in Llama3-8B is somewhat less pronounced than in Qwen, with parameter importance appearing more diffuse overall. These inter-family differences may be attributable to variations in architecture (such as normalization, attention mechanisms, or feed-forward design), pre-training data, or other modeling choices, leading to distinct strategies of information flow and representation across the network depth.

# E  LAYER-WISE IMPORTANCE ESTIMATION METHODS COMPARISON

To investigate which layers contribute most to model performance, we employed four different strategies to compute layer-wise importance:

1. **Cumulate importance of parameters:** For each parameter $p$ in a layer, we compute the product $p\frac{\partial \mathcal{L}}{\partial p}$, and sum across all parameters in the layer:

$$I_{\text{layer}} = \sum_{p \in \text{layer}} p\frac{\partial \mathcal{L}}{\partial p} \tag{11}$$

2. **Module-wise rank aggregation:** For each module (e.g., attention, MLP, normalization), we calculate the importance score, rank layers by their score within each module, and aggregate rankings to obtain a total rank for each layer.

3. **Masking out:** For each layer, we mask out its parameters (i.e., set to zero) and evaluate the change in loss:

$$I_{\text{layer}} = \mathcal{L}(\text{model with layer } l \text{ masked}) - \mathcal{L}(\text{original model}) \tag{12}$$

4. **Fisher information:** For each parameter $p$ in a layer, using the Fisher information approximation

$$F(p) = \mathbb{E}\left[\left(\frac{\partial \log p(y|x)}{\partial p}\right)^2\right] \tag{13}$$

   Layer-level Fisher importance is obtained by summing over all parameters in the layer.

To further understand the significance and robustness of these metrics, we conducted a preliminary experiment on the Qwen3-1.7B-Base in the medical domain with dev subset of MMLU, CMMLU to detect the importance of layers, focusing on how different gradient computation strategies affect downstream performance.

Table 11: Performance of different expansion methods on medical-domain tasks (**best result in each column is bolded**). The numbers in parentheses after each method in the table indicate which layers were expanded. The Qwen3-1.7B-Base model has a total of 28 layers, indexed from 0 to 27.

| Methods Name | mmlu | cmmlu | medqa | cmb | mmcu |
|---|---|---|---|---|---|
| Qwen3-1.7B-Base | 62.57 | 66.86 | 48.39 | 63.67 | 69.17 |
| Uniformly Expansion (6,13,20,27) | 59.06 | 64.98 | 48.78 | 64.25 | 70.10 |
| Uniformly Expansion for first 16 layers (3,7,11,15) | 59.60 | 64.91 | 48.78 | 64.07 | 69.80 |
| Uniformly Expansion for last 16 layers (15,19,23,27) | 61.60 | 66.15 | 49.32 | **65.55** | 71.09 |
| Importance Cumulation (23,24,25,27) | 62.63 | 66.81 | 50.19 | 63.85 | 69.48 |
| Rank Aggregation (22,24,25,27) | 62.72 | 66.86 | 50.57 | 63.97 | 69.78 |
| Masking Out (22,23,25,27) | **62.80** | **66.89** | **50.75** | 65.43 | **71.98** |
| Fisher (23,24,25,26) | 61.84 | 66.43 | 49.15 | 64.13 | 68.82 |

Table 11 compares the effect of different layer selection methods for expansion on a variety of medical-domain tasks using Qwen3-1.7B-Base. Several key observations can be made:

**1. Similarity of selected layers across methods.** All importance calculation methods lead to the selection of similar layers for expansion. For instance, the layers chosen by methods such as Importance Cumulation (23,24,25,27), Rank Aggregation (22,24,25,27), Masking Out (22,23,25,27), and Fisher (23,24,25,26) significantly overlap, especially in the last 6 layers of the model (layers 22 and above). This convergence strongly validates our previous observations that general capability-critical layers tend to be concentrated in the latter half of the model in Appendix D.

In addition, the results show that uniform expansion into the last 16 layers (Uniformly Expansion for last 16 layers (15,19,23,27)) consistently outperforms expansion into the first 16 layers (Uniformly

Expansion for first 16 layers (3,7,11,15)) or uniformly across all layers, further supporting the result in Appendix D.

**2. Robustness of expansion results across methods.** Despite minor variability in the specific layers chosen by each method, the final performance of all importance-based expansion approaches is consistently better than both the vanilla baseline and uniform expansion. For example, on the MedQA dataset, all methods using calculated importance exceed the baseline score (e.g., Masking Out achieves 50.75 vs. baseline 48.39), and on MMLU-med, Rank Aggregation achieves 67.95 versus the baseline 66.49. Crucially, the differences in scores among Masking Out, Rank Aggregation, Importance Cumulatation, and Fisher are relatively small for most tasks (typically less than 2 points), indicating that the overall framework is robust to the choice of importance calculation technique. Since our principal contribution is the training paradigm rather than the specific importance metric, for subsequent experiments, we employ the masking out approach, which demonstrated the strongest effect in preliminary experiment.

## F THEORETICAL ANALYSIS

Our theoretical analysis relies on several simplifying assumptions as outlined below. We discuss the rationality and limitations of each assumption:

(A1) **Linearized Model Structure:** We model the transformer as a stack of $L$ independent residual blocks, effectively ignoring cross-layer coupling effects such as those arising from pre-norm and residual connections.

*Justification:* In our layer expansion scheme, **the newly added layers are always separated by at least one original frozen layer and never arranged in a cascading manner**. This design substantially weakens direct coupling between newly expanded layers, which, in turn, reduces the degree of inter-layer interaction and nonlinearity affecting our analysis. And this abstraction is commonly used in theoretical studies (e.g., NTK analysis or pruning literature) to make layerwise analysis tractable.

(A2) **Loss Function Smoothness:** We assume the loss function $\ell(\cdot, \cdot)$ is $\beta$-smooth and $L_\infty$-Lipschitz with respect to predictions.

*Justification:* Standard loss functions such as cross-entropy (with stability improvement) and mean squared error are widely established to satisfy these properties. These conditions allow us to relate small output perturbations to controlled changes in loss, facilitating theoretical bounds.

(A3) **Training Dynamics:** Our analysis assumes training is performed with a first-order SGD-like optimizer, disregarding effects from Adam or other adaptive methods.

*Justification:* First-order SGD provides well-understood theoretical properties and is commonly used in theoretical deep learning research. While Adam introduces adaptive scaling that can affect convergence, many results (e.g., generalization gap bounds) transfer qualitatively between SGD and Adam in practice.

(A4) **NTK Regime and Sensitivity:** Our analysis of layer sensitivity relies on the NTK (Neural Tangent Kernel) approximation (Jacot et al., 2018), which essentially assumes the model behaves locally linearly around its current parameters. Moreover, we should consider the model training process to be relatively stable, with no anomalous occurrences such as gradient explosion.

*Justification:* This assumption is particularly well-motivated in our setting for two reasons. First, our adaptation protocol only updates a small number of newly introduced parameters while keeping the vast majority of the pre-trained weights frozen and decouples parameters to maximize the retention of general capabilities. This ensures that the parameter changes remain minimal, keeping the network within the local linear (NTK) regime throughout adaptation. Second, unlike random initialization, our starting point is a well-trained model on a large general-domain corpus, which already provides robust and meaningful representations. Perturbations induced by finetuning are thus intrinsically local in the function space and less likely to induce sudden or nonlinear model behavior, further enhancing the validity of the NTK approximation.

Overall, these assumptions enable us to derive interpretable upper bounds and provide actionable layer selection criteria, but should be considered as idealizations. The correspondence between these theoretical insights and practical behavior is also validated in our empirical experiments.

### F.1 OPTIMALITY OF LEAST-IMPORTANT BLOCK EXPANSION FOR PRESERVING GENERAL CAPABILITIES

**Notation:** Let $M_0$ denote the original base model, and $M_S^{(T)}$ denote the model after $T$ steps of adaptation, wherein only the set $S$ of $k$ layers are unfrozen and updated, and $\ell(\cdot, y)$ is the loss function (e.g., cross-entropy) which is $L$-Lipschitz and $\beta$-smooth in its first argument. $\Delta^{(l)}$ represents the importance score of layer $l$ as defined below.

**Layer Importance Score:**

$$\Delta^{(l)} := \mathbb{E}_{x \sim D_{gen}} \left[ \ell(M_0^{(-l)}(x), y(x)) - \ell(M_0(x), y(x)) \right]$$

where $M_0^{(-l)}$ is $M_0$ with the $l$-th layer masked out.

**Theorem F.1** (Upper Bound on Generalization Gap by Layer Importance). *Let $S \subseteq [L]$ be the set of layers selected for expansion/adaptation, and $G(S)$ denote the source-domain generalization gap after adaptation, i.e.,*

$$G(S) := \mathbb{E}_{x \sim D_{gen}} \left[ \ell(M_S^{(T)}(x), y(x)) - \ell(M_0(x), y(x)) \right].$$

*Under function-preserving initialization, limited adaptation steps, and $L$-Lipschitz and $\beta$-smooth loss, the following upper bound holds:*

$$G(S) \leq C \sum_{l \in S} \Delta^{(l)} + O\left(k(\overline{\Delta W})^2\right)$$

*where $C$ is a constant depending on the learning rate, steps, loss smoothness, and initialization, and $\overline{\Delta W}$ is the maximal per-layer parameter change over adaptation.*

*Proof.* **Step 1: Output Deviation Linearization.** By function-preserving initialization, $M_S^{(0)}(x) = M_0(x)$. After adaptation, since only layers in $S$ are modified and changes are small (Assumption A4), the output difference admits a first-order Taylor expansion:

$$M_S^{(T)}(x) - M_0(x) \approx \sum_{l \in S} J_l(x)\, \Delta W_l$$

where $J_l(x) = \left. \frac{\partial M}{\partial W_l} \right|_{W=W_0}$ and $\Delta W_l = W_l^{(T)} - W_l^{(0)}$.

**Step 2: Lipschitz Property Application.** By $L$-Lipschitzness of $\ell(\cdot, y)$ in its first argument,

$$|\ell(M_S^{(T)}(x), y) - \ell(M_0(x), y)| \leq L \left\| M_S^{(T)}(x) - M_0(x) \right\|_2.$$

Taking the expectation over $x \sim D_{gen}$,

$$G(S) \leq L\, \mathbb{E}_x \left[ \| M_S^{(T)}(x) - M_0(x) \|_2 \right].$$

**Step 3: Breaking by Layer via Triangle Inequality.** According to Assumption A1 and using the triangle inequality,

$$\| M_S^{(T)}(x) - M_0(x) \|_2 \leq \sum_{l \in S} \| J_l(x)\, \Delta W_l \|_2,$$

thus,

$$G(S) \leq L \sum_{l \in S} \mathbb{E}_x \left[ \| J_l(x)\, \Delta W_l \|_2 \right].$$

**Step 4: Relating to Layer Importance Score.** Recall the definition:

$$\Delta^{(l)} = \mathbb{E}_x \left[ \ell(M_0^{(-l)}(x), y) - \ell(M_0(x), y) \right].$$

By Taylor expansion and Lipschitz continuity,

$$|\ell(M_0^{(-l)}(x), y) - \ell(M_0(x), y)| \approx L \|J_l(x) W_l^{(0)}\|_2,$$

so for small modifications,

$$\mathbb{E}_x[\|J_l(x)\,\Delta W_l\|_2] \leq \frac{\|\Delta W_l\|_2}{\|W_l^{(0)}\|_2} \Delta^{(l)} + O(\|\Delta W_l\|_2^2).$$

Assume $\|\Delta W_l\|_2 \leq \overline{\Delta W}$ for all $l \in S$ and $\|W_l^{(0)}\|_2$ are similar or lower-bounded by $w_0 > 0$, so

$$G(S) \leq L \frac{\overline{\Delta W}}{w_0} \sum_{l \in S} \Delta^{(l)} + O\big(k(\overline{\Delta W})^2\big).$$

**Step 5: Optimization Control.** In standard SGD (Assumption A3), $\overline{\Delta W}$ is controlled by learning rate $\eta$, steps $T$, batch size $N$, and bounded gradients:

$$\overline{\Delta W} \lesssim \frac{\eta T}{N} \max_{t,i} \|\nabla_{W_l}\ell(M_0(x_i), y_i)\|_2.$$

Thus, all learning and initialization constants can be absorbed into a scalar constant $C$ (Assumption A3 and A4).

**Step 6: Conclusion.** Thus,

$$G(S) \leq C \sum_{l \in S} \Delta^{(l)} + O\left(k(\overline{\Delta W})^2\right).$$

which completes the proof. $\qquad\qquad\square$

**Due to the use of residual connections, the original block and the expanded block can be viewed as a single aggregated unit. Importantly, before training, the addition of the new block does not alter the model's output, and thus the overall importance of the aggregated block remains exactly the same as that of the original block (i.e., $\Delta^{(l)}$).** As a result, when we train the parameters of the new block, it is effectively equivalent to adapting the aggregated block as a whole, whose importance is still characterized by the original importance score $\Delta^{(l)}$. This justifies why the potential impact of training the expanded layer is governed by the original layer's importance.

The tightness of the derived upper bound hinges on both the local linearity of the expansion regime and the control over parameter updates during adaptation. In cases where the expansion layers are initialized to be function-preserving and the adaptation is performed with sufficiently small learning rates and moderate step sizes, the Taylor and Lipschitz approximations used in the proof become increasingly sharp. Thus, the upper bound is not only theoretically attainable, but also approaches the realistic generalization gap observed in practice under these conditions. This means that minimizing the sum $\sum_{l \in S} \Delta^{(l)}$ when selecting layers for expansion is not merely a mathematical convenience—it is a principled, actionable strategy for controlling catastrophic forgetting and generalization degradation. As a consequence, our criterion provides practical guidance: by limiting updates to those layers with the lowest importance scores, practitioners can reliably minimize negative transfer from domain adaptation, especially when adapting large pre-trained models with limited new capacity.

### F.2 Optimality of Importance-Based Learning Rate Adjustment for Modules

We provide a rigorous analysis of learning rate reallocation in Stage 2. Specifically, let the importance of each parameter $\theta_j$ in the general domain be defined as

$$I_{\theta_j} = \left| \frac{\partial L_{\text{gen}}}{\partial \theta_j} \right|$$

where $L_{\text{gen}}$ denotes the general-domain loss and $I_{\theta_j}$ quantifies the sensitivity of the overall performance with respect to $\theta_j$. Under the constraint of a fixed average learning rate, our strategy assigns lower learning rates to parameters with high general-domain importance, and higher learning rates to those deemed less important. This importance-weighted reallocation is provably optimal for minimizing the upper bound of catastrophic forgetting in the general domain, subject to the constant average learning rate constraint. Furthermore, we formulate and analytically solve the underlying constrained optimization problem to ensure that our reallocation approach achieves relative optimality in practice.

**Setup and Notation** Let $D_{gen}$ be the general domain distribution with loss $L_{gen}(\theta)$. With $\theta^*$ as the original pre-trained parameters, we define parameter importance $I_j \triangleq \theta_j \frac{\partial L_{gen}}{\partial \theta_j}|_{\theta^*}$ and unit importance:

$$I_{U_i} \triangleq \frac{1}{|U_i|} \sum_{j \in U_i} I_j \in [0, 1] \tag{14}$$

under learning rate budget constraint:

$$\sum_i \frac{|U_i|}{|\Theta_\sim|} lr_{U_i} = lr_{base} \tag{15}$$

### F.2.1 Upper Bound on Forgetting

Define forgetting as:

$$F \triangleq L_{gen}(\theta(T)) - L_{gen}(\theta^*) \tag{16}$$

Assuming $L_{gen}$ is $\beta$-smooth, the first-order Taylor expansion provides:

$$F \le \nabla_\theta L_{gen}(\theta^*)^\top \Delta(T) + \frac{\beta}{2} \|\Delta(T)\|^2 \tag{17}$$

Due to parameter freezing, the gradient $\nabla_\theta L_{gen}(\theta^*)$ is only non-zero for expanded parameters:

$$\nabla_\theta L_{gen}(\theta^*) = \sum_i \sum_{j \in U_i} I_j e_j \tag{18}$$

where $I_j = \frac{\partial L_{gen}}{\partial \theta_j}$, $e_j$ are basis vectors.

Assuming gradient descent with per-group step size $\eta_{U_i}$ and $T$ steps, for each parameter $j \in U_i$ (Assumption A4):

$$\Delta_j(T) \approx -T\eta_{U_i} \frac{\partial L_{med}}{\partial \theta_j} \tag{19}$$

Substitute into the smoothness bound:

$$F \le \sum_i \sum_{j \in U_i} I_j \Delta_j(T) + \frac{\beta}{2} \sum_i \sum_{j \in U_i} (\Delta_j(T))^2 \tag{20}$$

$$\le \sum_i |U_i| \cdot |I_{U_i}| \cdot (T\eta_{U_i}G) + \frac{\beta}{2}T^2 \sum_i |U_i|\eta_{U_i}^2 G^2 \tag{21}$$

where $G := \max_j |\frac{\partial L_{med}}{\partial \theta_j}|$ upper-bounds the adaptation gradients.

The derived upper bound encompasses all possible learning rate allocations and ensures conservative control over catastrophic forgetting. Note that if group gradients $G$ or importance scores $I_{U_i}$ are heterogeneous, a more refined bound can be obtained by analyzing variance rather than worst-case values.

### F.3 OPTIMAL IMPORTANCE-DRIVEN LEARNING RATE REALLOCATION

**Problem Statement:**
We aim to allocate learning rates $\eta_{U_i}$ for each parameter group $U_i$ so as to minimize the upper bound on forgetting:

$$F \leq a \sum_i w_i I_i \eta_{U_i} + b \sum_i w_i \eta_{U_i}^2$$

where $w_i = |U_i|$ is the number of parameters in group $U_i$, $I_i = |I_{U_i}|$ indicates the average importance of parameters in $U_i$, $a, b > 0$ are constants determined by training steps, gradient norms, and the smoothness constant ($\beta$ (Assumption A2)). The constraint is that the average learning rate remains fixed:

$$\sum_i w_i \eta_{U_i} = W \eta_{avg}$$

where $W = \sum_i w_i$ is the total number of trainable parameters.

**Lagrangian Formulation:**
Introduce a Lagrange multiplier $\lambda$ and write the Lagrangian:

$$\mathcal{L}(\{\eta_{U_i}\}, \lambda) = a \sum_i w_i I_i \eta_{U_i} + b \sum_i w_i \eta_{U_i}^2 + \lambda \left( \sum_i w_i \eta_{U_i} - W \eta_{avg} \right)$$

**Optimality Condition:**
Taking derivatives and setting to zero, we obtain for each $j$:

$$\frac{\partial \mathcal{L}}{\partial \eta_{U_j}} = a w_j I_j + 2 b w_j \eta_{U_j} + \lambda w_j = 0$$

$$\implies \eta_{U_j}^* = -\frac{a}{2b} I_j - \frac{\lambda}{2b}$$

Including the constraint:

$$\sum_j w_j \eta_{U_j}^* = W \eta_{avg}$$

Plugging in the expression for $\eta_{U_j}^*$ gives:

$$-\frac{a}{2b} \sum_j w_j I_j - \frac{\lambda}{2b} W = W \eta_{avg}$$

Solving for $\lambda$:

$$\lambda = -2b \eta_{avg} - \frac{a}{W} \sum_j w_j I_j$$

So the optimal learning rate for group $U_j$ is:

$$\eta_{U_j}^* = \eta_{avg} - \frac{a}{2b} \left( I_j - \frac{1}{W} \sum_{j'} w_{j'} I_{j'} \right) \tag{22}$$

**Interpretation and Guidance:**
When the theoretical upper bound is tight—which is often the case in well-controlled, locally linear training regimes—this result has direct practical utility. Notably, the optimal learning rate allocation $\eta_{U_j}^*$ is an affine (linear) function of the group importance $I_j$. Our method, which assigns $\text{lr}_U = 2 \cdot (1 - I_{\text{unit}}) \cdot \text{lr}_{\text{base}}$, can be viewed as a simplified implementation of the derived optimal form. By decreasing the learning rate for groups with high general-domain importance and increasing it for those with low importance, this strategy effectively minimizes the risk of catastrophic forgetting while respecting the global learning rate constraint. Thus, our approach provides actionable guidance for tailoring learning rates based on parameter importance in continual learning and domain adaptation.

## G  EXPERIMENT ABOUT THE NUMBER OF EXPANDED LAYERS

In Stage 1, determining the optimal number of expanded layers emerges as a crucial hyperparameter. To investigate this, we conducted systematic experiments across various model scales in the medical domain by expanding different numbers of layers. These comprehensive experiments aim to provide empirical insights into selecting the most effective layer expansion strategy, offering valuable guidance for future research in this direction.

Table 12: Comparative Performance of Different Layer Expansion Strategies across Model Scales and Medical Tasks. **Bold** indicates the best-performing setup for each task; underline shows the second-best. This highlights optimal and near-optimal choices for each scenario.

| Model | MMLU | CMMLU | MedQA | MMCU-Medical | CMB |
|---|---|---|---|---|---|
| *Qwen3-1.7B* | | | | | |
| Vanilla | 62.57 | 66.86 | 48.39 | 69.17 | 63.67 |
| 1-layer | 62.31 | 66.23 | 48.08 | 69.95 | 61.40 |
| 2-layer | 62.48 | **66.91** | 48.63 | 70.78 | 62.89 |
| 4-layer | **62.80** | 66.89 | **50.75** | 71.98 | **65.43** |
| 8-layer | 61.84 | 66.02 | 49.57 | **72.41** | 65.00 |
| 16-layer | 60.96 | 64.65 | 48.86 | 70.13 | 64.88 |
| *Qwen3-4B* | | | | | |
| Vanilla | **73.19** | 77.92 | 62.77 | 82.44 | 78.92 |
| 1-layer | 72.98 | 77.69 | 63.39 | 82.83 | 78.21 |
| 2-layer | 73.10 | 77.84 | 63.08 | 82.80 | 78.48 |
| 4-layer | 72.95 | **78.77** | **64.49** | **84.58** | **79.87** |
| 8-layer | 73.06 | 77.65 | 65.02 | 84.22 | 78.81 |
| 16-layer | 72.06 | 77.11 | 62.61 | 82.09 | 78.61 |
| *Qwen3-8B* | | | | | |
| Vanilla | 76.94 | 82.09 | 66.30 | 86.45 | 81.67 |
| 1-layer | 76.84 | 82.06 | 67.87 | 86.95 | 81.50 |
| 2-layer | 76.70 | 82.10 | 67.93 | 87.99 | 82.90 |
| 4-layer | 76.77 | 82.11 | **69.24** | **89.84** | **85.80** |
| 8-layer | 76.77 | 82.15 | 68.34 | 88.02 | 84.85 |
| 16-layer | **77.12** | **82.28** | 68.56 | 87.76 | 84.32 |
| *LLaMA3-8B* | | | | | |
| Vanilla | 65.33 | 50.83 | 58.91 | 46.29 | 35.61 |
| 1-layer | 65.29 | 51.12 | 58.97 | 50.83 | 40.45 |
| 2-layer | 65.61 | 50.98 | 59.56 | 55.92 | 47.83 |
| 4-layer | 65.25 | 51.73 | 60.82 | 63.17 | 54.65 |
| 8-layer | 65.17 | 51.92 | 61.17 | 67.03 | 61.78 |
| 16-layer | **65.12** | **52.45** | **61.92** | **70.86** | **65.31** |

For general language tasks such as MMLU and CMMLU, all models largely preserve their baseline performance regardless of the number of expanded layers. This indicates that layer expansion does not compromise the models' general language capabilities and robustness.

However, for domain-specific medical tasks (MedQA, MMCU-Medical, and CMB), the impact of layer expansion is more pronounced. Across all Qwen model variants (1.7B, 4B, and 8B), expanding 4 layers consistently yields optimal performance, as shown by the bolded results in Table 12. Specif-

ically, the Qwen3-1.7B, 4B, and 8B models improve on MMCU-Medical by up to 2.8%, 2.1%, and 3.4%, respectively, when increasing from baseline to 4-layer expansion. Notably, expanding beyond 4 layers (e.g., to 8 or 16 layers) does not systematically improve performance—and in several cases, results in diminishing or even degraded accuracy. This suggests that moderate layer expansion (4 layers) achieves a balance between performance gain and model stability, while excessive expansion may introduce optimization difficulties, overfitting, or disrupt the pre-trained knowledge representations, leading to suboptimal outcomes.

In contrast, the LLaMA3-8B model displays a unique trend: performance improvements are continuous as more layers are expanded, with the best results observed at expanding 16 layers. The gains are considerable for tasks like MMCU-Medical and CMB, where scores rise dramatically from 46.29% and 35.61% in the vanilla model to 70.86% and 65.31% with 16 expanded layers. This behavior contrasts with the Qwen models and is likely due to LLaMA's more limited Chinese capability in its original configuration. The need for extensive architectural expansion reflects the necessity to build new, specialized representations to compensate for baseline deficiencies when addressing Chinese-centric tasks. Therefore, while moderate layer expansion is optimal for models pre-trained on Chinese data (Qwen), more substantial expansion may be required for models less adapted to the target language or domain (LLaMA).

Overall, these results indicate that expanding more layers does not guarantee better performance. For well-aligned models, excessive expansion may lead to interference with the original knowledge or cause optimization instability. In contrast, for models lacking target domain competence, increased expansion helps establish the missing representations, albeit at the cost of greater computational complexity.

## H  TAKE PRETRAIN DATA AS IMPORTANCE SOURCE

Our previous experiments employed the dev sets of MMLU and CMMLU as benchmark datasets for gradient-based importance estimation. However, such high-quality and carefully curated benchmarks are often scarce, especially in practical industrial scenarios. To investigate the robustness of our ADEPT method under more realistic conditions where benchmark data may not be available, we explore the use of noisier pretraining data for importance estimation.

Table 13: General Competence Detection Pretrain Corpus. #Examples means the number of examples we used.

| Dataset | #Examples | Hugging Face Link |
|---|---|---|
| FineWeb_Edu | 500 | HuggingFaceFW/fineweb-edu |
| FineWeb_Edu_Chinese V2.1 | 500 | HuggingFaceFW/fineweb-edu |

Specifically, we utilize the FineWebEdu and FineWebEdu-Chinese datasets (Data overview and links in Table 13), extracting the top 500 samples with the highest educational scores from the first 10,000 entries in each corpus to serve as our importance estimation set. Compared to curated benchmarks, these datasets are much more accessible in real-world applications. Furthermore, the computational cost for filtering out such high-quality samples is negligible relative to the overall cost of large-scale pretraining.

This experimental setting allows us to rigorously evaluate the robustness of ADEPT when real-world, easily accessible pretraining data replaces ideal benchmark datasets for importance-based layer expansion decisions.

Table 14 summarizes the performance of our ADEPT method when the importance estimation is conducted with either high-quality benchmark data or more easily accessible pretraining data across different model scales. Overall, the results demonstrate that ADEPT not only consistently outperforms the vanilla baseline but also shows remarkable robustness across most scenarios when using pretraining data for importance calculation. In Qwen3 series models, the difference between benchmark-based and pretraining-data-based importance estimation is minimal. In several cases, the latter even slightly surpasses the benchmark version (e.g., Qwen3-1.7B on MMLU and Qwen3-8B on MMLU and CMMLU), validating the practical applicability and flexibility of our approach.

Table 14: Performance comparison of ADEPT with benchmark-based and pretraining-data-based importance estimation across model scales. **Bold** indicates the best performance per column; underline marks the second-best.

| Model | MMLU | CMMLU | MedQA | MMCU-Medical | CMB |
|---|---|---|---|---|---|
| *Qwen3-1.7B* | | | | | |
| Vanilla | 62.57 | 66.86 | 48.39 | 69.17 | 63.67 |
| ADEPT (Benchmark) | 62.80 | **66.89** | **50.75** | **71.98** | **65.43** |
| ADEPT (PT Data) | **62.85** | 66.87 | 49.39 | 70.84 | 63.07 |
| *Qwen3-4B* | | | | | |
| Vanilla | **73.19** | 77.92 | 62.77 | 82.44 | 78.92 |
| ADEPT (Benchmark) | 72.95 | **78.77** | **64.49** | **84.58** | **79.87** |
| ADEPT (PT Data) | 73.14 | 77.96 | 63.94 | 83.34 | 79.62 |
| *Qwen3-8B* | | | | | |
| Vanilla | 76.94 | 82.09 | 66.30 | 86.45 | 81.67 |
| ADEPT (Benchmark) | 76.77 | 82.11 | **69.24** | **89.84** | **85.80** |
| ADEPT (PT Data) | **76.83** | **82.20** | 67.56 | 87.20 | 83.92 |
| *LLaMA3-8B* | | | | | |
| Vanilla | 65.33 | 50.83 | 58.91 | 46.29 | 35.61 |
| ADEPT (Benchmark) | 65.25 | **51.73** | **60.82** | **63.17** | **54.65** |
| ADEPT (PT Data) | **65.21** | 50.27 | 59.13 | 60.29 | 51.32 |

For LLaMA3-8B, ADEPT with pretraining data still yields clear improvements over the vanilla baseline on all tasks, particularly in domain-specific metrics such as MedQA and MMCU-Medical. However, compared to the benchmark-based ADEPT, the pretraining-data variant shows slightly lower performance, with a gap of approximately 1–5% across tasks. This modest drop can be attributed to two main factors: first, the inherent discrepancy between noisier pretraining data and expertly curated benchmarks introduces less precise gradient signals for importance estimation. Second, LLaMA3-8B's weaker baseline in Chinese tasks means its optimization is more sensitive to the quality of importance source, and benefits more from highly targeted benchmark data. Nonetheless, even with this gap, the pretraining-data approach remains highly valid, especially in practical scenarios where access to dedicated benchmarks is limited.

In summary, ADEPT demonstrates strong effectiveness and robustness when layer expansion is guided by pretraining data, making it highly suitable for real-world deployment. The slight performance drop observed in LLaMA3-8B highlights the additional value of benchmark data for models or tasks with substantial baseline limitations, but does not diminish the overall utility of our method in resource-constrained settings.

# I  TOKEN DISTRIBUTION SHIFT

Following the methodology proposed by Lin et al. (2024), we conducted a comprehensive analysis of token distribution shifts between the base and aligned models using the MMLU (Massive Multitask Language Understanding) dataset. The analysis focuses on identifying and quantifying the changes in token prediction patterns that occur during the alignment process.

Our analysis procedure consists of the following steps:

1) For each position in the input text, we use the aligned model with greedy decoding to generate the output token $o_t$.

2) We then examine how this token is ranked in the base model's probability distribution $P_{base}$. This ranking, denoted as $\eta$, serves as our primary metric for categorizing token shifts.

3) Based on the base ranking $\eta$, we classify each token position into three categories:

- Unshifted positions ($\eta = 1$): The token is top-ranked in both base and aligned models
- Marginal positions ($1 < \eta \leq 3$): The token has a relatively high probability in the base model
- Shifted positions ($\eta > 3$): The token is unlikely to be sampled by the base model

4) For shifted tokens, we calculate *Rank Improvement Ratio*: $\frac{\text{base\_rank}}{\text{aligned\_rank}}$

Our analysis of the MMLU dataset revealed significant distribution shifts between the base and continual pretrained models by ADEPT. Figure 14 visualizes the most significantly shifted tokens, where the size of each token is proportional to its rank improvement ratio.

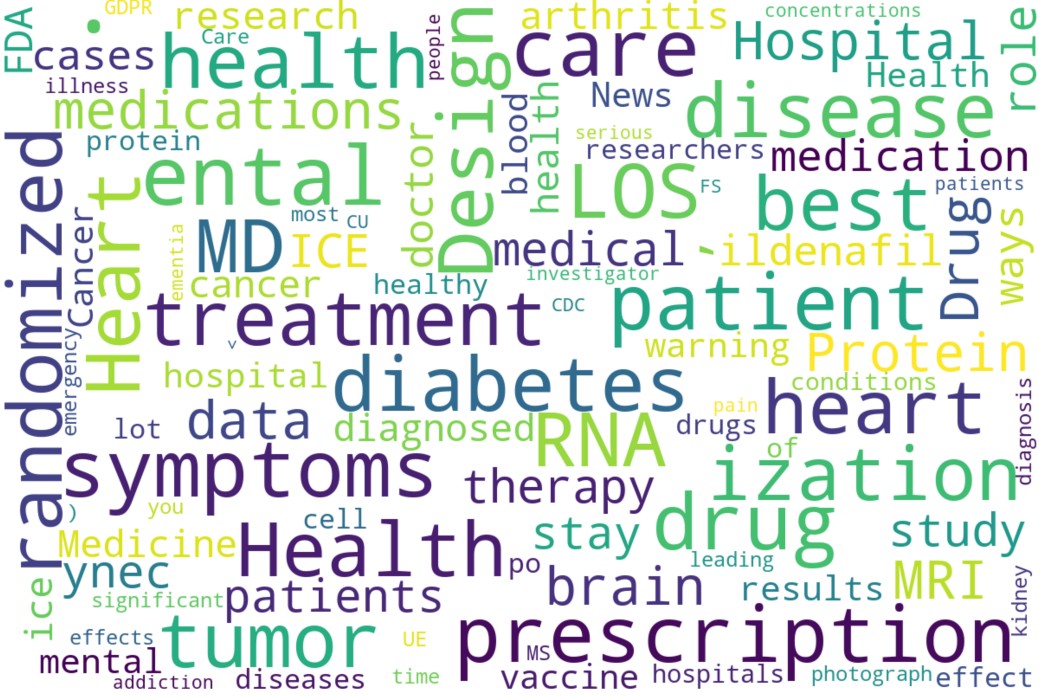

Figure 14: Word cloud visualization of shifted tokens. The size of each token represents its rank improvement ratio ($\frac{\text{base\_rank}}{\text{aligned\_rank}}$), indicating the magnitude of distributional shift during alignment. Larger tokens indicate more significant shifts in the model's prediction patterns.

Our analysis of the MMLU dataset revealed significant and efficient distribution shifts between the base and aligned models. Figure 14 visualizes the most significantly shifted tokens, where the size of each token is proportional to its rank improvement ratio.

The analysis revealed a notably efficient token distribution shift pattern. Specifically, only 2.18% of tokens underwent significant shifts (compared to 5.61% in full pretraining), with 88.78% remaining unshifted and 9.04% showing marginal changes (Totally 645496 tokens analyzed). This represents a more focused and efficient alignment compared to full pretraining scenarios, which typically show higher shift percentages (unshifted: 75.59%, marginal: 18.80%, shifted: 5.61%).

Most remarkably, the shifted tokens demonstrate a clear concentration in medical terminology and medicine-related concepts. Key examples include: "prescription", "diagnosis", "symptoms", "diabetes", "arthritis", "tumor", "MRI", "therapy", "treatment", "hospital", "care", "patients".

This specialized distribution stands in stark contrast to the more general token shifts observed in full pretraining, where top shifted tokens (such as <|im_end|>, "CIF", "Registered", "progression",

"median") show no particular domain focus and more noise. This comparison suggests that ADEPT achieved a more targeted and efficient knowledge injection, specifically enhancing the model's medical domain expertise while maintaining stability in other areas. The lower percentage of shifted tokens (2.18% vs 5.61%) combined with their high domain relevance indicates a more precise and economical alignment process that effectively injects medical knowledge without unnecessary perturbation of the model's general language capabilities.

These findings suggest that domain-specific alignment can be achieved with minimal token distribution disruption while maintaining high effectiveness in knowledge injection. This efficiency in token shifting demonstrates the potential for targeted domain adaptation without the broader distributional changes typically seen in full pretraining scenarios.

Similarly, in mathematical domain alignment (Figure 15), we observed an even more efficient token distribution shift. The analysis shows only 1.24% of tokens underwent significant shifts, with 91.51% remaining unshifted and 7.25% showing marginal changes. This represents an even more concentrated alignment compared to full pretraining (unshifted: 85.45%, marginal: 10.18%, shifted: 4.37%).

The shifted tokens clearly reflect mathematical and scientific terminology, as evidenced by terms such as "theorem", "quantum", "parameters", "physics", and "equation". This highly focused shift pattern, utilizing merely one-third of the token shifts compared to full pretraining (1.24% vs 4.37%), demonstrates the effectiveness of our approach in precisely targeting mathematical knowledge injection while maintaining model stability in other domains.

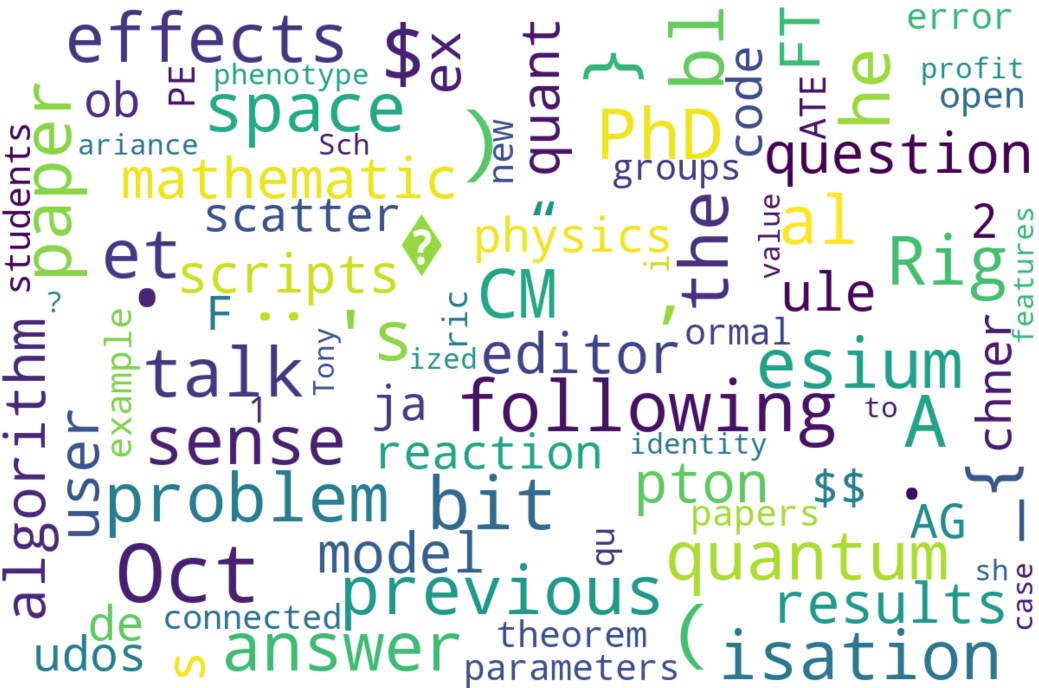

Figure 15: Word cloud visualization of shifted tokens in mathematical domain alignment. The predominance of mathematical and scientific terminology demonstrates the precise targeting of domain-specific knowledge.

## J  LINEAR MERGE OF DOMAIN-SPECIFIC EXTENSIONS: RESULTS AND INSIGHTS

In Table 15, we compare the performance of the Vanilla model and the Merged Model, which was constructed by linearly merging the domain-specific extension layers (with equal weights of 0.5 for medical and mathematical domains) after independent training. Our results show that the merged

Table 15: Performance comparison of Vanilla and Merged Models on multiple benchmarks (Qwen3-1.7B and Qwen3-4B).

|  | MMLU | CMMLU | GSM8K | ARC-E | ARC-C | MedQA | MMCU | CMB |
|---|---|---|---|---|---|---|---|---|
| **Qwen3-1.7B** | | | | | | | | |
| Vanilla | 62.57 | 66.86 | 57.62 | 81.44 | 51.19 | 48.39 | 69.17 | 63.67 |
| Merged Model | 62.70 | 65.83 | 60.80 | 81.06 | 51.94 | 48.39 | 68.61 | 64.83 |
| **Qwen3-4B** | | | | | | | | |
| Vanilla | 73.19 | 77.92 | 69.07 | 85.52 | 59.13 | 62.77 | 82.44 | 78.92 |
| Merged Model | 72.96 | 77.99 | 73.16 | 85.27 | 58.96 | 62.83 | 82.83 | 78.42 |

model does not exhibit any significant collapse, and in some indicators even surpasses the original base model. For example, on the GSM8K benchmark for Qwen3-1.7B, the merged model achieves 60.80%, compared to 57.62% for the vanilla model. This demonstrates the generalization and extensibility of our method, enabling fusion across multiple vertical domains.

Our extension approach ensures that each newly added layer is separated by at least one original frozen layer, rather than being directly adjacent. This design leads to greater stability during model merging. On one hand, if the merged models were purely cascaded, the non-linear transformations introduced could lead to more unpredictable interactions between layers. On the other hand, because each layer operates within a consistent contextual environment provided by surrounding frozen layers during continual pre-training, we believe that this fixed hierarchical structure imposes constraints that make the semantic representations learned by the new layers more aligned in certain dimensions. As a result, the merging process becomes more reliable and beneficial to overall model performance.

It is worth noting that our merging strategy adopts the simplest possible weighted average. The specific merging algorithm is not the focus of this work; we believe that with more scientific weighting schemes, even better results can be obtained. Here, we hope to stimulate further research and provide preliminary insights based on our observations.

## K  USE OF LLM

In the preparation of this article, we utilized large language models (LLM) solely for writing assistance purposes. Specifically, we employed the GPT-4.1-0414 model to polish language expressions, condense sentences, and improve the overall clarity and readability of the text. The model was used exclusively for editing and refining manuscript language and did not participate in any conceptual or technical aspects of this work.

All research ideas, theoretical proof methods, experimental designs, and visualizations were conceived, executed, and finalized by the authors without the involvement of any LLM tools. The development of new concepts, formulation and validation of proofs, experimental setups, analysis of results, and the creation of figures were performed independently by the research team. At no point was the LLM model used to generate, modify, or validate the scientific content, methodology, or results presented in this article.

We emphasize that the role of GPT-4.1-0414 in this research was strictly limited to linguistic enhancement at the writing stage, and that all substantive intellectual and scientific contributions originate solely from the authors.

## L  ALGORITHM

Please see Algorithm 1.

---

**Algorithm 1** ADEPT

---

**Require:** Pretrained LLM $M_0$ with layers $\{\Theta^{(1)}, \ldots, \Theta^{(L)}\}$, domain probing corpus $\mathcal{D}_{\text{probe}}$, continual pretraining corpus $\mathcal{D}_{\text{train}}$, number of layers to expand $k$, base learning rate $\text{lr}_{\text{base}}$, update interval $T_{\text{update}}$

1: # Stage 1: General-Competence Guided Selective Layer Expansion
2: Compute base loss $\mathcal{L}_{\text{base}} \leftarrow \frac{1}{|\mathcal{D}_{\text{probe}}|} \sum_x \ell(M_0(x), x)$
3: **for** $l \leftarrow 1$ to $L$ **do**
4:     Temporarily mask layer $l$ to get $M_0^{(-l)}$
5:     Compute masked loss $\hat{\mathcal{L}}^{(l)} \leftarrow \frac{1}{|\mathcal{D}_{\text{probe}}|} \sum_x \ell(M_0^{(-l)}(x), x)$
6:     Compute importance score $\Delta^{(l)} \leftarrow \hat{\mathcal{L}}^{(l)} - \mathcal{L}_{\text{base}}$
7: **end for**
8: Select $k$ least-important layers $\mathcal{S}_k \leftarrow \text{LowestK}(\{\Delta^{(l)}\})$
9: **for** each $l \in \mathcal{S}_k$ **do**
10:     Duplicate parameters $\tilde{\Theta}^{(l)} \leftarrow \Theta^{(l)}$                 ▷ Identity copy
11:     Initialize $W_{\text{MHSA}}^{\text{out}} = 0, W_{\text{FFN}}^{\text{out}} = 0$       ▷ Function Preserving Init
12:     Freeze original $\Theta^{(l)}$, mark $\tilde{\Theta}^{(l)}$ as trainable
13: **end for**
14: # Stage 2: Adaptive Unit-Wise Decoupled Tuning
15: **for** each training step $t$ **do**
16:     **if** $t \mod T_{\text{update}} == 0$ **then**
17:         **for** each expanded layer $\tilde{\Theta}^{(l)}$ **do**
18:             Partition into semantic units $\{U_1, \ldots, U_n\}$
19:             **for** each unit $U_i$ **do**
20:                 Compute gradient-based importance $I_{U_i} \leftarrow \frac{1}{|U_i|} \sum_{j \in U_i} \theta_j \cdot \nabla_{\theta_j} \mathcal{L}$
21:                 Assign adaptive learning rate $\text{lr}_{U_i} \leftarrow 2 \cdot (1 - I_{U_i}) \cdot \text{lr}_{\text{base}}$
22:             **end for**
23:         **end for**
24:     **end if**
25:     Sample training sequence $x = (x_1, x_2, \ldots, x_T) \sim \mathcal{D}_{\text{train}}$
26:     Compute autoregressive loss:
27:         $\mathcal{L} = -\sum_{t=1}^{T} \log P(x_t \mid x_{<t}; \Theta)$
28:     Update parameters $\{\tilde{\Theta}^{(l)}\}$ using adaptive learning rates $\{\text{lr}_{U_i}\}$
29: **end for**

---

# M   SENSITIVITY ANALYSIS OF IMPORTANCE-SCORE UPDATE INTERVALS

To assess the effect of the update frequency in Stage 2, we conduct a systematic sensitivity analysis on the interval at which unit importance scores are recomputed. While the main experiments adopt an update interval of 500 steps, it remains unclear how sensitive ADEPT is to more or less frequent updates. To this end, we perform additional experiments on Qwen3-4B-Base in the medical domain, evaluating intervals ranging from 10 to 5000 steps. This setup allows us to quantify how different recomputation frequencies influence model performance and overall training time. As shown in Table 16 and Table 17, we report both the medical-domain and general-domain performance of ADEPT under different update intervals, together with the corresponding training time (Descriptions of these newly added benchmarks and their evaluation protocols are provided in Appendix O). This allows us to clearly quantify how the recomputation frequency impacts domain-specific adaptation, general knowledge retention, and overall efficiency.

Interestingly, **the most frequent update setting (every 10 steps) does not yield the best results**. We hypothesize that this is due to training stability: excessively frequent adjustments of unit-wise learning rates may introduce noise, as the importance scores are estimated via a first-order approximation and thus inherently sensitive to stochastic fluctuations. The performance of the 500-step interval is largely comparable to that of the 100-step interval across benchmarks, while the latter already offers a substantially reduced training time comparing to 10-step interval. This further indicates that the **computational overhead of backpropagation dominates the overall training cost,**

Table 16: Sensitivity of ADEPT to the update interval for recomputing unit importance scores on `Qwen3-4B-Base` in the medical domain.

| Interval | PubMedQA (%) | MedQA (%) | MMCU (%) | CMB (%) | CMB-Clin (%) | Time |
|---|---|---|---|---|---|---|
| Qwen3-4B-Base | 73.60 | 62.77 | 82.44 | 78.92 | – | – |
| Step-10 | **77.60** | 63.26 | 83.51 | 79.04 | 54.12 | 5d 10h |
| Step-100 | 77.40 | 64.45 | 84.34 | **80.46** | **55.28** | 2d 12h |
| Step-500 | 77.20 | **64.49** | **84.58** | 79.87 | 54.40 | 2d 11h |
| Step-1000 | 77.40 | 62.97 | 82.62 | 79.44 | 53.64 | 2d 11h |
| Step-5000 | 77.20 | 62.05 | 81.48 | 77.58 | 52.26 | 2d 11h |

Table 17: Sensitivity of ADEPT to the update interval for recomputing unit importance scores on `Qwen3-4B-Base` on general-purpose benchmarks.

| Interval | TruthfulQA-MC1 (%) | TruthfulQA-MC2 (%) | CEval (%) | CMMLU (%) | BBH (%) | HellaSwag (%) | MMLU (%) | Time |
|---|---|---|---|---|---|---|---|---|
| Qwen3-4B-Base | 36.84 | 53.38 | 79.49 | 77.92 | 70.73 | **55.41** | **73.19** | – |
| Step-10 | 37.47 | 54.24 | **79.90** | 77.66 | 71.05 | 54.24 | 72.98 | 5d 10h |
| Step-100 | 37.68 | 54.33 | 79.68 | 78.60 | **71.51** | 55.21 | 73.01 | 2d 12h |
| Step-500 | **38.31** | **54.63** | 79.00 | **78.77** | 71.08 | 55.35 | 72.95 | 2d 11h |
| Step-1000 | 37.21 | 53.39 | 78.90 | 78.65 | 70.70 | 54.26 | 72.43 | 2d 11h |
| Step-5000 | 37.09 | 53.31 | 79.05 | 77.68 | 70.66 | 54.00 | 72.44 | 2d 11h |

**and that the recomputation of importance scores is relatively lightweight**. Notably, even an extremely infrequent update interval such as 5000 steps (i.e., only six updates throughout training) maintains competitive performance, **highlighting the robustness of ADEPT**.

We also oberserve the parameter importance during training and find that the overall distribution of importance across modules remains relatively stable after the initial update. Subsequent updates produce smaller changes, which explains ADEPT's stable effectiveness.

# N    APPLICABILITY TO SUPERVISED FINE-TUNING

To demonstrate ADEPT's transferability beyond continual pretraining, we further conduct Supervised Fine-Tuning (SFT) experiments in the medical domain on two widely used instruction-tuned backbones, `LLaMA3-8B-Instruct` and `Qwen3-8B`.

**SFT Experimental Setup.** For the SFT experiments, we fine-tune each model on the `MMedBench` training split (the SFT portion of the MMedC dataset). We adopt a mixed training regime that interleaves CoT-style samples (including a detailed `rationale`) and non-CoT samples (without rationales), with each item appearing once in both formats. We train with a global batch size of 128, a learning rate of $1 \times 10^{-5}$, and all other optimization and regularization hyperparameters identical to those used during CPT. LoRA is applied with rank 128. The full SFT dataset contains 75,156 samples, and all models are trained for 3 epochs. For ADEPT, we employ a single-layer expansion and recompute unit-importance scores every 100 steps, consistent with our CPT-stage procedures. To illustrate how each SFT item appears once in both CoT and non-CoT formats, we provide an example pair in Examples 4 and 5. Both correspond to the same underlying clinical question, with the former including a detailed rationale and the latter containing only the concise answer.

Overall, the SFT results reveal several noteworthy patterns. First, lightweight adaptation methods such as LoRA and TaSL perform relatively well in the SFT setting, substantially better than in CPT. This reflects the suitability of parameter-efficient adapters for supervised updates performed on relatively small training corpora. Second, fully updating all model parameters (SFT-Full) yields consistently weaker performance, likely due to the large degree of parameter perturbation introduced by full-model fine-tuning on a relatively small SFT dataset. Finally, while ADEPT does not exhibit the same magnitude of gains as in the CPT experiments, it **remains competitive and stable across both medical and general benchmarks.** Given the limited size of the supervised dataset used here,

Table 18: SFT results on `LLaMA3-8B-Instruct` with different adaptation strategies. "Vanilla" denotes the original pretrained model without any further training. All metrics are reported in %. Best and second-best scores for each metric are highlighted in bold and underlined, respectively.

| Method | Medical Domain | | | | | General Benchmarks | | | | | | |
| --- | --- | --- | --- | --- | --- | --- | --- | --- | --- | --- | --- | --- |
| | PubMedQA | CMB-Clin | MedQA | MMCU | CMB | CEval | TruthQA-MC1 | TruthQA-MC2 | CMMLU | BBH | HellaSwag | MMLU |
| Vanilla | **78.80** | - | 62.68 | 57.36 | 52.40 | 51.04 | 35.12 | 51.03 | **51.91** | 67.90 | 56.32 | 65.30 |
| SFT-Full | 75.60 | 47.28 | 55.06 | 57.90 | 50.20 | 42.64 | 35.49 | 42.64 | 43.57 | 19.96 | 52.82 | 55.74 |
| SFT-Lora | **78.80** | 52.88 | 60.56 | **61.61** | 53.34 | 51.63 | 37.45 | 51.63 | 51.69 | 67.86 | 57.53 | 64.28 |
| Llama-Pro | 78.20 | 51.44 | 61.74 | 57.46 | 51.66 | 51.26 | 35.98 | 51.26 | 51.78 | 66.76 | 57.06 | 65.24 |
| TaSL | 76.40 | 53.14 | 61.32 | 59.85 | 52.30 | **51.76** | 36.32 | 51.86 | 51.76 | 67.62 | **57.73** | 64.64 |
| ADEPT | **78.80** | **53.76** | **63.92** | 60.85 | **53.81** | 51.70 | 37.10 | **52.10** | 51.90 | **68.00** | 57.20 | **65.40** |

Table 19: SFT results on `Qwen3-8B` with different adaptation strategies. "Vanilla" denotes the original pretrained model without any further training. All metrics are reported in %. Best and second-best scores for each metric are highlighted in bold and underlined, respectively.

| Method | Medical Domain | | | | | General Benchmarks | | | | | | |
| --- | --- | --- | --- | --- | --- | --- | --- | --- | --- | --- | --- | --- |
| | PubMedQA | CMB-Clin | MedQA | MMCU | CMB | CEval | TruthQA-MC1 | TruthQA-MC2 | CMMLU | BBH | HellaSwag | MMLU |
| Vanilla | **78.40** | - | 63.55 | 79.88 | 73.17 | 78.75 | 35.86 | 53.49 | 77.97 | 79.18 | 55.85 | 74.59 |
| SFT-Full | 76.20 | 52.48 | 65.04 | 79.28 | 75.53 | 74.59 | 37.57 | 55.91 | 75.39 | 56.90 | 56.90 | 73.03 |
| SFT-Lora | 77.40 | 51.84 | **66.84** | 82.29 | 77.50 | 78.82 | **37.82** | 55.94 | 78.26 | 80.47 | 56.12 | 74.87 |
| Llama-Pro | 78.00 | **54.16** | 63.70 | 80.59 | 75.38 | 78.75 | 35.98 | 53.93 | 77.75 | 80.63 | 55.95 | 74.59 |
| TaSL | 77.20 | 52.14 | 63.94 | 78.71 | 75.65 | 78.45 | 36.71 | **56.32** | **78.41** | 80.06 | 56.92 | **75.09** |
| ADEPT | 77.80 | 53.36 | 65.94 | **83.02** | **79.51** | **78.90** | 36.96 | 54.60 | 77.98 | **81.21** | **57.06** | 74.72 |

we anticipate that ADEPT's benefits, particularly its targeted capacity expansion, would become more pronounced when scaling SFT to larger and more diverse training corpora.

---

**Example 4**

**Problem:** A 23-year-old pregnant woman at 22 weeks gestation presents with burning upon urination for one day, worsening despite increased fluid intake and cranberry extract. She is otherwise well and is followed regularly for prenatal care. Vitals are normal (T 97.7°F, BP 122/77 mmHg, HR 80/min, RR 19/min, SpO$_2$ 98%). Physical examination shows no costovertebral angle tenderness and a gravid uterus. Which of the following is the best treatment for this patient?

**Analysis:** This patient is pregnant, afebrile, and presents with symptoms of acute cystitis, with no signs of pyelonephritis. Safe antibiotic selection during pregnancy is essential:

- **Nitrofurantoin** is a first-line therapy for uncomplicated cystitis in pregnancy (especially after the first trimester), with an excellent safety profile.

- **Ampicillin** and **ceftriaxone** may be used for certain infections but are not first-line for simple cystitis in pregnant patients.

- **Doxycycline** is contraindicated during pregnancy due to risks of teratogenicity and adverse effects on fetal bone and tooth development.

Given her presentation and gestational age, nitrofurantoin is the safest and most appropriate therapy.
Therefore, the correct answer is **D. Nitrofurantoin**.

---

**Example 5**

**Problem:** A 23-year-old pregnant woman at 22 weeks gestation presents with dysuria and worsening urinary discomfort for one day. Vitals and examination are normal. Which of the following is the best treatment for this patient?
**Answer:** D. Nitrofurantoin.

## O  EXTENDED EVALUATION ON ADDITIONAL BENCHMARKS AND DOMAINS

In this section, we incorporate a substantially more diverse collection of domain-specific and general-purpose benchmarks to more rigorously evaluate the robustness and effectiveness of our method.

**Extended Medical-Domain Benchmarks.**   To enrich the evaluation of medical-domain capabilities, we additionally include two complementary benchmarks. **CMB-Clin** (Wang et al., 2023b) is an open-ended clinical QA benchmark containing 74 complex consultation questions (each question will be followed by several sub-questions and we will score each sub-question independently) across all major medical specialties. Each question is paired with a reference answer, and evaluation follows an LLM-as-a-judge protocol, where GPT-5 compares model outputs against the reference to compute a pairwise win rate to Vanilla model. (We omit the percent sign (%) in the presentation.) **PubMedQA** (Jin et al., 2019) is a biomedical question answering dataset constructed from PubMed abstracts. Each question asks whether a specific biomedical claim is supported by the evidence in the abstract, and the answer is one of *yes*, *no*, or *maybe*. The benchmark therefore evaluates a model's ability to read a short biomedical abstract and make an evidence-based judgment. We report accuracy following standard practice.

**Extended Mathmatic-Domain Benchmarks.**   To more comprehensively assess improvements in mathematical reasoning, we alsp include two challenging benchmarks that target advanced problem-solving skills. **GPQA** (Rein et al., 2024) is a multiple-choice benchmark of 448 expert-written questions designed to demand deep scientific reasoning and deliberate problem-solving. The questions are intentionally difficult, providing a rigorous testbed for evaluating whether Math-domain training improves a model's capacity for complex, expert-level reasoning. **GSM-Plus** (Li et al., 2024) is an adversarial extension of GSM8K that introduces controlled variations such as added statements and altered question targets to evaluate robustness in mathematical reasoning. These perturbations reduce reliance on pattern matching and require models to generalize their reasoning beyond surface cues, making GSM-Plus a stringent benchmark for assessing the stability of Mathmatic-domain training. We report accuracy for both benchmarks.

**Extended General-Capability Benchmarks.**   To more fully assess the general-capability performance of our trained models, we introduce a broader set of diverse and challenging benchmarks that are largely orthogonal to our existing evaluation suite. These additional datasets provide a more comprehensive evaluation of the model's overall general ability beyond the settings covered in our primary experiments. **BBH** (Suzgun et al., 2023) is a curated subset of 23 challenging tasks from BIG-Bench that remain difficult for LLMs. These tasks emphasize multi-step reasoning, abstraction, and compositional generalization, making BBH a stringent measure of cross-domain general capability. **HellaSwag** (Zellers et al., 2019) is a challenging commonsense inference benchmark created through adversarial filtering. Models must choose the most plausible continuation among highly confounding distractors, making it a strong test of robustness in everyday reasoning. **CE-val** (Huang et al., 2023) is a comprehensive Chinese exam-style benchmark covering a wide range of subjects and professional knowledge areas. It assesses broad general-domain understanding and factual reasoning through multiple-choice questions. For BBH, HellaSwag, and CEval, we report accuracy. **TruthfulQA** (Lin et al., 2022) evaluates a model's factuality by testing whether it can avoid reproducing widely held misconceptions. The benchmark comprises 817 questions across 38 domains, each constructed so that factually incorrect but popular answers are tempting. We report performance under both official TruthfulQA metrics. **MC1** evaluates single-answer multiple choice: given a question and candidate options, the model must select the uniquely correct answer, and accuracy is computed over all questions. **MC2** evaluates multi-answer probability assignment: given a question and sets of true and false reference answers, the score is the normalized total probability that the model assigns to the true answer set. MC1 thus measures strict answer accuracy, whereas MC2 assesses how much probability mass the model places on factually correct responses.

Table 20 reports the results under the *Medical domain*, where we directly reuse the same medical-domain checkpoint from Table 1 and evaluate it on additional medical benchmarks as well as broader general-capability tasks. The results show that ADEPT maintains consistently strong performance on the newly introduced medical benchmarks, demonstrating robust domain adaptation. Notably, ADEPT also exhibits a clear advantage in mitigating catastrophic forgetting on general-capability

Table 20: Comparison of ADEPT and baseline methods on a broad suite of medical and general benchmarks. All results are obtained using the **same medical-domain–trained checkpoint** reported in Table 1, here evaluated on additional medical and general tasks to further assess cross-domain generalization. For ease of comparison, all metrics are uniformly mapped to the $[0, 100]$ range.

| | Medical | | General | | | | |
|---|---|---|---|---|---|---|---|
| Method | PubMedQA | CMB-Clin | TruthQA-MC1 | TruthQA-MC2 | CEval | BBH | HellaSwag |
| *LLaMA-3-8B-Base* | | | | | | | |
| Vanilla | 76.60 | – | 26.81 | 43.95 | 50.45 | 62.33 | **60.46** |
| PT-Full | 77.00 | 56.84 | 23.75 | 37.74 | 49.70 | 47.47 | 58.03 |
| Replay | **78.20** | 59.20 | **29.50** | 44.15 | **52.75** | 51.48 | 58.71 |
| Llama-Pro | 76.40 | 54.92 | 24.85 | 38.03 | 49.70 | 62.31 | 60.00 |
| PT-LoRA | 73.20 | 53.48 | 24.24 | 38.16 | 49.93 | 61.77 | 59.79 |
| TASL | 72.40 | 50.00 | 26.35 | 39.46 | 50.51 | 61.92 | 58.34 |
| ADEPT | **78.20** | **59.70** | 28.40 | **45.03** | 52.11 | **62.52** | 60.39 |
| *Qwen3-1.7B-Base* | | | | | | | |
| Vanilla | 69.20 | – | 32.19 | 48.80 | 65.53 | **53.05** | 49.15 |
| PT-Full | 70.20 | 51.92 | 29.87 | 45.24 | 62.78 | 46.38 | 48.73 |
| Replay | 69.40 | 52.76 | 31.82 | 46.28 | 63.19 | 49.17 | 48.30 |
| Llama-Pro | **70.40** | 52.88 | 28.27 | 42.42 | 60.03 | 37.37 | 48.74 |
| PT-LoRA | 68.40 | 48.08 | 26.81 | 42.53 | 64.04 | 37.87 | 48.76 |
| TASL | 68.00 | 45.67 | 25.95 | 40.83 | 59.88 | 36.15 | 47.15 |
| ADEPT | 69.60 | **53.84** | **34.39** | **51.05** | **66.64** | 52.96 | **49.28** |
| *Qwen3-4B-Base* | | | | | | | |
| Vanilla | 73.60 | – | 36.84 | 53.38 | **79.49** | 70.73 | **55.41** |
| PT-Full | 76.60 | 52.64 | 33.05 | 47.10 | 73.11 | 63.22 | 53.15 |
| Replay | 76.40 | 51.64 | 33.05 | 48.83 | 73.18 | 65.32 | 50.79 |
| Llama-Pro | 76.60 | 52.16 | 34.27 | 49.33 | 77.71 | 69.37 | 52.82 |
| PT-LoRA | 74.00 | 51.36 | 33.90 | 49.08 | 78.01 | 58.02 | 53.54 |
| TASL | 72.60 | 51.80 | 35.11 | 47.54 | 76.43 | 60.33 | 53.76 |
| ADEPT | **77.20** | **54.40** | **38.31** | **54.63** | 78.60 | **71.08** | 55.35 |
| *Qwen3-8B-Base* | | | | | | | |
| Vanilla | 77.40 | – | 35.13 | 52.29 | **82.91** | **76.69** | **59.25** |
| PT-Full | 78.80 | 50.36 | 32.93 | 47.41 | 80.09 | 68.81 | 56.46 |
| Replay | **79.00** | 52.46 | 32.93 | 48.50 | 79.72 | 69.73 | 55.75 |
| Llama-Pro | 78.60 | 51.64 | 35.74 | 52.04 | 78.16 | 71.80 | 54.87 |
| PT-LoRA | 78.00 | 51.16 | 32.56 | 48.58 | 81.20 | 71.11 | 57.33 |
| TASL | 78.20 | 52.40 | 31.64 | 49.92 | 80.79 | 70.14 | 58.39 |
| ADEPT | **79.00** | 52.88 | 37.58 | 53.91 | 82.54 | 76.33 | 59.05 |

benchmarks, outperforming baseline methods by a large margin. Table 21 summarizes the results under the *Mathematical domain*. We observe similar trends: ADEPT retains strong mathematical reasoning performance on extended math benchmarks while preserving general abilities more effectively than competing approaches. These findings further **validate the robustness and effectiveness of ADEPT across heterogeneous domains**.

## P    CODE-DOMAIN EVALUATION

To further examine the applicability of ADEPT beyond natural-language and scientific domains, we additionally conduct experiments in the *Code domain*. Due to computational constraints, we select `Qwen3-4B` as a representative model and perform continual pretraining on a Python-only corpus constructed from two public datasets: `Swallow-Code-v2`[2] and `Python-Codes-25k`[3]. The

---

[2] https://hf-mirror.com/datasets/tokyotech-llm/swallow-code-v2
[3] https://hf-mirror.com/datasets/flytech/python-codes-25k

Table 21: Comparison of ADEPT and baseline methods on a broad suite of medical and general benchmarks. All results are obtained using the **same mathmatical-domain–trained checkpoint** reported in Table 1, here evaluated on additional medical and general tasks to further assess cross-domain generalization. For ease of comparison, all metrics are uniformly mapped to the $[0, 100]$ range. *GSM+* refers to the GSM-Plus benchmark.

| Method | Math | | General | | | | |
| | GPQA | GSM+ | TQA-MC1 | TQA-MC2 | CEval | BBH | HellaSwag |
|---|---|---|---|---|---|---|---|
| *LLaMA-3-8B* | | | | | | | |
| Vanilla | 22.32 | 30.03 | 26.80 | 43.94 | 50.44 | 62.33 | 60.46 |
| PT-Full | 23.99 | 33.19 | 29.13 | **45.09** | 46.80 | 62.93 | 55.60 |
| Replay | 27.23 | 33.62 | **29.25** | 44.14 | 51.70 | 62.51 | 56.10 |
| Llama-Pro | 25.44 | 29.54 | 26.68 | 43.22 | 50.96 | **63.51** | 58.65 |
| PT-LoRA | 25.22 | 29.90 | 27.78 | 43.46 | 50.14 | 62.75 | 59.11 |
| TASL | 24.86 | 30.26 | 27.71 | 41.68 | 49.68 | 62.37 | 58.91 |
| ADEPT | **27.78** | **34.44** | 28.78 | 44.31 | **51.89** | 63.47 | **60.77** |
| *Qwen3-1.7B-Base* | | | | | | | |
| Vanilla | 26.12 | 50.16 | 32.19 | 48.80 | 65.53 | 53.05 | 49.15 |
| PT-Full | 27.23 | 52.11 | 29.87 | 49.73 | 62.85 | 48.20 | 48.99 |
| Replay | 27.34 | 52.32 | 33.23 | **51.23** | 62.43 | 48.93 | 49.04 |
| Llama-Pro | 27.01 | 54.37 | 32.19 | 49.13 | 62.26 | 51.45 | 48.91 |
| PT-LoRA | 26.56 | 51.80 | 30.23 | 46.97 | 63.59 | 32.62 | 48.44 |
| TASL | 26.56 | 52.25 | 32.68 | 49.03 | 59.45 | 45.62 | 48.27 |
| ADEPT | **31.02** | **54.82** | **33.65** | 50.00 | **66.27** | **54.40** | **49.17** |
| *Qwen3-4B-Base* | | | | | | | |
| Vanilla | 27.68 | 61.62 | 36.84 | 53.38 | **79.49** | 70.73 | **55.41** |
| PT-Full | 26.44 | 57.37 | 31.82 | 47.83 | 74.07 | 68.80 | 50.61 |
| Replay | 26.10 | 57.87 | 32.06 | 49.14 | 75.48 | 69.73 | 51.94 |
| Llama-Pro | 27.23 | 61.74 | 33.05 | 49.44 | 77.27 | 71.11 | 52.34 |
| PT-LoRA | 26.89 | 60.26 | 31.33 | 47.31 | 77.71 | 71.80 | 51.64 |
| TASL | 26.86 | 62.63 | 32.31 | 48.18 | 77.43 | 70.83 | 53.23 |
| ADEPT | **29.90** | **63.76** | **38.31** | **54.25** | 78.26 | **72.21** | 55.29 |
| *Qwen3-8B-Base* | | | | | | | |
| Vanilla | 35.26 | 63.42 | 35.13 | 52.29 | **82.91** | 76.69 | **59.25** |
| PT-Full | 35.66 | 64.54 | 32.93 | 49.83 | 79.42 | 73.15 | 55.83 |
| Replay | 34.55 | 65.14 | 34.27 | 50.18 | 78.08 | 74.62 | 55.64 |
| Llama-Pro | 35.44 | 64.36 | 29.86 | 46.05 | 81.64 | 75.50 | 56.16 |
| PT-LoRA | 33.66 | **67.01** | 33.78 | 51.64 | 78.38 | 79.81 | 54.86 |
| TASL | 35.19 | 65.58 | 34.16 | 49.38 | 77.98 | 75.55 | 54.42 |
| ADEPT | **38.91** | 66.72 | **38.18** | **53.39** | 82.39 | **79.83** | 59.16 |

former provides a large, professionally curated Python corpus, while the latter contributes high-quality SFT-style Python tasks covering code generation, code-oriented natural language understanding, behavior analysis, and educational coding variations. Combined, the corpus contains approximately **13.7B tokens** ($\sim 4.2 \times 10^4$ samples of length $\approx 512$ tokens), enabling a controlled examination of ADEPT's behavior when adapting models to a specialized programming domain.

For downstream assessment, we adopt three widely used code domain benchmarks, all evaluated using the standard `pass@k` metric: **HumanEval** (Chen, 2021), a functional correctness benchmark for code synthesis; **MBPP** (Austin et al., 2021), a curated set of introductory-level programming tasks; and **CRUXEval** (Gu et al., 2024), which tests execution-based correctness across diverse constraint-solving problems. Together, these benchmarks provide a comprehensive view of coding capability under continual pretraining in the code domain.

Table 22 summarizes the results of Code-domain continual pretraining. Across the three code domain benchmarks, **ADEPT** demonstrates the most balanced and robust performance among lightweight adaptation methods. While `PT-Full` and `Replay` benefit from updating the entire

Table 22: Code-domain and general-capability results of `Qwen3-4B-Base` under different CPT strategies. All metrics are mapped to the $[0, 100]$ range. Best and second-best results for each column are highlighted in bold and underlined, respectively. *TQA* means *TruthfulQA*.

| Method | Code | | | | | | | | | General | | | | |
|---|---|---|---|---|---|---|---|---|---|---|---|---|---|---|
| | HumanEval | | | MBPP | | | CRUXEval | | | TQA-MC1 | TQA-MC2 | HellaSwag | BBH | CEval |
| | pass@1 | pass@5 | pass@10 | pass@1 | pass@5 | pass@10 | pass@1 | pass@5 | pass@10 | | | | | |
| Vanilla | 50.37 | 78.61 | 86.59 | 8.71 | 32.42 | 49.80 | **48.95** | **76.69** | **84.69** | **36.84** | 53.38 | **55.41** | 70.73 | 79.49 |
| PT-Full | 50.79 | 83.85 | **89.02** | 26.50 | 58.62 | 67.70 | 46.64 | 72.62 | 80.18 | 32.07 | 48.29 | 52.30 | 63.94 | 71.77 |
| Replay | **53.23** | 80.31 | 85.37 | 19.30 | 40.67 | 54.36 | 43.12 | 69.55 | 78.29 | 33.54 | 49.73 | 53.23 | 66.46 | 74.00 |
| Llama-Pro | 49.70 | 79.69 | 87.10 | 20.86 | 45.86 | 59.81 | 42.81 | 70.65 | 78.42 | 34.27 | 51.97 | 53.99 | 66.29 | 77.27 |
| PT-LoRA | 36.77 | 74.43 | 82.93 | 28.17 | **61.64** | **70.04** | 39.65 | 70.97 | 79.80 | 31.82 | 49.28 | 51.94 | 34.48 | 75.85 |
| TaSL | 37.42 | 75.21 | 84.37 | 25.94 | 58.19 | 66.53 | 40.18 | 70.05 | 80.63 | 32.45 | 48.70 | 52.67 | 33.96 | 77.21 |
| ADEPT | 51.13 | **84.81** | 87.20 | **31.17** | 55.07 | 67.98 | 46.68 | 74.57 | 83.69 | 35.13 | **53.83** | 54.98 | **70.77** | **79.63** |

Table 23: Multilingual medical and general-capability evaluation of `Qwen3-4B-Base` after continual pretraining on a multilingual medical corpus. All metrics are mapped to the $[0, 100]$ range, and best/second-best results are highlighted in bold and underlined. Language abbreviations: *ES (Spanish), FR (French), JA (Japanese), RU (Russian), EN (English), CN (Chinese). TQA* means *TruthfulQA*.

| Method | Multilingual Medical | | | | Multilingual General | | | | CN/EN General | | | | |
|---|---|---|---|---|---|---|---|---|---|---|---|---|---|
| | MMedBench | | | | MMMLU | | | | TQA-MC1 | TQA-MC2 | MMLU | CMMLU | CEval |
| | ES | FR | JA | RU | ES | FR | JA | RU | EN | EN | EN | CN | CN |
| Vanilla | 73.78 | 75.08 | 55.78 | 73.05 | **66.68** | **66.19** | 60.85 | 62.56 | **36.84** | 53.38 | 73.19 | 77.92 | **79.49** |
| PT-Full | 73.34 | **75.40** | **66.33** | 59.68 | 61.77 | 60.86 | 56.91 | 59.68 | 30.60 | 43.97 | 66.91 | 68.88 | 69.61 |
| Replay | 74.47 | 72.83 | 62.31 | 59.54 | 63.47 | 62.33 | 58.19 | 59.54 | 30.72 | 45.21 | 68.39 | 72.19 | 71.62 |
| Llama-Pro | 73.74 | 74.43 | 53.27 | 60.70 | 65.32 | 64.43 | 59.79 | 60.70 | 35.86 | 52.54 | 72.03 | 77.08 | 77.79 |
| PT-LoRA | 72.36 | 73.95 | 57.29 | 61.33 | 64.22 | 63.56 | 58.26 | 61.33 | 30.82 | 46.75 | 70.27 | 72.52 | 74.89 |
| TaSL | 71.22 | 73.88 | 56.45 | 61.58 | 65.49 | 62.59 | 58.97 | 60.49 | 30.70 | 45.67 | 70.40 | 71.65 | 76.98 |
| **ADEPT** | **74.56** | 74.92 | 62.26 | **74.61** | 66.64 | 66.15 | **61.01** | **63.19** | 36.60 | **54.01** | **73.69** | **78.67** | 78.63 |

model, ADEPT achieves competitive results despite modifying only four layers, and consistently outperforms `Llama-Pro` under the same expansion budget. LoRA shows clear instability across tasks, whereas ADEPT maintains stable performance across HumanEval, MBPP, and CRUXEval. In addition, ADEPT preserves general capabilities substantially better than full-parameter CPT and Replay, matching or exceeding `Llama-Pro` on most general benchmarks. Overall, **ADEPT provides a favorable trade-off, combining strong performance, stable generalization, and parameter efficiency**.

## Q   MULTILINGUAL MEDICAL EVALUATION

In Table 1, the continual pretraining data are predominantly English and Chinese. To further strengthen the domain-transfer setting and evaluate cross-lingual robustness, we extend our study to *multilingual medical tasks*. This setting introduces a substantially larger linguistic and domain distribution shift, providing a stringent test of whether ADEPT can retain its effectiveness under multilingual medical evaluation.

For CPT, we construct a multilingual medical corpus by combining the multilingual portion of `MMedC` with the *training split* of the multilingual portion of `MMedBench`, explicitly removing all English and Chinese samples and retaining only the remaining languages. The resulting corpus comprises approximately **16B tokens** spanning diverse medical subdomains and typologically varied languages. For evaluation, we use the multilingual section of the `MMedBench` test set to measure multilingual medical capability, and `MMMLU` (Hendrycks et al., 2020) to assess multilingual general capability. Finally, we report English and Chinese general capabilities using `TruthfulQA`, `MMLU`, `CMMLU`, and `CEval`.

Table 24: Layer-importance rankings for General, Math, Medicine, and calibrated variants (lower index = higher importance).

| Setting | Layer Importance Ranking |
|---|---|
| General | 0, 2, 1, 5, 4, 3, 6, 7, 9, 8, 15, 16, 10, 11, 13, 26, 20, 17, 14, 21, 24, 12, 18, 19, 22, 25, 27, 23 |
| Math | 0, 2, 9, 4, 3, 1, 5, 6, 16, 8, 15, 11, 7, 17, 21, 13, 10, 20, 24, 12, 18, 14, 19, 22, 26, 27, 23, 25 |
| Medicine | 2, 0, 5, 6, 4, 1, 7, 9, 3, 17, 11, 15, 21, 10, 8, 16, 14, 13, 12, 18, 20, 24, 19, 26, 22, 23, 27, 25 |
| Math (Calibrated) | 16, 15, 6, 8, 27, 24, 7, 9, 5, 14, 4, 11, 10, 18, 19, 0, 20, 21, 23, 12, 22, 13, 26, 17, 3, 2, 25, 1 |
| Medicine (Calibrated) | 16, 24, 6, 7, 15, 5, 23, 9, 10, 8, 11, 4, 14, 18, 19, 20, 21, 0, 27, 12, 2, 3, 22, 25, 17, 13, 1, 26 |

Table 23 presents the multilingual medical and general-capability evaluation results. A clear pattern emerges in the Japanese subset of MMedBench, where both `PT-Full` and `Replay` obtain the strongest scores among all methods. This aligns with the fact that the Japanese portion exhibits a relatively large distributional shift, and full-parameter as well as replay-based strategies tend to absorb such shifts more directly, yielding higher task-specific performance. However, this adaptation is accompanied by a noticeable decline in multilingual general capability and in CN/EN evaluations.

In contrast, ADEPT maintain consistently stronger CN/EN general performance, showing minimal degradation on TruthfulQA, MMLU, CMMLU, and CEval. Among all evaluated methods, ADEPT achieves the most balanced multilingual behavior. Its **importance-aware expansion and decoupled tuning effectively limit overfitting to individual languages while preserving broad generalization**, resulting in stable improvements across multilingual medical benchmarks and multilingual as well as CN/EN general tasks.

## R  DOMAIN-CRITICAL OR GENERAL-NONCRITICAL: WHICH LAYERS SHOULD BE EXPANDED?

**Motivation for Expanding General-Noncritical Layers.**   The design choice in ADEPT to expand layers that are least important for the general domain is grounded in several considerations. First, injecting new domain knowledge is relatively easy for gradient-based optimization, whereas catastrophic forgetting is far more harmful and can substantially degrade model abilities; prior work (Dai et al., 2022b; Geva et al., 2021a) has shown that domain-specific fine-tuning may improve in-domain performance but often causes large drops in general skills. Thus, ADEPT intentionally minimizes interference with general-critical parameters. Second, new domain-specific information may overwrite existing domain knowledge (e.g., medical factual updates), meaning that even accurately identifying domain-important layers does not guarantee preservation of previously learned domain-specific representations. In contrast, general-critical layers correspond to knowledge that must remain stable, making them a more reliable target for protection. Third, domain-important layers frequently overlap with general-critical layers due to shared semantic structures across domains, so expanding them still risks inducing forgetting. Finally, in practical settings, high-quality domain data may be insufficient to reliably identify domain-critical units, whereas general-noncritical units provide a safe and universal expansion region that works consistently across different domains.

**Probing and Expanding Domain-Critical Layers.**   To analyze whether expanding domain-critical layers could further improve adaptation, we first probe the importance distribution for the mathematics and medical domains and examine their overlap with the general-critical set. The probing procedure follows exactly the same strategy used for identifying general-critical layers. For the Medical domain, the probe corpus consists of 500 Chinese and 500 English samples drawn from the training split of `MMedBench`; for the Mathematics domain, we use the `math-500`[4] dataset.

Table 24 reports the probed layer-importance rankings for the General, Math, and Medical domains. We observe **substantial overlap** across these three distributions, with early layers consistently ranked as highly important in all domains. This pattern indicates that early transformer layers encode fundamental semantic representations shared across tasks and domains. As a result, **directly expanding domain-important layers would risk interfering with these shared representations, making them unsuitable as expansion targets during domain-specific continual pretraining**.

---

[4] https://huggingface.co/datasets/HuggingFaceH4/MATH-500

Table 25: Comparison of Math-domain and General performance between *domain-critical expansion* (calibrated) and *least-general-critical expansion* (default ADEPT) on `Qwen3-1.7B-Base`. All metrics are mapped to $[0, 100]$. Best and second-best results per column are marked in **bold** and underline, respectively.

| Method | Math-Domain Benchmarks | | | | | | General Benchmarks | | | | | |
|---|---|---|---|---|---|---|---|---|---|---|---|---|
| | GPQA | GSM+ | GSM8K | ARC-e | ARC-c | BBH | TQA-MC1 | TQA-MC2 | CEval | HellaSwag | MMLU | CMMLU |
| Qwen3-1.7B-Base | 26.12 | 50.16 | 57.62 | 81.44 | 51.19 | 53.05 | 32.19 | 48.80 | 65.53 | 49.15 | 62.57 | 66.86 |
| ADEPT (least-general-critical expansion) | 31.03 | **54.83** | 70.51 | **82.48** | 52.62 | 54.40 | **33.66** | **50.00** | 66.27 | **49.17** | **62.62** | **67.06** |
| ADEPT (domain-critical expansion) | **31.79** | 52.47 | **71.02** | 81.78 | **53.84** | **55.26** | 32.82 | 48.36 | 65.16 | 48.83 | 61.35 | 66.26 |

Table 26: Comparison of Medical-domain and General performance between *domain-critical expansion* (calibrated) and *least-general-critical expansion* (default ADEPT) on `Qwen3-1.7B-Base`. All metrics are mapped to $[0, 100]$. Best and second-best results per column are marked in **bold** and underline, respectively.

| Method | Medical-Domain Benchmarks | | | | | General Benchmarks | | | | | |
|---|---|---|---|---|---|---|---|---|---|---|---|
| | PubMedQA | CMB-Clin | MedQA | MMCU | CMB | TQA-MC1 | TQA-MC2 | CEval | HellaSwag | MMLU | CMMLU |
| Qwen3-1.7B-Base | 69.20 | – | 48.39 | 69.17 | 63.67 | 32.19 | 48.80 | 65.53 | 49.15 | 62.57 | 66.86 |
| ADEPT (least-general-critical expansion) | **69.60** | **53.84** | **50.75** | 71.98 | 65.43 | **34.39** | **51.05** | 66.64 | **49.28** | **62.80** | **66.89** |
| ADEPT (domain-critical expansion) | 67.00 | 52.92 | 49.70 | **72.55** | **66.82** | 32.31 | 48.30 | **66.90** | 48.80 | 61.69 | 66.15 |

Therefore, we introduce an *importance calibration* technique: for each layer we subtract its general-domain importance rank from its domain-specific importance rank, and use this calibrated score to identify layers that are important for the target domain but not important for the general domain(see the calibrated results in Table 24). We then perform layer expansion on this calibrated domain-important subset, while keeping the same decoupling strategy as in ADEPT. In subsequent experiments, we refer to this variant as domain-critical expansion.

Table 25 and Table 26 summarize the empirical results comparing calibrated domain-critical expansion with the original ADEPT strategy of expanding the least important layers for the general domain, across both Mathematics and Medical CPT settings. We observe that domain-critical expansion yields slightly stronger domain-knowledge injection, demonstrating that focusing on domain-specific importance can indeed enhance in-domain adaptation. However, this improvement consistently comes at the cost of **greater degradation on general benchmarks**. This behavior reflects an inherent trade-off between aggressive domain adaptation and preservation of broad general abilities.

These findings reaffirm the motivation behind ADEPT's original design: expanding the **least important layers for the general domain** minimizes interference with general-critical parameters and reduces the risk of catastrophic forgetting. While domain-critical expansion is a useful alternative when stronger domain specialization is desired, the general-noncritical strategy offers a more balanced and stable solution. We have therefore included the domain-critical expansion variant as an optional configuration in the released ADEPT codebase to support both adaptation preferences.

In the figures below, we present the domain importance profiles across all layers for both the medical and mathematical domains, including both the original (uncalibrated) and calibrated attention variants. Here, parameter importance may be more sensitive to fine-grained calibration, so we primarily focus on layer importance. We can observe that without calibration, the similarity in importance distribution across general, math, and medicine domains is remarkably high. We can observe that the early layers related to semantic understanding are of paramount importance across all tasks. Notably, from a layer-wise perspective, the medical domain exhibits important layers distributed throughout all layers, likely because medical tasks demand a complex combination of reasoning, factual recall, and contextual understanding. In contrast, the mathematical domain shows a concentration of importance in the middle or slightly earlier-middle layers, suggesting that the core computational and deductive processes for mathematical reasoning are primarily localized in these regions.

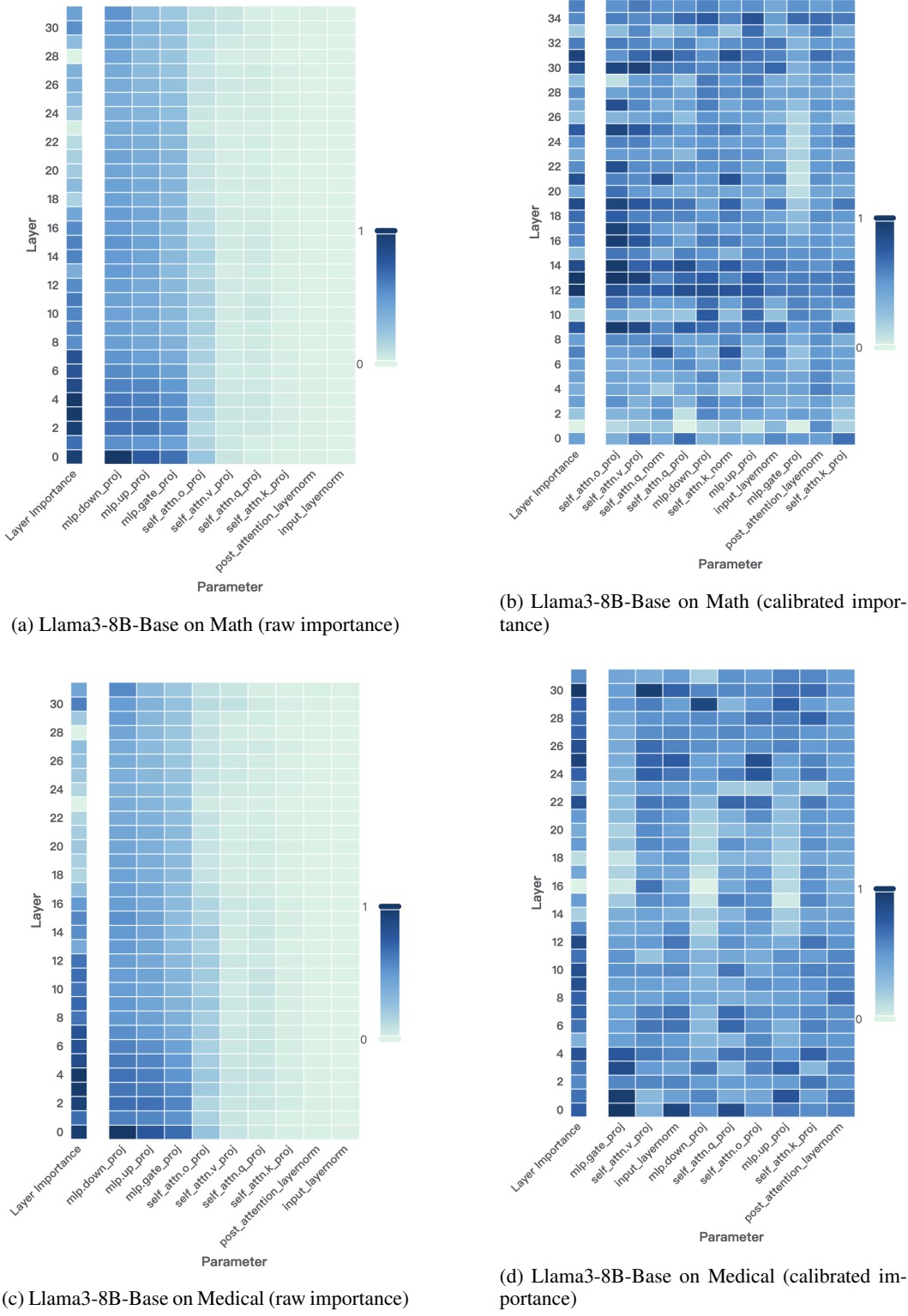

(a) Llama3-8B-Base on Math (raw importance)

(b) Llama3-8B-Base on Math (calibrated importance)

(c) Llama3-8B-Base on Medical (raw importance)

(d) Llama3-8B-Base on Medical (calibrated importance)

Figure 16: Importance visualization for Llama3-8B-Base across Math and Medical domains, with and without calibration.

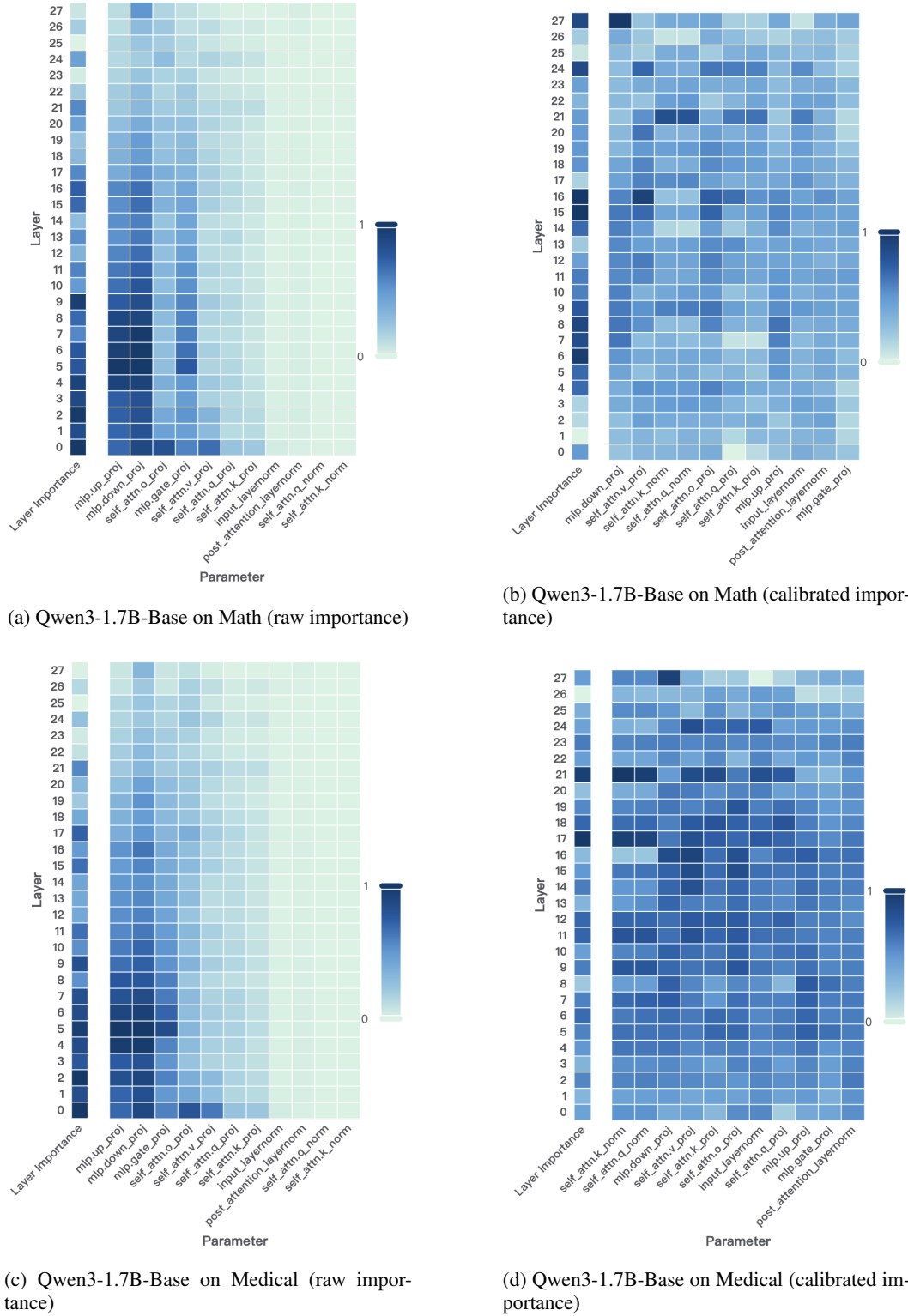

(a) Qwen3-1.7B-Base on Math (raw importance)

(b) Qwen3-1.7B-Base on Math (calibrated importance)

(c) Qwen3-1.7B-Base on Medical (raw importance)

(d) Qwen3-1.7B-Base on Medical (calibrated importance)

Figure 17: Importance visualization for Qwen3-1.7B-Base across Math and Medical domains, with and without calibration.

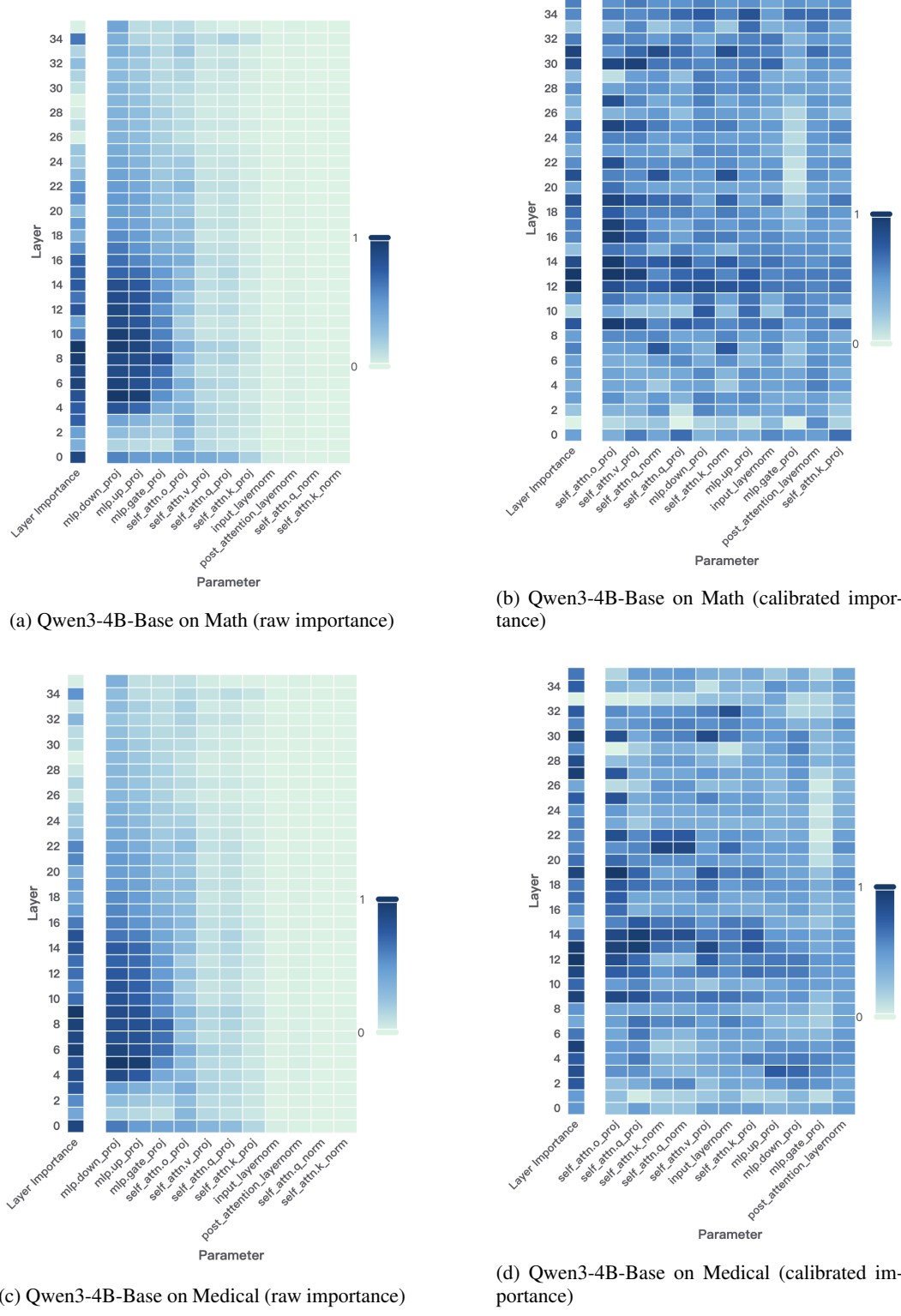

(a) Qwen3-4B-Base on Math (raw importance)

(b) Qwen3-4B-Base on Math (calibrated importance)

(c) Qwen3-4B-Base on Medical (raw importance)

(d) Qwen3-4B-Base on Medical (calibrated importance)

Figure 18: Importance visualization for Qwen3-4B-Base across Math and Medical domains, with and without calibration.

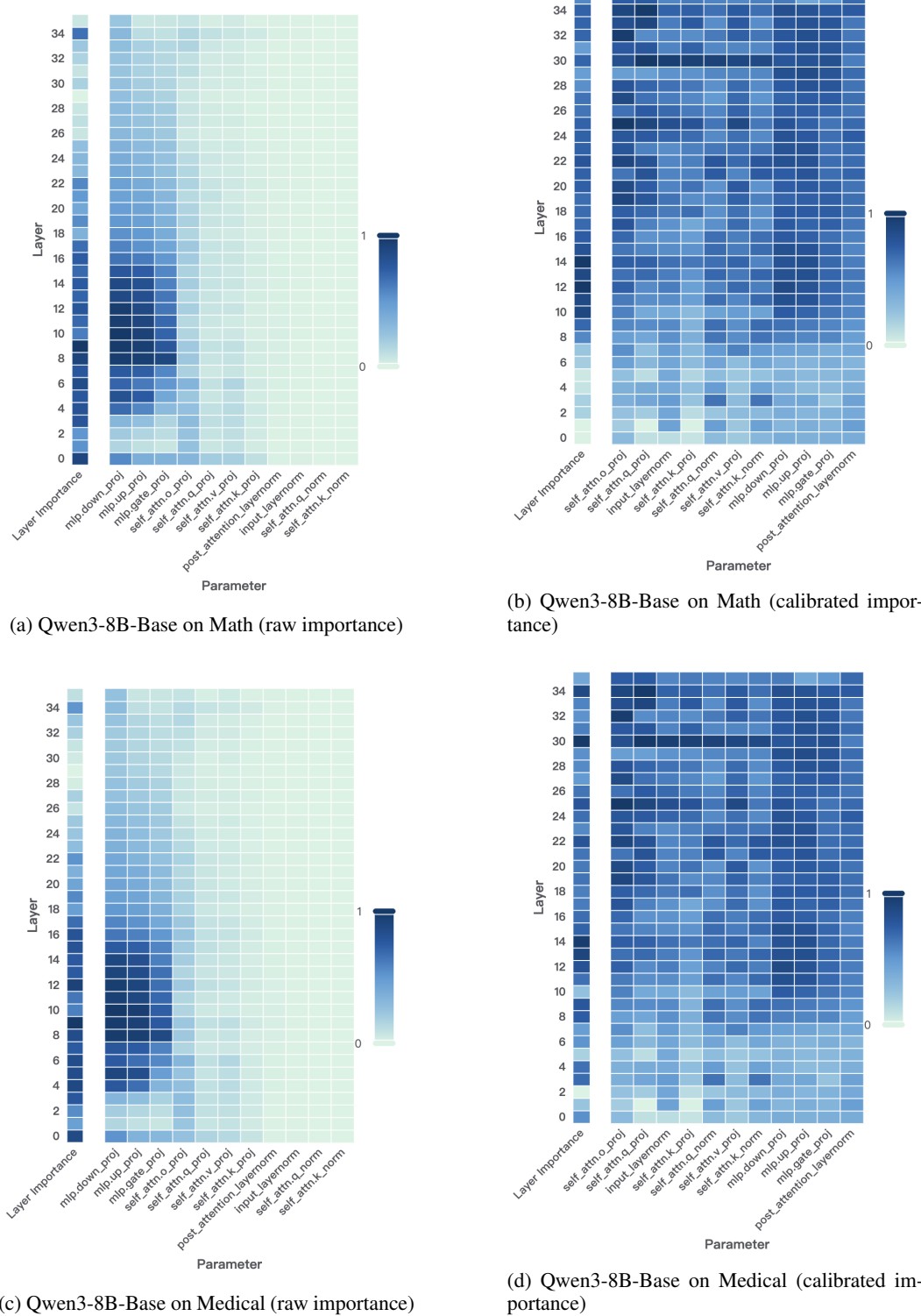

(a) Qwen3-8B-Base on Math (raw importance)

(b) Qwen3-8B-Base on Math (calibrated importance)

(c) Qwen3-8B-Base on Medical (raw importance)

(d) Qwen3-8B-Base on Medical (calibrated importance)

Figure 19: Importance visualization for Qwen3-8B-Base across Math and Medical domains, with and without calibration.

Table 27: Comparison of Math-domain and General performance between different zero-Initialization strategies on `Qwen3-1.7B-Base`. All metrics are mapped to $[0, 100]$. Best and second-best results per column are marked in **bold** and underline, respectively.

| Method | Math-Domain Benchmarks (%) | | | | | General Benchmarks (%) | | | | | | |
|---|---|---|---|---|---|---|---|---|---|---|---|---|
| | GPQA | GSM+ | GSM8K | ARC-e | ARC-c | BBH | TQA-MC1 | TQA-MC2 | CEval | HellaSwag | MMLU | CMMLU |
| Qwen3-1.7B-Base | 26.12 | 50.16 | 57.62 | 81.44 | 51.19 | 53.05 | 32.19 | 48.80 | 65.53 | 49.15 | 62.57 | 66.86 |
| ADEPT | **31.03** | **54.83** | 70.51 | **82.48** | **52.62** | **54.40** | **33.66** | **50.00** | 66.27 | 49.17 | 62.62 | 67.06 |
| ADEPT_up_projection | 29.01 | 53.14 | **71.11** | 82.10 | 51.62 | 51.82 | 32.68 | 47.92 | **66.34** | 48.54 | 62.32 | 66.72 |
| ADEPT_gate_projection | 26.89 | 54.19 | 69.58 | 81.81 | 51.79 | 52.53 | 33.29 | 47.88 | 66.20 | 48.61 | 62.28 | 66.33 |

# S  ANALYSIS OF ZERO-INITIALIZATION STRATEGIES IN MLP EXPANSION: UP-PROJECTION, DOWN-PROJECTION, AND GATE PROJECTION

## S.1  PRELIMINARY OBSERVATION: DOMINANT IMPORTANCE OF MLPS AND DIFFERENT ROLES OF PROJECTIONS.

The dominance of MLP layers observed in Figure 2 is not merely a consequence of their parameter scale, but also reflects their intrinsic representational role in storing and conveying factual knowledge, as supported by a growing body of interpretability research. Prior work has shown that feed-forward networks in transformers behave as key-value memories (Geva et al., 2021b), that individual "knowledge neurons" in FFNs encode relational and entity-level facts (Dai et al., 2022a), and that modifying mid-layer FFNs can directly alter factual outputs during generation (Meng et al., 2022). Additional studies further demonstrate that specific MLP pathways are repeatedly reused when reasoning with particular facts (Yao et al., 2024), reinforcing the view that MLPs serve as central repositories of semantic and factual information. Complementing this, large-scale analyses indicate that the factual capacity of MLPs grows linearly with parameter count (Nichani et al., 2025), underscoring that their quantitative dominance aligns with their qualitative representational role. Together, these findings explain why ADEPT's importance probe naturally highlights MLPs as high-value components for preserving and modifying general-domain competence.

Building on this understanding, zero-initializing the MLP down-projection (the output projection) emerges as a principled and effective design choice. First, this choice is consistent with established model-expansion techniques such as LLaMA-Pro, which similarly employ zero-initialized output projections to guarantee strict function preservation. Second, zeroing the down-projection ensures that the replicated MLP branch remains a lossless adapter, preserving the original key–value mappings without interfering with existing knowledge; formal justification is provided in Appendix F. Third, zero initialization actively facilitates knowledge acquisition: because ADEPT's decoupled update mechanism assigns learning rates inversely proportional to parameter importance, zeroing the new projection reduces its initial importance and thereby increases its learning rate. This allows the expanded region to rapidly absorb new information, matching the established view that MLPs act as the primary storage units for factual knowledge . Finally, zero-initializing the down-projection is particularly suitable for domain-specific knowledge updates. Since the down-projection corresponds to the "value" component of the MLP memory , resetting it permits efficient learning of new values while preserving the semantic keys encoded by the up-projection. This structure is desirable for controlled knowledge rewriting (e.g., updating medical facts), where modifying values while maintaining stable keys is essential. In contrast, zeroing the gate projection provides minimal semantic capacity for learning, and zeroing the up-projection disrupts key retrieval, making both alternatives less suitable for reliable and interpretable knowledge injection.

In summary, the empirical importance of MLPs, their theoretically grounded role as key-value memories, and the optimization dynamics introduced by ADEPT jointly motivate zero-initializing the down-projection as the most principled and effective strategy for stable, function-preserving model expansion.

## S.2  EXPERIMENT ON DIFFERENT LAYER EXPANSION STRATEGIES

To validate the correctness of our theoretical conjecture above, we employed different initialisation methods for layer expansion.

Table 28: Comparison of Medicine-domain and General performance between different zero-Initialization strategies on `Qwen3-1.7B-Base`. All metrics are mapped to $[0, 100]$. Best and second-best results per column are marked in **bold** and underline, respectively.

| Method | Medical-Domain Benchmarks (%) | | | | | General Benchmarks (%) | | | | | | |
|---|---|---|---|---|---|---|---|---|---|---|---|---|
| | PubMedQA | CMB-Clin | MedQA | MMCU | CMB | BBH | TQA-MC1 | TQA-MC2 | CEval | HellaSwag | MMLU | CMMLU |
| Qwen3-1.7B-Base | 69.20 | — | 48.39 | 69.17 | 63.67 | **53.05** | 32.19 | 48.80 | 65.53 | 49.15 | 62.57 | 66.86 |
| ADEPT | 69.60 | **53.84** | **50.75** | **71.98** | **65.43** | 52.96 | **34.39** | **51.05** | **66.64** | **49.28** | **62.80** | 66.89 |
| ADEPT_up_projection | **72.60** | 53.12 | 48.47 | 71.62 | 65.04 | 52.56 | 33.29 | 48.93 | 66.57 | 48.56 | 62.57 | **67.04** |
| ADEPT_gate_projection | 68.00 | 52.76 | 49.18 | 71.16 | 64.39 | 52.57 | 32.93 | 47.83 | 65.97 | 48.51 | 62.60 | 67.02 |

In our experiments, we observe that zero-initializing either the MLP up-projection or down-projection yields consistently strong performance across both mathematical and medical benchmarks, whereas zeroing the gate projection leads to a clear degradation in accuracy (Tables 27 and 28). This discrepancy arises from the distinct functional roles these projections play within the MLP and their interaction with the key–value structure of knowledge storage.

The up- and down-projections jointly form the semantic key–value mapping of the MLP: the up-projection builds high-dimensional semantic keys, while the down-projection retrieves values associated with those keys. Zero-initializing either side preserves the original key–value mapping of the pretrained MLP.

- When the *down-projection* is zeroed, the expanded branch produces no output at initialization, thus strictly maintaining functional equivalence with the original model. Meanwhile, the up-projection remains intact and continues to generate meaningful keys, enabling the expanded subspace to learn new value information. This behavior aligns with prior preliminary research on controlled knowledge injection and with theoretical views of MLPs as fact-storage modules.

- When the *up-projection* is zeroed, the semantic keys of the expanded branch are reinitialized, while the original MLP's key–value mapping remains untouched. Although this limits the ability of the expanded dimensions to encode richer key structures, the overall performance remains comparable to zero-down initialization, consistent with the small empirical gap observed in Tables 27 and 28.

In contrast, the gate projection plays a fundamentally different role from the up- and down-projections, which primarily modulate the dynamic importance of activation channels and govern amplitude-level feature selection, rather than providing a semantic space suitable for storing or modifying knowledge. Zeroing the gate projection, therefore, does not offer the expanded dimensions any meaningful representational capacity for knowledge injection or rewriting. This misalignment is reflected in the empirical results: zero-gate consistently underperforms zero-up and zero-down across mathematical reasoning tasks and medical tasks in Tables 27 and 28. In contrast, zero-up and zero-down preserve the core semantic pathways of the MLP, enabling the expanded dimensions to learn new information effectively. Consequently, these results reinforce the distinct functional decomposition of MLPs into key–value mapping and gating components in our preliminaries, and they offer practical guidance for selecting initialization strategies in ADEPT's layer expansion.

