# OpenReview forum: "ADEPT: Continual Pretraining via Adaptive Expansion and Dynamic Decoupled Tuning"
_ICLR.cc/2026/Conference — ICLR 2026 Poster_

### Official Review · Reviewer_nRAZ · 2025-10-27

**Soundness:** 3
**Presentation:** 4
**Contribution:** 3
**Rating:** 8
**Confidence:** 4

**Summary:**

This paper introduces **ADEPT** (Adaptive Expansion and Dynamic Decoupled Tuning), a two-stage framework designed to improve continual pretraining (CPT) for large language models (LLMs) when adapting them to specialized domains like mathematics and medicine. The primary challenges it addresses are catastrophic forgetting (losing general knowledge) and the limited capacity of models to absorb new information. The method is based on the key insight that LLMs exhibit functional specialization, where different layers and parameters have varied importance for preserving general knowledge. In its first stage, ADEPT selectively expands the model by duplicating layers least critical to general competence, adding capacity with minimal interference. The second stage performs decoupled tuning on these new layers, applying asymmetric learning rates to protect general-critical parameter units while allowing adaptable ones to absorb domain knowledge. Experiments demonstrate that ADEPT outperforms the baselines in both target and general domain performance, while being more efficient by tuning only small number of parameters in less than half the training time.

**Strengths:**

The paper demonstrates a fair degree of originality by effectively combining existing ideas in a novel manner. The significance of the work is clear, as it addresses an catastrophic forgetting problem in LLM area. The clarity of the writing is commendable, with proper use of English grammar and well-designed visualizations that aid in understanding the proposed methods.

- **Novelty Through Methodological Rigor and Transparency**: While the individual components—such as parameter importance analysis, layer expansion, and adaptive tuning—are not entirely new, the paper’s novelty stems from the rigorous process used to select and combine them. The authors build a compelling case for their approach by meticulously justifying their unique combination of existing techniques. This is best illustrated in the appendix, where a systematic comparison of four different methods for probing layer importance is presented. By transparently demonstrating that their chosen "masking out" strategy delivers superior performance, the authors provide a clear, evidence-based rationale for their design choice, which greatly enhances the reader's understanding and trust in the proposed method.

- **Significance and Strong Motivation**: The work addresses the critical and highly relevant problem of catastrophic forgetting during domain-adaptive continual pretraining, a key challenge for deploying specialized LLMs. The authors establish a compelling motivation not just by stating the problem, but by conducting a pilot study (Section 2) to provide an empirical foundation for their core hypothesis of "functional specialization". This initial analysis, which demonstrates that different layers and parameter units contribute unequally to preserving general knowledge, serves as a powerful justification for the entire ADEPT framework.

- **Exceptional Clarity and Presentation**: The paper is written with outstanding clarity, making the complex methodology accessible. The logical flow from the problem statement to the empirical validation is seamless. This is further enhanced by well-designed visualizations; for instance, the main methodology diagram (Figure 3) provides an intuitive step-by-step overview of the ADEPT process , while the analytical figures on activation distributions (Figure 4) and token shifts (Figure 5) offer insightful, visual proof of the method's effectiveness.

**Weaknesses:**

While the paper is methodologically sound and the results are compelling, its primary weakness lies in the limited scope of its experimental validation, which could be expanded to further solidify the claims of generality.

* **Scope of Domain Adaptation**: The authors make a strong case for ADEPT's effectiveness by testing on two distinct and challenging domains: Mathematics (emphasizing reasoning) and Medicine (emphasizing factual knowledge). This is a commendable choice. However, the claim of the framework's general applicability would be significantly bolstered by including experiments in other complex domains. For instance, code generation represents a crucial application area for LLMs that combines logical structure, strict syntax, and algorithmic reasoning. Demonstrating ADEPT's success in adapting a model to a specific programming language or codebase would provide more robust evidence of its versatility.

**Questions:**

Thank you for this well-written and insightful paper. The proposed ADEPT framework is an elegant and effective solution to a significant problem in continual learning for LLMs. The empirical results are strong and the analysis is thorough. To further strengthen the work and clarify some points for the reader, I have the following questions and suggestions:

1. **On the Generality of the Framework**: The choice of Mathematics and Medicine as test domains is excellent, as they represent distinct challenges (reasoning vs. factual knowledge). However, to fully substantiate the claim of the framework's general applicability, further evidence would be beneficial.
    - Question: Have the authors considered evaluating ADEPT on a domain with fundamentally different characteristics, such as code generation? Success in a domain governed by strict syntax and algorithmic logic would provide powerful evidence for the method's versatility.

2. **On the Dynamics of Decoupled Tuning**: The methodology for Stage 2 mentions that unit importance scores are periodically recomputed to keep the learning rates adaptive throughout training.
    - Question: Could the authors please specify the update interval for these importance scores used in the experiments (e.g., every 500 steps, as noted in the appendix )? Further, was any sensitivity analysis performed on this interval? It would be interesting to know if performance is robust to less frequent updates, which could further improve efficiency.

3. **Minor Proofreading and Formatting Suggestions**: I noticed a few minor typos that could be easily addressed. Could the authors perform a final proofread to catch minor errors? A few examples I noted include:

    - Page 2, Figure 1: The label "Target Domian Extension" appears to have a typo.

    - Page 3, Section 2.2: In the first paragraph, the corpus is described as something that "servers as the probing ground". The correct grammar would be "serves as the probing ground".

    - Page 29, Appendix E: In the table 9, the method name "Importance Cumulatation" is misspelled. It should likely be "Importance Cumulation" or "Importance Accumulation".

---

> ### Author Response · Authors · 2025-11-22
> **Response to Reviewer nRAZ (1/2)**
>
> Dear Reviewer nRAZ:
>
> We sincerely appreciate your thoughtful comments and the positive assessment of our work, and for noting your deep familiarity with the domain. We address your questions in detail below.
>
> ## **W1&Q1: Apply ADEPT on code and multilingual domains**
> To evaluate ADEPT’s versatility beyond the math and medical domains, we conduct additional experiments in two challenging settings: **code generation and multilingual medical transfer**. The detailed experimental setup and full results are provided in the updated manuscript (Appendix P–Q). Experimental results show that our ADEPT method also achieves strong performance on more challenging domains.
>
> **Code Domain:** Code is an ideal stress test because it simultaneously requires strict grammatical correctness and multi-step logical reasoning, and is also of high practical importance. The pre-training data includes _swallow-code-v2 and python-codes-25k_, totaling **13.7B tokens**. We evaluate on three complementary code-generation datasets, along with general-domain benchmarks:
>
> **Table: Performance of Qwen3-4B trained on code domain under different CPT methods**
> |**Domain**|**Code**|-|-|-|-|-|-|-|-|**General**|-|-|-|-|
> |-|-|-|-|-|-|-|-|-|-|-|-|-|-|-|
> |**Method**|HumanEval pass@1|pass@5|pass@10|MBPP pass@1|pass@5|pass@10|CRUXEval pass@1|pass@5|pass@10|TQA-MC1|TQA-MC2|HellaSwag|BBH|CEval|
> |Vanilla|50.37|78.61|86.59|8.71|32.42|49.80|**48.95**|**76.69**|**84.69**|**36.84**|_53.38_|**55.41**|_70.73_|_79.49_|
> |PT-Full|50.79|_83.85_|**89.02**|26.50|_58.62_|67.70|46.64|72.62|80.18|32.07|48.29|52.30|63.94|71.77|
> |PT-LoRA|36.77|74.43|82.93|_28.17_|**61.64**|**70.04**|39.65|70.97|79.80|31.82|49.28|51.94|34.48|75.85|
> |Llama-Pro|49.70|79.69|87.10|20.86|45.86|59.81|42.81|70.65|78.42|34.27|51.97|53.99|66.29|77.27|
> |Replay|**53.23**|80.31|85.37|19.30|40.67|54.36|43.12|69.55|78.29|33.54|49.73|53.23|66.46|74.00|
> |ADEPT|_51.13_|**84.81**|_87.20_|**31.17**|55.07|_67.98_|_46.68_|_74.57_|_83.69_|_35.13_|**53.83**|_54.98_|**70.77**|**79.63**|
>
>
> **Multilingual Medical Domain:** To evaluate ADEPT in multilingual transfer settings where the model must learn new knowledge and new languages simultaneously, we train on _MMedC_ excluding Chinese and English, covering French, Japanese, Russian, Spanish, totaling 16B tokens, and evaluate on multilingual medical benchmarks.
>
> **Table: Performance of Qwen3-4B trained on multilingual corpora under different CPT methods**
> |**Domain**|**Multilingual Medical**|-|-|-|**Multilingual General**|-|-|-|**CN/EN General**|-|-|-|-|
> |-|-|-|-|-|-|-|-|-|-|-|-|-|-|
> |**Method**|MMedBench ES|FR|JA|RU|MMMLU ES|FR|JA|RU|TQA-MC1|TQA-MC2|MMLU|CMMLU|CEval|
> |Vanilla|73.78|_75.08_|55.78|_73.05_|_66.68_|**66.19**|_60.85_|_62.56_|**36.84**|_53.38_|_73.19_|_77.92_|**79.49**|
> |PT-Full|73.34|**75.40**|**66.33**|59.68|61.77|60.86|56.91|59.68|30.60|43.97|66.91|68.88|69.61|
> |LoRA|72.36|73.95|57.29|61.33|64.22|63.56|58.26|61.33|30.82|46.75|70.27|72.52|74.89|
> |Llama-Pro|73.74|74.43|53.27|60.70|65.32|64.43|59.79|60.70|35.86|52.54|72.03|77.08|77.79|
> |Replay|_74.47_|72.83|_62.31_|59.54|63.47|62.33|58.19|59.54|30.72|45.21|68.39|72.19|71.62|
> |ADEPT|**74.56**|74.92|62.26|**74.61**|**66.64**|_66.15_|**61.01**|**63.19**|_36.60_|**54.01**|**73.69**|**78.67**|_78.63_|
>
> Overall, across both new settings, ADEPT exhibits robust and stable improvements and outperforms baselins, demonstrating its effectiveness in handling complex tasks.

---

> ### Author Response · Authors · 2025-11-22
> **Response to Reviewer nRAZ (2/2)**
>
> ## **Q2: Sensitivity to probing frequency**
> Update interval for importance scores used in our experiments is indeed every 500 steps.
>
> To assess ADEPT's sensitivity to the frequency of importance-score recomputation, we conduct experiments on Qwen3-4B-Base in **medical domain CPT**, **varying update intervals from 10 to 5,000 steps**. We report the model performance in the target and general domains, as well as the training times of each variants (more details in the revised manuscript **Appendix M**). This allows us to quantify how different recomputation frequencies affect ADEPT’s performance and stability.
>
> **Table: Medical domain performance under varying probing intervals.**
>
> |**Interval**|PubMedQA (%)|MedQA (%)|MMCU (%)|CMB (%)|CMB-Clin (%)|Time|
> |-|-|-|-|-|-|-|
> |Baseline|73.60|62.77|82.44|78.92|--|--|
> |Step-10|**77.60**|63.26|83.51|79.04|54.12|5d 10h|
> |Step-100|_77.40_|_64.45_|_84.34_|**80.46**|**55.28**|2d 12h|
> |Step-500|77.20|**64.49**|**84.58**|_79.87_|_54.40_|2d 11h|
> |Step-1000|_77.40_|62.97|82.62|79.44|53.64|2d 11h|
> |Step-5000|77.20|62.05|81.48|77.58|52.26|2d 11h|
>
> **Table: General domain performance under varying probing intervals.**
>
> |**Interval**|TruthfulQA-MC1 (%)|TruthfulQA-MC2 (%)|CEval (%)|CMMLU (%)|BBH (%)|HellaSwag (%)|MMLU (%)|Time|
> |-|-|-|-|-|-|-|-|-|
> |Baseline|36.84|53.38|79.49|77.92|70.73|**55.41**|**73.19**|--|
> |Step-10|37.47|54.24|**79.90**|77.66|71.05|54.24|72.98|5d 10h|
> |Step-100|_37.68_|_54.33_|_79.68_|78.60|**71.51**|55.21|_73.01_|2d 12h|
> |Step-500|**38.31**|**54.63**|79.00|**78.77**|_71.08_|_55.35_|72.95|2d 11h|
> |Step-1000|37.21|53.39|78.90|_78.65_|70.70|54.26|72.43|2d 11h|
> |Step-5000|37.09|53.31|79.05|77.68|70.66|54.00|72.44|2d 11h|
>
> Key observations include:
> 1. **Frequent updates are not always optimal.** Step-10 does not consistently produce the best overall performance, indicating the importance of **training stability**, as frequent modifications of learning rates may introduce instability.
> 2. **Step-100 and Step-500 show comparable strong performance.** Across most benchmarks, Step-500 and Step-100 intervals achieve similar results, outperforming the others while reducing computation time.
> 3. **Long intervals maintain reasonable performance.** Even with a 5000-step interval, performance drops only slightly, indicating ADEPT's stability. We also oberserve the parameter importance during training and find that the overall distribution of importance across modules remains relatively stable after the initial update. Subsequent updates produce smaller changes, which explains ADEPT's stable effectiveness.
>
> Our results show that **ADEPT is robust across a wide range of update frequencies**.
>
>
> ## **Q3: Response to Minor Proofreading and Formatting Suggestions**:
> We thank the reviewer for carefully pointing out these typos. We have corrected all mentioned issues, including the labels in Figure 1, the wording in Section 2.2, and the spelling in Table 9 of Appendix E. We will also conduct a final thorough proofreading pass to ensure consistency and correctness throughout the paper.

---

> ### Author Response · Authors · 2025-11-27
> **Gentle Follow‑Up on Our Discussion and Revisions**
>
> Dear Reviewer nRAZ,
>
> We hope this message finds you well. We sincerely appreciate the time and expertise you have devoted to reviewing our manuscript. Your thoughtful comments and positive assessment of our work have been truly encouraging, and we are deeply grateful for your constructive suggestions.
> In response to your feedback, we have carefully revised and strengthened our manuscript. Specifically, we have:
> + Added comprehensive new experiments in both the code generation domain and multilingual medical transfer, demonstrating that ADEPT consistently achieves robust and balanced improvements across these more challenging settings.
> + Conducted a detailed sensitivity study on the probing‑frequency schedule, evaluating update intervals from 10 to 5,000 steps and reporting their effects on both performance and training time. The results show that ADEPT remains stable and effective across a wide range of frequencies.
> + Provided additional explanations regarding the stability of importance‑score updates during training, clarifying why fewer updates remain effective in practice.
> + Corrected the typos and minor formatting issues you kindly pointed out, and performed an additional proofreading pass to ensure clarity and consistency throughout the manuscript.
>
> We sincerely hope that these revisions have addressed your concerns. Your insights have significantly improved the clarity and completeness of our work, and we would be very grateful for any further thoughts you might be willing to share.
> As the discussion phase is drawing to a close, we fully understand that your schedule may be demanding. If you could spare a moment to let us know whether our updates have sufficiently resolved your concerns, it would mean a great deal to us.
> Thank you again for your time, thoughtful evaluation, and invaluable guidance.
>
> Warmest regards,
>
> All Authors of Submission #2077

---

> > ### Comment · Reviewer_nRAZ · 2025-11-28
> >
> > I appreciate the authors' detailed response and the effort put into the additional experiments during the discussion phase.
> >
> >
> > However, upon reviewing the new results for the Coding and Multilingual medical tasks, I noticed that **TASL**, which served as a key baseline in the main experiments, was not included. Could you please clarify the reason for excluding this baseline in these specific domains?”

---

> > > ### Author Response · Authors · 2025-11-28
> > > **Sincere Appreciation and Response to Reviewer nRAZ's Feedback**
> > >
> > > Dear Reviewer nRAZ,
> > >
> > > Thank you very much for your careful reading of our updated experiments and for noticing the absence of TaSL in the newly added Coding and Multilingual medical results. We greatly appreciate your attention to detail.
> > >
> > > Let us clarify the situation. In our domain‑adaptation CPT setting, TaSL did not show consistently strong performance compared with the other baselines. For example, across the original experiments, **TaSL generally performs close to or slightly below PT‑LoRA and Llama‑Pro on average**, and it is often not among the strongest methods in either the general or domain‑specific metrics. Meanwhile, methods such as Replay or PT‑Full tend to achieve higher or more stable average scores across domains, and ADEPT shows the best overall results in most configurations. Because of this pattern, TaSL was not the most representative baseline for highlighting differences in domain‑adaptive CPT performance.
> > >
> > > In the newly added Coding and Multilingual medical tasks, LoRA‑based methods exhibit particularly limited effectiveness. For instance, PT‑LoRA mostly underperforms other baselines, and ADEPT in both the Coding benchmarks (HumanEval, CruxEval) and the Multilingual tasks (MMedBench and MMMLU across ES/FR/JA/RU) in most cases. **Since TaSL is built upon LoRA, we expected similar limitations to apply**. Under our time and compute constraints during the discussion phase, we prioritized running baselines that demonstrated stronger or more stable behavior in these more challenging domains.
> > >
> > > That said, we fully recognize that our experimental results are incomplete without the corresponding TaSL runs. We sincerely apologize for not finishing them in our former response. We want to assure you that **we have already started running the missing TaSL experiments, and we expect to complete them and provide the full results within the next 24 hours. We will notify you immediately once they are ready.**
> > >
> > >
> > > Thank you again for your thoughtful comments and for carefully examining the updated content. We appreciate your patience and your support for our work.
> > >
> > > Best regards,
> > >
> > > All Authors of Submission #2077

---

> > > > ### Author Response · Authors · 2025-11-29
> > > > **Updated TaSL Results on Coding and Multilingual Medical Tasks**
> > > >
> > > > Dear Reviewer nRAZ,
> > > >
> > > > Thank you again for your patience. As promised, we have completed the missing TaSL experiments for both the Coding and Multilingual medical domains. We sincerely appreciate your earlier feedback, which helped us improve the completeness of our study.
> > > > The newly added results are shown below.
> > > >
> > > > **Table: Performance of Qwen3-4B trained on code domain under different CPT methods**
> > > >
> > > > | Method      | HumanEval pass@1 | pass@5 | pass@10 | MBPP pass@1 | pass@5 | pass@10 | CRUXEval pass@1 | pass@5 | pass@10 | TQA-MC1 | TQA-MC2 | HellaSwag | BBH   | CEval  |
> > > > |-------------|------------------|--------|---------|--------------|--------|---------|------------------|--------|---------|----------|----------|-----------|--------|--------|
> > > > | Vanilla     | 50.37            | 78.61  | 86.59   | 8.71         | 32.42  | 49.80   | **48.95**        | **76.69** | **84.69** | **36.84** | _53.38_ | **55.41** | _70.73_ | _79.49_ |
> > > > | PT-Full     | 50.79            | _83.85_| **89.02**| 26.50        | _58.62_| 67.70   | 46.64            | 72.62  | 80.18   | 32.07    | 48.29    | 52.30     | 63.94  | 71.77  |
> > > > | Replay      | **53.23**        | 80.31  | 85.37   | 19.30        | 40.67  | 54.36   | 43.12            | 69.55  | 78.29   | 33.54    | 49.73    | 53.23     | 66.46  | 74.00  |
> > > > | Llama-Pro   | 49.70            | 79.69  | 87.10   | 20.86        | 45.86  | 59.81   | 42.81            | 70.65  | 78.42   | 34.27    | 51.97    | 53.99     | 66.29  | 77.27  |
> > > > | PT-LoRA     | 36.77            | 74.43  | 82.93   | _28.17_      | **61.64**| **70.04**| 39.65           | 70.97  | 79.80   | 31.82    | 49.28    | 51.94     | 34.48  | 75.85  |
> > > > | TaSL        | 37.42            | 75.21  | 84.37   | 25.94        | 58.19  | 66.53   | 40.18            | 70.05  | 80.63   | 32.45    | 48.70    | 52.67     | 33.96  | 77.21  |
> > > > | ADEPT       | _51.13_          | **84.81**| _87.20_| **31.17**    | 55.07  | _67.98_| _46.68_          | _74.57_| _83.69_| _35.13_  | **53.83**| _54.98_   | **70.77**| **79.63**|
> > > >
> > > > **Table: Performance of Qwen3-4B trained on multilingual corpora under different CPT methods**
> > > >
> > > > | Method      | MMedBench ES | MMedBench FR | MMedBench JA | MMedBench RU | MMMLU ES | MMMLU FR | MMMLU JA | MMMLU RU | TQA-MC1 | TQA-MC2 | MMLU   | CMMLU  | CEval  |
> > > > |-------------|--------------|--------------|--------------|--------------|----------|----------|----------|----------|----------|----------|--------|--------|--------|
> > > > | Vanilla     | 73.78        | _75.08_      | 55.78        | _73.05_      | _66.68_  | **66.19**| _60.85_  | _62.56_  | **36.84**| _53.38_  | _73.19_| _77.92_| **79.49**|
> > > > | PT-Full     | 73.34        | **75.40**    | **66.33**    | 59.68        | 61.77    | 60.86    | 56.91    | 59.68    | 30.60    | 43.97    | 66.91  | 68.88  | 69.61  |
> > > > | Replay      | _74.47_      | 72.83        | _62.31_      | 59.54        | 63.47    | 62.33    | 58.19    | 59.54    | 30.72    | 45.21    | 68.39  | 72.19  | 71.62  |
> > > > | Llama-Pro   | 73.74        | 74.43        | 53.27        | 60.70        | 65.32    | 64.43    | 59.79    | 60.70    | 35.86    | 52.54    | 72.03  | 77.08  | 77.79  |
> > > > | PT-LoRA | 72.36        | 73.95        | 57.29        | 61.33        | 64.22    | 63.56    | 58.26    | 61.33    | 30.82    | 46.75    | 70.27  | 72.52  | 74.89  |
> > > > | TaSL        | 71.22        | 73.88        | 56.45        | 61.58        | 65.49    | 62.59    | 58.97    | 60.49    | 30.70    | 45.67    | 70.40  | 71.65  | 76.98  |
> > > > | ADEPT       | **74.56**    | 74.92        | 62.26        | **74.61**    | **66.64**| _66.15_  | **61.01**| **63.19**| _36.60_  | **54.01**| **73.69**| **78.67**| _78.63_|
> > > >
> > > > Consistent with our observations from the original experiments, **TaSL performs similarly to PT‑LoRA in our domain‑adaptation CPT setting, but does not emerge as one of the strongest baselines.** This pattern appears again in the new domains: on the Coding tasks, TaSL stays close to PT‑LoRA but below stronger baselines such as PT‑Full and Replay; and in the Multilingual medical setting, TaSL mostly remains in the same performance range as LoRA‑based approaches. This was the primary reason we initially did not include TaSL under tight compute and time constraints.
> > > >
> > > > We hope these newly added results help fully address your question, and we are grateful for your constructive suggestions that led to a more complete evaluation.
> > > >
> > > > We have updated the content in the PDF as well as the relevant sections in our responses to the other reviewers. Thank you again for your patience and thoughtful engagement with our submission.
> > > >
> > > > Best regards,
> > > >
> > > > All Authors of Submission #2077

---

### Official Review · Reviewer_eXJC · 2025-10-29

**Soundness:** 4
**Presentation:** 4
**Contribution:** 4
**Rating:** 6
**Confidence:** 4

**Summary:**

The paper proposes ADEPT, a two-stage framework for domain-adaptive continual pretraining (CPT) of LLMs: (1) General-competence guided selective layer expansion—duplicate the least general-critical layers (identified via layer-ablation loss deltas) with function-preserving init; (2) Adaptive unit-wise decoupled tuning—partition each expanded layer into functional units and assign inverse-importance learning rates to protect general-critical units while adapting flexible ones. On math and medical corpora, ADEPT reports gains over full-parameter CPT and other baselines on both domain metrics and general benchmarks.

**Strengths:**

1. Clear, principled motivation from functional specialization observations (early layers more general-critical; heterogeneous unit importance), which directly informs where to expand and how to tune.
2. Sound expansion design via function-preserving identity copy (zeroing output projections) to avoid initial interference, consistent with Net2Net/FPI ideas.
3. Simple, implementable LR rule, periodically refreshed, giving a practical recipe for decoupled protection vs. adaptation.
4. Strong empirical results across Qwen3 (1.7B/4B/8B) and Llama3-8B on GSM8K/ARC and MedQA/MMCU/CMB; ADEPT often improves both domain and general benchmarks vs. full-CPT, replay, LLaMA-Pro, and LoRA/TaSL
5. Ablations and analyses show both stages matter; uniform expansion underperforms selective expansion; token-shift analysis suggests focused domain injection.

**Weaknesses:**

1. Importance identification and Fig. 2:

(a) From the training-domain perspective, it seems more appropriate to identify domain-specific important layers, not just those important to the general domain. Would it further boost performance if expansion happens on important layers for the target training domain? Could you report per-domain importance profiles and their overlap with the “general-critical” set?

(b) This raises a potential domain conflict at expanded layers: a layer deemed “not important” for the general domain might be crucial for math, yet gets expanded and trained on medical data. How often does this conflict occur, and how does ADEPT mitigate it?

(c) MLP dominance. Fig. 2 suggests MLPs dominate general-knowledge importance. Is this driven by parameter scale or by an intrinsic representational role of MLPs?

2. It makes sense that zero-initializing the expanded layer’s output projection, and it promotes stability, but if many MLP output projections are marked important, does zeroing them hinder adaptation? To ensure that forward computation remains
unchanged, making other modules (e.g., mlp gate/up projection) should also work.
 - Clarification: #264, does the ffn out projection mean down projection?

3. In Table 2, why does uniform expansion produce inferior results in the medical domain? considering it enables more adaptation capacity

**Questions:**

see weakness

---

> ### Author Response · Authors · 2025-11-22
> **Response to Reviewer eXJC (1/3)**
>
> Dear Reviewer eXJC:
>
> We appreciate your encouraging evaluation and the insightful suggestions, please find our detailed responses below.
>
> ## **W1(a): Expanding domain-critical layers**
> We first clarify our rationale for expanding the least general-critical layers in ADEPT, and then report the analyses and experiments addressing expanding target-domain critical layers.
>
> **(1) Motivation for expanding general-noncritical layers.**
> 1. First, **injecting new domain knowledge is relatively easy for gradient-based optimization, whereas catastrophic forgetting is far more damaging**. Prior work shows that domain-specific fine-tuning often improves in-domain performance at the cost of substantial drops in general skills[1][2]. ADEPT therefore focuses on minimizing interference with general-critical parameters.
> 2. Second, **new domain information may overwrite existing one (e.g., medical factual updates)**. Thus, even perfectly identifying current domain-important layers does not guarantee preservation or usage of previous domain expertise. In contrast, general-critical layers encode knowledge that should be preserved, making them a more reliable target for protection.
> 3. Third, **domain-important and general-critical layers may overlap due to shared semantics**. Expanding such overlapping layers could still trigger forgetting.
> 4. Finally, **General-nonimportant units, instead, offer a consistent, domain-agnostic, and safe expansion target.**. In practice, low-resource domains may lack sufficient representative domain data to reliably find domain-critical units.
>
> **(2) Probing and expanding domain-critical layers**
>
> We first examine the distribution of domain-important layers for math, medical and general domains. On model Qwen3-1.7B-Base, the computed layer importance rankings are as follows (see **Appendix R** in the revised manuscript for more details and visualizations, as well as results in other backbone models including Qwen3-4B-Base, Qwen3-8B-Base, LLama3-8B-Base):
> * **General**: 0,2,1,5,4,3,6,7,9,8,15,16,10,11,13,26,20,17,14,21,24,12,18,19,22,25,27,23
> * **Math**: 0,2,9,4,3,1,5,6,16,8,15,11,7,17,21,13,10,20,24,12,18,14,19,22,26,27,23,25
> * **Medical**: 2,0,5,6,4,1,7,9,3,17,11,15,21,10,8,16,14,13,12,18,20,24,19,26,22,23,27,25
>
> We observed **non-negligible overlap of important layers, particularly at early layers**, suggesting they may encode fundamental semantic representations important cross domains. Domain-critical expansion should therefore avoid these shared layers.
>
> We then introduce an **importance calibration** technique: for each layer, we compute domain-specific (1)importance rank and (2)normalized importance score and then calculate their differences from general-domain signals (subtracting a layer's general importance rank/score from its domain-specific ones). The two differences are normalized and combined equally (0.5/0.5) to obtain a calibrated domain-specific importance. The **calibrated rankings** are:
> * **Math**: 16,15,6,8,27,24,7,9,5,14,4,11,10,18,19,0,20,21,23,12,22,13,26,17,3,2,25,1
> * **Medicine**: 16,24,6,7,15,5,23,9,10,8,11,4,14,18,19,20,21,0,27,12,2,3,22,25,17,13,1,26
>
> These calibrated rankings also provide insights into domain-specific functional specialization:
> **Math** concentrates in the **middle depth**, indicating these layers may be in charge of reasoning and calculation.
> **Medical important layers are more distributed**, appearing in early, middle, and late parts of the LLM, which may reflect that medical tasks require combining diverse types of knowledge and abilities.(As illustrated in the **figures in Appendix R**, the results are more intuitively evident.)
>
> After calibration, layers with higher scores, i.e., important for the target domain but not for the general domain, are selected for expansion, while keeping the same decoupling strategy as in ADEPT.
>
> **Table: Calibrated domain-critical expansion vs. least general-critical expansion (Original ADEPT) on math domain**
>
> |**Domain**|**Math**|||||**General**|-|-|-|-|-|-|
> |-|-|-|-|-|-|-|-|-|-|-|-|-|
> |**Method**|GPQA|GSM+|GSM8K|ARC-e|ARC-c|BBH|TQA-MC1|TQA-MC2|CEval|HellaSwag|MMLU|CMMLU|
> |Qwen3-1.7B-Base|26.12|50.16|57.62|81.44|51.19|53.05|32.19|48.80|65.53|49.15|62.57|66.86|
> |ADEPT (original)|_31.03_|**54.83**|_70.51_|**82.48**|52.62|_54.40_|**33.66**|**50.00**|**66.27**|**49.17**|**62.62**|**67.06**|
> |ADEPT (Calibrated Math-critical expansion)|**31.79**|_52.47_|**71.02**|_81.78_|**53.84**|**55.26**|_32.82_|_48.36_|_65.16_|_48.83_|_61.35_|_66.26_|
> (continued below)

---

> ### Author Response · Authors · 2025-11-22
> **Response to Reviewer eXJC (2/3)**
>
> ## **Continuation of W1(a)**
> **Table: Calibrated domain-critical expansion vs. least general-critical expansion (Original ADEPT) on medical domain**
>
> |**Domain**|**Medical**|-|-|-|-|**General**|-|-|-|-|-|
> |-|-|-|-|-|-|-|-|-|-|-|-|
> |**Method**|PubMedQA|CMB-Clin|MedQA|MMCU|CMB|TQA-MC1|TQA-MC2|CEval|HellaSwag|MMLU|CMMLU|
> |Qwen3-1.7B-Base|_69.20_|--|48.39|_69.17_|63.67|32.19|_48.80_|65.53|_49.15_|_62.57_|_66.86_|
> |ADEPT (Original)|**69.60**|**53.84**|**50.75**|_71.98_|_65.43_|**34.39**|**51.05**|_66.64_|**49.28**|**62.80**|**66.89**|
> |ADEPT (Calibrated Med-critical expansion)|67.00|_52.92_|_49.70_|**72.55**|**66.82**|_32.31_|48.30|**66.90**|48.80|61.69|66.15|
>
> Expanding calibrated domain-critical layers **boosts target domain knowledge slightly but increases general forgetting**. This reflects an inherent trade-off between stability and aggressiveness in domain adaptation. We will add this expansion as an optional configuration in the released ADEPT code.
>
>
> ## **W1(b): Potential domain conflicts in expanded layers**
> We acknowledge the reviewer’s concern regarding potential domain conflicts. Fortunately, our empirical findings indicate that such **conflicts are rare in practice**. As shown in the importance ranking results aforementioned, layers identified as less important for the general domain are also of low-importance for math and medical domains. **This cross-domain agreement leaves little room for conflicts** in the low-importance region where ADEPT performs expansion.
>
> Moreoever, this situation can be conceptually interpreted as a multi-task tradeoff between previously learned and newly injected knowledge. If certain domain abilities (e.g., mathematics) are also crucial to retain, ADEPT provides a flexible mechanism to further reduce conflict: the probing dataset can incorporate a **mixture of general and target-domain data**. This adjusts the importance scoring to jointly reflect multiple desired capabilities, ensuring that high-importance layers for any selected competency are preserved and effectively mitigating the potential conflict.
>
>
> ## **W1(c): MLP dominance in importance**
> The MLP dominance observed in Fig.2 is a very interesting phenomenon. We believe this is not only a by‑product of their parameter scale, but also reflects MLPs' intrinsic representational role in storing and conveying factual knowledge, **supported by established interpretability studies**.
>
> Geva et al.[1] interpret transformer FFNs as key–value memories. Dai et al.[2] show “knowledge neurons” in FFNs carry relation and entity facts. Meng et al.[3] demonstrate that editing mid-layer FFNs alters factual generation. "Knowledge circuits"[4] shows that certain MLP neurons are repeatedly used in inference with specific facts. These studies highlight MLPs’ role in preserving factual and semantic knowledge, thus critical for general-domain competence.
>
> Additionally, parameter scale is also a contribution. Nichani et al.[5] show MLP factual capacity grows linearly with parameter count, reinforcing their importance.
>
> In summary, MLPs' dominant importance is consistent with both effects. The **alignment with prior interpretability research** provides a clear rationale for our method: **ADEPT's importance probe correctly flags MLPs as high-value for general competence**.

---

> ### Author Response · Authors · 2025-11-22
> **Response to Reviewer eXJC (3/3)**
>
> ## **W2: Zero-initializing the MLP output projection**
> Zero-initializing the MLP output projection is a principled choice **supported by prior work, preserves original MLP memory, and facilitates knowledge injection and updates**:
> 1. Zero-initializing the MLP output projection is **consistent with prior, mature expansion methods** (e.g., LLaMA-Pro[6]).
>
> 2. It does **not interfere with existing knowledge in original MLPs**. The replicated layer serves as a lossless “adapter” (proved in Appendix F). Zeroed output ensures functional equivalence and will not affect the KV-like representations in the original MLP.
>
> 3. **Zero-initialization actually facilitates adapation.** Zero-initialization helps the expanded layer focus on learning new knowledge rather than entangling with and being constrained by pre-existing information. As ADEPT’s Step-2 decoupling strategy assigns learning rate according to parameter importance, zeroing reduces importance, which then results in a **larger learning rate on the region**. This encourages the parameters to actively acquire and absorb new knowledge, consistent with MLPs’ role in storing facts[1–5]. Retaining original parameters could suppress learning due to high importance and low learning rate.
>
> 4. **Zero-initializing the output projection is more suitable for domain adaptation updates.**
> Since the output projection corresponds to the MLP value[1], zeroing it allows learning new values while the **up projection still captures meaningful semantic keys**. This is especially desirable in domain-specific updates, e.g., updating medical facts, where modifying the value while keeping the key structure stable reflects **knowledge rewriting**. In contrast, zeroing the gate leaves little room for new knowledge; zeroing up projection would impair the model’s ability to retrieve semantic keys, making it less suitable for controlled knowledge injection.
>
> **Clarification:** Yes, the “FFN output projection” refers to the down-projection (the second linear layer) of the feed-forward network.
>
> ## **W3: Uniform expansion capacity**
> We clarify that **uniform expansion does not provide more adaptation capacity than ADEPT**. In Table 2, both methods expand **the same number of layers** and introduce **the same amount of additional parameters**. The only difference lies in **where** the new layers are inserted: Uniform expansion follows a fixed periodic insertion pattern (as in Llama-Pro), where for a model with L layers and N new layers to insert, **a new layer is added every L/N layers**, unlike ADEPT that inserts layers at least general-critical layers.
>
> The inferior performance of uniform expansion therefore strengthens the rationale of ADEPT’s design: our expansion targets regions that minimizes interference with core capabilities and are more adaptable, thus allocating capacity where new domain knowledge can be absorbed more effectively. We have revised the manuscript **(Line 317 and 425)** to better clarify the exact mechanics of uniform expansion.
>
> ## **Reference**
> [1] Transformer feed-forward layers are key value memories.
>
> [2] Knowledge Neurons in Pretrained Transformers
>
> [3] Locating and Editing Factual Associations in GPT.
>
> [4] Knowledge Circuits in Pretrained Transformers NIPS 2024
>
> [5] Understanding Factual Recall in Transformers via Associative Memories. ICLR 2025
>
> [6] LLaMA Pro: Progressive LLaMA with Block Expansion. ACL 2024.

---

> ### Author Response · Authors · 2025-11-27
> **Gentle Follow‑Up on Our Discussion and Revisions**
>
> Dear Reviewer eXJC,
>
> We hope this message finds you well. We sincerely appreciate the time and expertise you have devoted to reviewing our manuscript. Your thoughtful and technically insightful comments have been invaluable to us, and we are truly grateful for your constructive feedback.
> In response to your suggestions, we have carefully revised and strengthened our manuscript. Specifically, we have:
> + Provided a clearer rationale for expanding general‑noncritical layers, and conducted new analyses on domain‑critical expansion, including calibrated layer importance, revised rankings, and comparative experiments across math and medical domains.
> + Added discussion on the rarity of domain conflicts in expanded layers, and sketched a possible way to adjust the probing dataset to better balance multiple competencies when needed.
> + Expanded our explanation of MLP dominance by connecting our observations to established interpretability findings on FFN–based factual storage and transformer memory mechanisms.
> + Further justified our choice of zero‑initializing MLP output projections, clarified its effect on preserving existing knowledge, and discussed its advantages for controlled domain adaptation.
> + Improved the explanation of uniform expansion to make clear that it uses the same parameter budget as ADEPT but allocates capacity differently, reinforcing why ADEPT’s targeted expansion is more effective.
>
> We sincerely hope that these revisions have adequately addressed your concerns. Your insights have greatly improved the technical clarity and conceptual grounding of our work, and we would be deeply grateful for any additional thoughts you might be willing to share.
> As the discussion phase is drawing to a close, we fully understand that your schedule may be demanding. If you could kindly spare a moment to let us know whether our revisions have sufficiently addressed your concerns, it would mean a great deal to us.
> Thank you again for your time, thoughtful evaluation, and invaluable guidance.
>
> Warmest regards,
>
> All Authors of Submission #2077

---

> ### Comment · Reviewer_eXJC · 2025-11-28
>
> Thanks for your detailed response. Most of my concerns have been addressed. I will raise my score to acceptance.

---

> > ### Author Response · Authors · 2025-11-28
> > **Sincere Appreciation and Response to Reviewer eXJC's Feedback**
> >
> > Dear Reviewer eXJC,
> >
> > Thank you very much for your thoughtful follow‑up and for recognizing our work. We are truly delighted and deeply encouraged by your decision to raise the score. It means a great deal to us, especially coming from a reviewer who has engaged so carefully with our submission.
> >
> > After submitting our responses, we try to exploratory experiments related to **W2: zero‑initializing the MLP output projection**. In particular, we are investigating **whether similar benefits arise when applying zero initialization to other components, to better understand the scope and mechanism behind this design choice**. Once we obtain stable and meaningful results, we would be very happy to share our findings with you.
> >
> > Thank you again for your generous support, your insightful feedback, and your confidence in our work.
> >
> > Warmest regards,
> >
> > All Authors of Submission #2077

---

> ### Author Response · Authors · 2025-12-01
> **Updated Findings About Zero‑initializing**
>
> Dear Reviewer eXJC,
>
> Thank you again for your thoughtful engagement. Following your earlier suggestion, we conducted additional experiments to systematically compare different zero‑initialization strategies within the expanded MLP, focusing on the up‑projection, down‑projection, and gate‑projection. Below, we summarize the key findings and provide the corresponding experiment tables for clarity.
>
> Key empirical findings from the new experiments are summarized as follows:
>
> + **Zero‑initializing the down‑projection consistently provides the best overall performance.** It preserves model functionality while offering the most effective capacity for controlled knowledge expansion, reaffirming its role as the most principled strategy for function‑preserving MLP extension.
>
> + **Zero‑initializing the up‑projection also performs strongly, often ranking among the top two methods across Math, Medicine, and General benchmarks.** This indicates that the semantic key pathway can also support stable and effective expansion, making zero‑up a solid alternative to zero‑down.
>
> + **In contrast, zero‑initializing the gate projection yields noticeably weaker results.** Because the gate primarily modulates activation magnitude rather than representing semantic information, zero‑gate does not create meaningful capacity for knowledge injection and is therefore not recommended.
>
> A more detailed technical analysis is provided in Appendix S.
>
> TABLE 1: Math-Domain Experiments
> |**Domain**|**Math**|||||**General**|-|-|-|-|-|-|
> |-|-|-|-|-|-|-|-|-|-|-|-|-|
> | Method | GPQA | GSM+ | GSM8K | ARC-e | ARC-c | BBH | TQA-MC1 | TQA-MC2 | CEval | HellaSwag | MMLU | CMMLU |
> | Qwen3‑1.7B‑Base | 26.12 | 50.16 | 57.62 | 81.44 | 51.19 | _53.05_ | 32.19 | _48.80_ | 65.53 | _49.15_ | _62.57_ | _66.86_ |
> | ADEPT | **31.03** | **54.83** | _70.51_ | **82.48** | **52.62** | **54.40** | **33.66** | **50.00** | _66.27_ | **49.17** | **62.62** | **67.06** |
> | ADEPT_up_projection | _29.01_ | 53.14 | **71.11** | _82.10_ | 51.62 | 51.82 | 32.68 | 47.92 | **66.34** | 48.54 | 62.32 | 66.72 |
> | ADEPT_gate_projection | 26.89 | _54.19_ | 69.58 | 81.81 | _51.79_ | 52.53 | _33.29_ | 47.88 | 66.20 | 48.61 | 62.28 | 66.33 |
>
>
> TABLE 2: Medical-Domain Experiments
>
>
> |**Domain**|**Medical**|-|-|-|-|**General**|-|-|-|-|-|
> |-|-|-|-|-|-|-|-|-|-|-|-|
> | Method | PubMedQA | CMB‑Clin | MedQA | MMCU | CMB | BBH | TQA-MC1 | TQA-MC2 | CEval | HellaSwag | MMLU | CMMLU |
> | Qwen3‑1.7B‑Base | 69.20 | — | 48.39 | 69.17 | 63.67 | **53.05** | 32.19 | 48.80 | 65.53 | _49.15_ | 62.57 | 66.86 |
> | ADEPT | _69.60_ | **53.84** | **50.75** | **71.98** | **65.43** | _52.96_ | **34.39** | **51.05** | **66.64** | **49.28** | **62.80** | 66.89 |
> | ADEPT_up_projection | **72.60** | _53.12_ | 48.47 | _71.62_ | _65.04_ | 52.56 | _33.29_ | _48.93_ | _66.57_ | 48.56 | 62.57 | **67.04** |
> | ADEPT_gate_projection | 68.00 | 52.76 | _49.18_ | 71.16 | 64.39 | 52.57 | 32.93 | 47.83 | 65.97 | 48.51 | _62.60_ | _67.02_ |
>
> We hope these additional findings clarify the behavior of different zero‑initialization strategies and further support the rationale behind our design. Thank you again for your constructive feedback and continued support.
>
> Warmest regards,
>
> All Authors of Submission #2077

---

### Official Review · Reviewer_gCgy · 2025-10-29

**Soundness:** 3
**Presentation:** 3
**Contribution:** 3
**Rating:** 6
**Confidence:** 3

**Summary:**

This paper proposes ADEPT which is a 2-stage framework for continual pretraining (CPT) of LLMs that aims to inject domain knowledge while avoiding catastrophic forgetting of general abilities. Stage1 selectively expands only those transformer layers that are least important for general competence. These layers are identified via a probing procedure that masks each layer and measures loss increase by duplicating them with function-preserving initialization. Stage2 decouples units such as attention and MLP, inside the expanded layers and assigns asymmetric learning rates inversely proportional to each unit's importance to the general domain, updating only the duplicates. Across Qwen3 models with sizes 1.7B, 4B, 8B and Llama3-8B, ADEPT reports higher target-domain accuracy and better retention on MMLU and CMMLU compare to the full CPT, LoRA, replay, and LLaMA-Pro, while tuning 15% of parameters and cutting training time more than 50%.

**Strengths:**

- The idea exploits functional specialization in LLMs where some layers and units are general-critical, while others are easier places to stuff domain knowledge. ADEPT explicitly measures that and routes updates accordingly.

- The method is simple with a mask-and-probe score to pick layers, a function-preserving copy to stay stable at step zero and a first-order importance signal to scale learning rates.

- On math and medical adaptation across several models, ADEPT generally boosts target-domain scores while avoiding the general-domain retention that is often seen after continual pretraining.

- The selective duplication step uses a straightforward probing signal and a function-preserving initialization to keep behavior unchanged at the start. The decoupled tuning step uses a first-order importance estimate to steer learning rates. It is kind of practical guardrail that you can integrate into a production CPT pipeline without rewriting everything.

**Weaknesses:**

- Layer masking for general ability and periodic unit-importance recomputation are not free. The paper shows strong results, but there is no crisp computation of how much extra compute the probing adds as models scale, or how sensitive it is to the quality or size of the probing corpus.

- The authors fix the number of expanded layers, which works but looks hardcoded. An auto-selection strategy would make it easier and boost the performance.

- The evaluation is largely multiple-choice accuracy. That is fine for comparability, but I would love to see at least some open-ended reasoning or robustness/safety checks.

- Medical gains for Llama3 rely on 8-layer expansion given Chinese weakness. That may suggest that the performance may hinge on relatively heavy expansion in challenging cross-lingual settings. How does ADEPT fare on code, law, finance, or multilingual transfer beyond Chinese and English?

**Questions:**

- Can you quantify the total compute overhead of importance probing and periodic unit re-weighting, and show how it scales from 2B to 8B or beyond?

- Can you autotune the number of expanded layers under a fixed parameter/time budget, instead of fixing by hand?

- Paper says: "Experiments on mathematical and medical benchmarks show that ADEPT outperforms full-parameter
CPT by up to 5.76% on the general domain". I am confused by this, aren't general-domain benchmarks MMLU and CMMLU? But it says experiments on mathematical and medical benchmarks. What is that improvement?

---

> ### Author Response · Authors · 2025-11-22
> **Response to Reviewer gCgy (1/3)**
>
> **Dear Reviewer gCgy:**
>
> We sincerely appreciate the reviewer’s insightful comments. We are encouraged by the positive evaluation and have thoroughly addressed each concern in our responses below.
>
> ## **W1&Q1: Probing overhead and probing dataset**
> **Probing overhead is minimal.**
> Layer importance is computed once (fully parallelizable), and unit importance is updated only every 500 steps via a single backward pass, adding negligible cost. As Table 6 (Appendix B.8) shows, ADEPT’s end-to-end time (probing + training) is still significantly faster than baselines.
>
> **Scaling and parallelism.**
> Layer-wise masking scales linearly with depth, but it is fully parallelizable across GPUs (e.g., assigning one layer per GPU). Gradient-based unit probing requires only a single backward pass per interval and thus scales favorably with model size. In practice, both steps are highly efficient. For example, probing on 580 samples (CMMLU + MMLU dev sets) takes only a few minutes on 8×H800 GPUs. Detailed timing breakdowns are provided in the table below.
>
> **Table: Time for layer masking and grad-based unit probing on single vs. 8 GPUs, along with total backward probing time**
> |Model|Layer Masking(1 GPU)|Layer Masking(8 GPUs)|Grad Probing(1 GPU)|Grad Probing(8 GPUs)| Backpropagation in Train(Total)|
> |-|-|-|-|-|-|
> |Qwen3-1.7B-Base|36m30s|8m16s|2m22s|1m50s| 16m20s|
> |Qwen3-4B-Base|1h10m|17m39s|3m14s|2m10s|25m43s|
> |Qwen3-8B-Base|1h32m|23m08s|5m24s|2m33s|29m03s|
> |Llama3-8B|1h24m|21m47s|6m26s|3m19s|40m42s|
>
> **The probing overhead scales sublinearly with model size and is small under parallelism**
>
> **Regarding sensitivity to probing corpus quality or size**, we study this in **Appendix H**, where we substitute the curated SFT probing corpus with noisier pretraining data and still observe competitive performance. This indicates that ADEPT does not rely on carefully engineered probing sets heavily and is **not sensitive to moderate variations in corpus size or quality.**
>
> ## **W2&Q2: Autotuning the Number of Expanded Layers**
>
> **Fixed layer count follows prior practice.**
> We agree that automatic selection of expanded layers is promising. For simplicity and stable parameter control, we follow common practice (e.g., LLaMA-Pro[1]) by fixing the number of expanded layers, yielding strong performance with straightforward implementation.
>
> **Expansion is fixed under parameter budget.**
> With a fixed parameter budget, the number of expandable layers is predetermined, as each Transformer layer adds a known parameter count.
>
> **Autotuning expansion under time budget is feasible.**
> Since training cost scales roughly linearly with expanded layers, we can estimate training time for different configurations and pick the maximum layer count within the time limit.
> Moreover, we provide empirical training times for expanding varying numbers of layers in Qwen3 models for reference. Notably, Qwen3-8B-Base shows a sharp time increase beyond 8 expanded layers, not due to algorithmic nonlinearity but GPU memory limits that force smaller batch sizes and thus longer training. Time estimates should therefore account for compute resources. Still, our measured times offer practical guidance for selecting expansion size under a fixed time budget.
>
> **Table: Training Time Schedule for Multi-Layer Adaptation Across Qwen3 Base Models**
>
> | Model             | 1 Layer     | 2 Layers       | 4 Layers        | 8 Layers        | 16 Layers       |
> |-------------------|-------------|----------------|-----------------|-----------------|-----------------|
> | Qwen3-1.7B-Base   | 24h         | 1d 2h          | 1d 9h           | 2d 1h           | 2d 20h          |
> | Qwen3-4B-Base     | 1d 17h      | 2d 4h          | 2d 11h          | 3d 4h           | 4d 10h           |
> | Qwen3-8B-Base     | 2d 18h      | 3d 2h          | 3d 15h          | 4d 6h           | 6d 22h          |
>
> **Adaptive expansion via importance threshold.**
> Beyond budget-aware tuning, a more dynamic approach is also possible: instead of expanding a fixed number of the least important layers, we can set an importance threshold and expand all layers below it, making expansion adaptive rather than predetermined.

---

> ### Author Response · Authors · 2025-11-22
> **Response to Reviewer gCgy (2/3)**
>
> ## **W3: Expanded Evaluation Beyond Multiple-Choice**
> While we use multiple-choice accuracy for comparability with prior work, our evaluation **already includes open-ended reasoning through GSM8K** which is based on **chain-of-thought reasoning where the model should generate open-ended solutions**, rather than choosing from given options.
> We find that **ADEPT still achieves the excellent performance even after expanding the number and diversity of evaluations**.
>
> To fully address the reviewer’s suggestion, we **substantially expand our evaluation** to include a broader range of benchmarks that emphasize open-ended reasoning, safety and robustness, and real-world applicability across multiple domains.
> * Mathematics:
>     - GSM-Plus: harder free-form math word problems;
>     - GPQA: graduate-level scientific reasoning questions.
> * Medicine:
>     - CMB-Clin: Chinese medical **reasoning with open-ended tasks** such as differential diagnosis and treatment planning;
>     - PubMedQA: requires **evidence-based reasoning** over biomedical literature abstracts and expressing **uncertainty**;
> * General-Domain:
>     - BBH: Big Bench Hard, with **free-form reasoning** tasks;
>     - HellaSwag: commonsense inference with adversarial contexts;
>     - CEval: complementary to CMMLU, covering Chinese professional exams;
>     - TruthfulQA: a benchmark for **model robustness** against false or misleading statements, assessing **safety**.
>
> The expended evaluation results are shown below:
>
> **Table: Qwen3-8B-Base on broader medical and general benchmarks**
>
> |**Domain**|**Medicine**|| **General**|||||
> |-|-|-|-|-|-|-|-|
> |**Method**|**PubMedQA**|**CMB-Clin** |**TruthQA-MC1**|**TruthQA-MC2**|**CEVAL** |**BBH**|**HellaSwag**|
> |Vanilla|77.40|--|_35.13_|_52.29_|**82.91**|**76.69**|**59.25**|
> |PT-Full|_78.80_|50.36|32.93|47.41|80.09|68.81|56.46|
> |Llama-Pro|78.60|51.64|35.74|52.04|78.16|71.80|54.87|
> |PT-LoRA|78.00|51.16|32.56|48.58|81.20|71.11|57.33|
> |Replay|**79.00**|_52.46_|32.93|48.50|79.72|69.73|55.75|
> |TASL|78.20|52.40|31.64|49.92|80.79|70.14|58.39|
> |**ADEPT**|**79.00**|**52.88**|**37.58**|**53.91**|_82.54_|_76.33_|_59.05_|
>
> **Table: LLaMA-3-8B on broader medical and general benchmarks**
>
> | **Domain** | **Medicine** || **General** |||||
> |-|-|-|-|-|-|-|-|
> | **Method** | **PubMedQA** | **CMB-Clin** | **TruthQA-MC1** | **TruthQA-MC2** | **CEval** | **BBH** | **HellaSwag** |
> | Vanilla    | 76.60 | --    | 26.81 | 43.95 | 50.45 | _62.33_ | **60.46** |
> | PT-Full    | _77.00_ | 56.84 | 23.75 | 37.74 | 49.70 | 47.47 | 58.03 |
> | Replay     | **78.20** | _59.20_ | **29.50** | _44.15_ | **52.75** | 51.48 | 58.71 |
> | Llama-Pro  | 76.40 | 54.92 | 24.85 | 38.03 | 49.70 | 62.31 | 60.00 |
> | PT-LoRA    | 73.20 | 53.48 | 24.24 | 38.16 | 49.93 | 61.77 | 59.79 |
> | TASL       | 72.40 | 50.00 | 26.35 | 39.46 | 50.51 | 61.92 | 58.34 |
> | **ADEPT**  | **78.20** | **59.70** | _28.40_ | **45.03** | _52.11_ | **62.52** | _60.39_ |
>
> **Table: Qwen3-8B-Base on broader math and general benchmarks**
> |**Domain**|**Math**|| **General**|||||
> |-|-|-|-|-|-|-|-|
> | **Method** | **GPQA** | **GSM+** | **TQA-MC1** | **TQA-MC2** | **CEval** | **BBH** | **HellaSwag** |
> | Vanilla    | 35.26 | 63.42 | _35.13_ | _52.29_ | **82.91** | 76.69 | **59.25** |
> | PT-Full    | _35.66_ | 64.54 | 32.93 | 49.83 | 79.42 | 73.15 | 55.83 |
> | Replay     | 34.55 | 65.14 | 34.27 | 50.18 | 78.08 | 74.62 | 55.64 |
> | Llama-Pro  | 35.44 | 64.36 | 29.86 | 46.05 | 81.64 | 75.50 | 56.16 |
> | PT-LoRA    | 33.66 | **67.01** | 33.78 | 51.64 | 78.38 | _79.81_ | 54.86 |
> | TASL       | 35.19 | 65.58 | 34.16 | 49.38 | 77.98 | 75.55 | 54.42 |
> | **ADEPT**  | **38.91** | _66.72_ | **38.18** | **53.39** | 82.39 | **79.83** | _59.16_ |
>
> **Table: LLaMA-3-8B on broader math and general benchmarks**
>
> | **Domain** | **Math** || **General** |||||
> |-|-|-|-|-|-|-|-|
> | **Method** | **GPQA** | **GSM+** | **TQA-MC1** | **TQA-MC2** | **CEval** | **BBH** | **HellaSwag** |
> | Vanilla    | 22.32 | 30.03 | 26.80 | 43.94 | 50.44 | 62.33 | _60.46_ |
> | PT-Full    | 23.99 | 33.19 | _29.13_ | **45.09** | 46.80 | 62.93 | 55.60 |
> | Replay     | _27.23_ | _33.62_ | **29.25** | 44.14 | _51.70_ | 62.51 | 56.10 |
> | Llama-Pro  | 25.44 | 29.54 | 26.68 | 43.22 | 50.96 | **63.51** | 58.65 |
> | PT-LoRA    | 25.22 | 29.90 | 27.78 | 43.46 | 50.14 | 62.75 | 59.11 |
> | TASL       | 24.86 | 30.26 | 27.71 | 41.68 | 49.68 | 62.37 | 58.91 |
> | **ADEPT**  | **27.78** | **34.44** | 28.78 | _44.31_ | **51.89** | _63.47_ | **60.77** |
>
> Across this richer and more comprehensive evaluation spectrum, **ADEPT consistently outperforms strong baselines**, bringing base models stronger open-ended reasoning ability, improved robustness and truthfulness. **More results on other backbones (Qwen3-1.7B-Base, Qwen3-4B-Base) can be found in Appendix O** in our revised manuscript.

---

> ### Author Response · Authors · 2025-11-22
> **Response to Reviewer gCgy (3/3)**
>
> ## **W4: ADEPT on challenging code and cross-lingual settings**
> **More layers help bridge larger knowledge gaps.**
> For Llama-3, expanding more layers is beneficial—**since our approach uses continued pretraining, and language patterns (like Chinese) must be acquired during this phase**, the model faces a dual challenge: learning Chinese linguistic competence **from scratch** while also absorbing specialized medical knowledge. This is akin to a non-Chinese speaker learning Chinese medicine, requiring greater learning capacity than injecting knowledge alone. Extra expanded layers thus enable better adaptation, aligning with ADEPT’s design of using layer expansion for knowledge injection.
>
> **Additional experiments in challenging settings.**
> To address the reviewer’s concern, we evaluated ADEPT on Qwen3-4B in two demanding scenarios: code and multilingual medical domains. **Full details are in Appendix P and Q. Experimental results show that our ADEPT method also achieves strong performance on more challenging domains.**
>
> **Code Domain:** Code is an ideal stress test because it simultaneously requires strict grammatical correctness and multi-step logical reasoning, and is also of high practical importance. The pre-training data includes _swallow-code-v2 and python-codes-25k_, totaling **13.7B tokens**. We evaluate on three complementary code-generation datasets, along with general-domain benchmarks:
>
> **Table: Performance of Qwen3-4B trained on code domain under different CPT methods**
> |**Domain**|**Code**|-|-|-|-|-|-|-|-|**General**|-|-|-|-|
> |-|-|-|-|-|-|-|-|-|-|-|-|-|-|-|
> |**Method**|HumanEval pass@1|pass@5|pass@10|MBPP pass@1|pass@5|pass@10|CRUXEval pass@1|pass@5|pass@10|TQA-MC1|TQA-MC2|HellaSwag|BBH|CEval|
> |Vanilla|50.37|78.61|86.59|8.71|32.42|49.80|**48.95**|**76.69**|**84.69**|**36.84**|_53.38_|**55.41**|_70.73_|_79.49_|
> |PT-Full|50.79|_83.85_|**89.02**|26.50|_58.62_|67.70|46.64|72.62|80.18|32.07|48.29|52.30|63.94|71.77|
> |Replay|**53.23**|80.31|85.37|19.30|40.67|54.36|43.12|69.55|78.29|33.54|49.73|53.23|66.46|74.00|
> |Llama-Pro|49.70|79.69|87.10|20.86|45.86|59.81|42.81|70.65|78.42|34.27|51.97|53.99|66.29|77.27|
> |PT-LoRA|36.77|74.43|82.93|_28.17_|**61.64**|**70.04**|39.65|70.97|79.80|31.82|49.28|51.94|34.48|75.85|
> | TaSL| 37.42  | 75.21  | 84.37   | 25.94        | 58.19  | 66.53   | 40.18            | 70.05  | 80.63   | 32.45    | 48.70    | 52.67     | 33.96  | 77.21  |
> |ADEPT|_51.13_|**84.81**|_87.20_|**31.17**|55.07|_67.98_|_46.68_|_74.57_|_83.69_|_35.13_|**53.83**|_54.98_|**70.77**|**79.63**|
>
>
> **Multilingual Medical Domain:** To evaluate ADEPT in multilingual transfer settings where the model must learn new knowledge and new languages simultaneously, we train on _MMedC_ excluding Chinese and English, covering French, Japanese, Russian, Spanish, totaling 16B tokens, and evaluate on multilingual medical benchmarks.
>
> **Table: Performance of Qwen3-4B trained on multilingual corpora under different CPT methods**
> |**Domain**|**Multilingual Medical**|-|-|-|**Multilingual General**|-|-|-|**CN/EN General**|-|-|-|-|
> |-|-|-|-|-|-|-|-|-|-|-|-|-|-|
> |**Method**|MMedBench ES|FR|JA|RU|MMMLU ES|FR|JA|RU|TQA-MC1|TQA-MC2|MMLU|CMMLU|CEval|
> |Vanilla|73.78|_75.08_|55.78|_73.05_|**66.68**|**66.19**|_60.85_|_62.56_|**36.84**|_53.38_|_73.19_|_77.92_|**79.49**|
> |PT-Full|73.34|**75.40**|**66.33**|59.68|61.77|60.86|56.91|59.68|30.60|43.97|66.91|68.88|69.61|
> |Replay|_74.47_|72.83|_62.31_|59.54|63.47|62.33|58.19|59.54|30.72|45.21|68.39|72.19|71.62|
> |Llama-Pro|73.74|74.43|53.27|60.70|65.32|64.43|59.79|60.70|35.86|52.54|72.03|77.08|77.79|
> |PT-LoRA|72.36|73.95|57.29|61.33|64.22|63.56|58.26|61.33|30.82|46.75|70.27|72.52|74.89|
> | TaSL| 71.22| 73.88        | 56.45        | 61.58        | 65.49    | 62.59    | 58.97    | 60.49    | 30.70    | 45.67    | 70.40  | 71.65  | 76.98  |
> |ADEPT|**74.56**|74.92|62.26|**74.61**|_66.64_|_66.15_|**61.01**|**63.19**|_36.60_|**54.01**|**73.69**|**78.67**|_78.63_|
>
> Across both challenging domains with task complexity and language diversity, ADEPT demonstrates the **most balanced and robust performance**, generally outperforming baselines, providing strong, consistent improvements over the base model while maintaining stability without catastrophic forgetting.
>
> ## **Q3: Clarification on Wording for General-Domain Improvement**
> We thank the reviewer for catching this wording issue. The phrase “Experiments on mathematical and medical benchmarks” in the paper is indeed a **typo**. It should read “Experiments on mathematical and medical domains.”
>
> Our intended meaning is: when ADEPT is trained on mathematics or medical domains, it not only improves performance on the corresponding in-domain benchmarks, but also yields measurable gains on general-domain benchmarks, achieving up to 5.76% improvement over full-parameter CPT. We have corrected the phrasing in the revised version to avoid confusion.
>
> ## **Reference**
> [1] LLaMA Pro: Progressive LLaMA with Block Expansion. ACL 2024.

---

> ### Author Response · Authors · 2025-11-27
> **Gentle Follow‑Up on Our Discussion and Revisions**
>
> Dear Reviewer gCgy,
>
> We hope this message finds you well. We sincerely appreciate the time and expertise you have devoted to reviewing our manuscript. Your thoughtful and encouraging feedback has been invaluable to us, and we are deeply grateful for your constructive suggestions.
> In response to your comments, we have carefully revised and strengthened our manuscript. Specifically, we have:
> + Clarified the probing overhead and demonstrated, through detailed timing studies and scaling analyses, that the probing cost is minimal and scales efficiently under parallelism.
> + Further clarified ADEPT’s robustness to variations in the probing corpus, noting that moderate noise or size changes do not significantly affect performance.
> + Added discussion and analyses on selecting the number of expanded layers, including budget‑aware tuning and a feasible adaptive strategy based on importance thresholds.
> + Substantially broadened our evaluation to cover more open‑ended reasoning, safety, robustness, and real‑world tasks across mathematics, medicine, and general domains.
> + Added new experiments on challenging knowledge-transfer scenarios, including code generation and multilingual medical domains, further validating the generalizability of ADEPT.
> + Clarified wording issues in the original manuscript to avoid ambiguity regarding general‑domain performance improvements.
> We sincerely hope that these revisions have addressed your concerns. Your insights have significantly improved the clarity and depth of our work, and we would be deeply grateful for any further thoughts you might be willing to share.
>
> As the discussion phase is drawing to a close, we fully understand that your schedule may be demanding. If you could spare a moment to let us know whether our updates have sufficiently resolved your concerns, it would mean a great deal to us.
> Thank you again for your time, thoughtful evaluation, and invaluable guidance.
>
> Warmest regards,
>
> All Authors of Submission #2077

---

### Official Review · Reviewer_E9mT · 2025-11-10

**Soundness:** 3
**Presentation:** 3
**Contribution:** 2
**Rating:** 4
**Confidence:** 3

**Summary:**

The paper proposes Adaptive Expansion and Dynamic Decoupled Tuning for domain-adaptive CPT (ADEPT). The authors utilize a two-stage training process:  1) General-competence-guided selective layer expansion, and 2) Adaptive unit-wise decoupled tuning.

In the first stage, the authors identify which layers are important for general knowledge and which are for domain knowledge by using curated probing data. In the second stage, they modularize the computation units and identify which units contribute to domain knowledge, updating these units with adaptive learning rates determined by gradient values. The authors use Llama3.1-8B and Qwen3 as base models and perform CPT for the math and medical domains. They experiment with model sizes ranging from 1.7B to 8B parameters and show that ADEPT outperforms the baselines.

**Strengths:**

- The paper is well written and easy to understand.
- The proposed method is generally well-motivated.
- The experiments are performed on various model sizes and show that the proposed method outperforms the baselines.

**Weaknesses:**

- The authors should conduct more experiments to justify their conclusions. Only MMLU and CMMLU are included to evaluate general knowledge, and only three domain-specific benchmarks are used. There are numerous benchmarks for both general and domain knowledge. To demonstrate that the proposed method is truly effective, the authors should include a more diverse and orthogonal set of benchmarks.

**Questions:**

- Does Table 6 in Section B.8 represent the end-to-end time from probing to training?
- Is the proposed method applicable to SFT and RL?
- How did you prepare $D_{\text{probe}} $?
- What is "Vanilla"? Is it the base model without CPT?
- Why is Vanilla better than PT-Full and Replay in the domain benchmarks for the 1.7B and 4B models? If training is properly executed, shouldn't they at least be better than the base model on the domain benchmarks?

---

> ### Author Response · Authors · 2025-11-22
> **Response to Reviewer E9mT (1/3)**
>
> **Dear Reviewer E9mT:**
>
> We sincerely appreciate your constructive suggestions, and we address your concerns below.
>
> ## **W1: Extensive multi-domain evaluations**
> We agree that more diverse and orthogonal evaluations can better demonstrate the effectiveness and generalizability of **ADEPT**. In response, we have substantially expanded our evaluations across **math, medical, and general domains**. **ADEPT still achieves the best performance after expanding the number and diversity of evaluations.**
>
> **Evaluations on Extended Benchmarks**
>
> The **newly added benchmarks** cover broader reasoning skills, more complex real-world constraints, and orthogonal evaluation formats:
> |**Domain**|**Benchmark**|**Description**|**Evaluation Method**|
> |-|-|-|-|
> |**Mathematics**|**GPQA**|Graduate-level competition assessing advanced problem-solving and reasoning|Generative + rule-based matching|
> ||**GSM-Plus**|Extending GSM8K to test robustness and stability under perturbed problems|Generative + rule-based matching|
> |**Medicine**|**CMB-Clin**|Chinese medical reasoning covering differential diagnosis, treatment planning, etc.|Generative + GPT-5 judged win-rate|
> ||**PubMedQA**| Evidence-based biomedical QA requiring reasoning over clinical literature| Multiple-choice|
> |**General**|**BBH**|Multifaceted reasoning with generation-based solutions across 27 tasks|Generative|
> ||**HellaSwag**|Commonsense inference for counterfactual plausibility| Multiple-choice|
> ||**CEVAL**|Comprehensive Chinese exam orthogonal to CMMLU|Multiple-choice|
> ||**TruthfulQA**|Measures model truthfulness and hallucination resistance|Multiple-choice|
>
> Here we report **results on Qwen3-8B-Base and LLaMA3-8B**; results on **other backbones (Qwen3-4B-Base, Qwen3-1.7B-Base) are in Appendix O** of the revised manuscript:
>
> **Table: Qwen3-8B-Base on broader medical and general benchmarks**
> |**Domain**|**Medicine**|| **General**|||||
> |-|-|-|-|-|-|-|-|
> |**Method**|**PubMedQA**|**CMB-Clin** |**TruthQA-MC1**|**TruthQA-MC2**|**CEVAL** |**BBH**|**HellaSwag**|
> |Vanilla|77.40|--|35.13|_52.29_|**82.91**|**76.69**|**59.25**|
> |PT-Full|_78.80_|50.36|32.93|47.41|80.09|68.81|56.46|
> |Llama-Pro|78.60|51.64|35.74|52.04|78.16|71.80|54.87|
> |PT-LoRA|78.00|51.16|32.56|48.58|81.20|71.11|57.33|
> |Replay|**79.00**|_52.46_|32.93|48.50|79.72|69.73|55.75|
> |TASL|78.20|52.40|31.64|49.92|80.79|70.14|58.39|
> |**ADEPT**|**79.00**|**52.88**|**37.58**|**53.91**|_82.54_|_76.33_|_59.05_|
>
> **Table: LLaMA-3-8B on broader medical and general benchmarks**
> |**Domain**|**Medicine**||**General**|||||
> |-|-|-|-|-|-|-|-|
> |**Method**|**PubMedQA**|**CMB-Clin**|**TruthQA-MC1**|**TruthQA-MC2**|**CEval**|**BBH**|**HellaSwag**|
> |Vanilla|76.60|--|26.81|43.95 | 50.45 | _62.33_ |  **60.46** |
> |PT-Full| _77.00_ | 56.84 | 23.75 | 37.74 | 49.70 | 47.47 | 58.03 |
> |Replay| **78.20** | _59.20_ | **29.50** | _44.15_ | **52.75** | 51.48 | 58.71 |
> |Llama-Pro| 76.40 | 54.92 | 24.85 | 38.03 | 49.70 | 62.31 | 60.00 |
> |PT-LoRA| 73.20 | 53.48 | 24.24 | 38.16 | 49.93 | 61.77 | _59.79_ |
> |TASL| 72.40 | 50.00 | 26.35 | 39.46 | 50.51 | 61.92 | 58.34 |
> |**ADEPT**| **78.20** | **59.70** | _28.40_ | **45.03** | _52.11_ | **62.52** |60.39 |
>
> **Table: Qwen3-8B-Base on broader math and general benchmarks**
> |**Domain**|**Math**|| **General**|||||
> |-|-|-|-|-|-|-|-|
> | **Method** | **GPQA** | **GSM+** | **TQA-MC1** | **TQA-MC2** | **CEval** | **BBH** | **HellaSwag** |
> | Vanilla    | 35.26 | 63.42 | _35.13_ | _52.29_ | _82.91_ | 76.69 | **59.25** |
> | PT-Full    | _35.66_ | 64.54 | 32.93 | 49.83 | 79.42 | 73.15 | 55.83 |
> | Replay     | 34.55 | 65.14 | 34.27 | 50.18 | 78.08 | 74.62 | 55.64 |
> | Llama-Pro  | 35.44 | 64.36 | 29.86 | 46.05 | 81.64 | 75.50 | 56.16 |
> | PT-LoRA    | 33.66 | **67.01** | 33.78 | 51.64 | 78.38 | _79.81_ | 54.86 |
> | TASL       | 35.19 | 65.58 | 34.16 | 49.38 | 77.98 | 75.55 | 54.42 |
> | **ADEPT**  | **38.91** | _66.72_ | **38.18** | **53.39** | **82.39** | **79.83** | _59.16_ |
>
> **Table: LLaMA-3-8B on broader math and general benchmarks**
> | **Domain** | **Math** || **General** |||||
> |-|-|-|-|-|-|-|-|
> | **Method** | **GPQA** | **GSM+** | **TQA-MC1** | **TQA-MC2** | **CEval** | **BBH** | **HellaSwag** |
> | Vanilla    | 22.32 | 30.03 | 26.80 | 43.94 | 50.44 | 62.33 | _60.46_ |
> | PT-Full    | 23.99 | 33.19 | _29.13_ | **45.09** | 46.80 | 62.93 | 55.60 |
> | Replay     | _27.23_ | _33.62_ | **29.25** | 44.14 | _51.70_ | 62.51 | 56.10 |
> | Llama-Pro  | 25.44 | 29.54 | 26.68 | 43.22 | 50.96 | **63.51** | 58.65 |
> | PT-LoRA    | 25.22 | 29.90 | 27.78 | 43.46 | 50.14 | 62.75 | 59.11 |
> | TASL       | 24.86 | 30.26 | 27.71 | 41.68 | 49.68 | 62.37 | 58.91 |
> | **ADEPT**  | **27.78** | **34.44** | 28.78 | _44.31_ | **51.89** | _63.47_ | **60.77** |
>
> Across the broader and more diverse benchmarks, models trained with ADEPT consistently achieve superior performance in both **target domains** and the **general domain**, providing stronger and more comprehensive evidence for the effectiveness and robustness of ADEPT.
>
> (continued below)

---

> ### Author Response · Authors · 2025-11-22
> **Response to Reviewer E9mT (2/3)**
>
> ## **Continuation of W1**
> **Continual Pretraining on Two New Domains**
>
> We further apply ADEPT to continual pretraining in **two new domains** with distinct symbolic structure and multilingual semantics: *code generation* and *multilingual medicine* where ADEPT successfully adapt base models to the target domains, which more thoroughly supports ADEPT’s **strong generalizability and practical effectiveness**. **The experiment setups and results are detailed in Appendices P and Q** of the revised manuscript. We warmly welcome you to explore them for additional validation evidence!
>
> ## **Q1: Probing time overhead**
> Yes, the time reported in Table 6 (Section B.8) **is end-to-end, including both probing and training**. The probing overhead is minimal:
> (1) initial layer-wise probing is fully parallelizable.
> (2) during training, unit importance is probed only every 500 steps with few backward pass, much cheaper than a full update.
> Crucially, ADEPT’s expansion mechanism reduces per-step training cost~(less backpropagation overhead), so the total runtime remains lower than baselines despite the small probing cost.
>
> ## **Q2: Apply ADEPT on SFT/RL**
> Yes, ADEPT can be adapted for both SFT and RL and our experiments show that ADEPT still achieves strong performance under SFT.
> Our focus on continual pre-training (CPT) is a deliberate design choice, motivated by three key points:
>
> 1. **CPT aligns with knowledge injection, not alignment.** CPT is widely viewed as the stage for injecting domain knowledge, whereas SFT/RL primarily handle alignment. Since our method is grounded in functional specialization of knowledge-storing units, CPT provides the ideal setting to validate our core assumption. Extending ADEPT to SFT or RL would only require a suitable probing objective.
>
> 2. **CPT induces stronger forgetting due to larger shifts.** CPT involves massive data volumes and significant distributional shifts, making it the dominant method for domain adaptation—but also more prone to catastrophic forgetting, which ADEPT directly addresses.
>
> 3. **ADEPT’s efficiency shines in long-horizon training.** By reducing backpropagation depth, ADEPT saves substantial compute—gains that matter most in costly, long-duration CPT scenarios.
>
> To demonstrate ADEPT's transferability to other training phases, we have additionally conducted **SFT experiments in the medical domain**. Here we report the results. Detailed experiment setups and more analysis can be found in **Appendix N** in the revised manuscript.
>
> **Table: LLaMA3-8B-Instruct SFT results with different strategies**
> |**Domain**|**Medical**||||| |**General**||||||
> |-|-|-|-|-|-|-|-|-|-|-|-|-|
> | **Method** | **PubMedQA** | **CMB-Clin** | **MedQA** | **MMCU** | **CMB** | **CEval** | **TruthQA-MC1** | **TruthQA-MC2** | **CMMLU** | **BBH** | **HellaSwag** | **MMLU** |
> | **Vanilla** | **78.80** | - | _62.68_ | 57.36 | 52.40 | 51.04 | 35.12 | 51.03 | **51.91** | _67.90_ | 56.32 | _65.30_ |
> | **SFT-Full** | 75.60 | 47.28 | 55.06 | 57.90 | 50.20 | 42.64 | 35.49 | 42.64 | 43.57 | 19.96 | 52.82 | 55.74 |
> | **SFT-LoRA** | **78.80** | 52.88 | 60.56 | **61.61** | _53.34_ | 51.63 | **37.45** | 51.63 | 51.69 | 67.86 | _57.53_ | 64.28 |
> | **LLaMA-Pro** | _78.20_ | 51.44 | 61.74 | 57.46 | 51.66 | 51.26 | 35.98 | 51.26 | 51.78 | 66.76 | 57.06 | 65.24 |
> | **TaSL** | 76.40 | _53.14_ | 61.32 | 59.85 | 52.30 | **51.76** | 36.32 | _51.86_ | 51.76 | 67.62 | **57.73** | 64.64 |
> | **ADEPT** | **78.80** | **53.76** | **63.92** | _60.85_ | **53.81** | _51.70_ | _37.10_ | **52.10** | _51.90_ | **68.00** | 57.20 | **65.40** |
>
> **Table: Qwen3-8B SFT results with different strategies**
> | **Domain** | **Medical** ||||| |**General**||||||
> |-|-|-|-|-|-|-|-|-|-|-|-|-|
> | **Method** | **PubMedQA** | **CMB-Clin** | **MedQA** | **MMCU** | **CMB** | **CEval** | **TruthQA-MC1** | **TruthQA-MC2** | **CMMLU** | **BBH** | **HellaSwag** | **MMLU** |
> | Vanilla | **78.40** | - | 63.55 | 79.88 | 73.17 | 78.75 | 35.86 | 53.49 | 77.97 | 79.18 | 55.85 | 74.59|
> | SFT-Full | 76.20 | 52.48 | 65.04 | 79.28 | 75.53 | 74.59 | _37.57_ | 55.91 | 75.39 | 56.90 | 56.90 | 73.03 |
> | SFT-LoRA | 77.40 | 51.84 | **66.84** | _82.29_ | _77.50_ | _78.82_ | **37.82** | _55.94_ | _78.26_ | 80.47 | 56.12 | _74.87_ |
> | Llama-Pro | _78.00_ | **54.16** | 63.70 | 80.59 | 75.38 | 78.75 | 35.98 | 53.93 | 77.75 | _80.63_ | 55.95 | 74.59 |
> | TaSL | 77.20 | 52.14 | 63.94 | 78.71 | 75.65 | 78.45 | 36.71 | **56.32** | **78.41** | 80.06 | _56.92_ | **75.09** |
> | **ADEPT** | 77.80 | _53.36_ | _65.94_ | **83.02** | **79.51** | **78.90** | 36.96 | 54.60 | 77.98 | **81.21** | **57.06** | 74.72 |
>
> The results show that ADEPT is fully applicable to SFT. On SFT, ADEPT delivers **stable, competitive, and generally better performance than established SFT baselines**, despite smaller gains than in CPT. Given the limited size of SFT data, we expect ADEPT to yield larger improvements as SFT scales to more training data. The same design can also naturally extend to RL.

---

> ### Author Response · Authors · 2025-11-22
> **Response to Reviewer E9mT (3/3)**
>
> ## **Q3: Probe data**
> The details of how we construct **$D_{probe}$ are provided in Appendix B.3**. We summarize the key points here for clarity: We use the development sets of MMLU and CMMLU as probing data. During probing, we perform backward passes using only the answer part to compute importance scores.
>
> ## **Q4：Clarification of the term "Vanilla"**
> Yes, "Vanilla" is the base model without CPT.
>
> ## **Q5："Vanilla" outperforms PT-Full and Replay on domain benchmarks in 1.7B and 4B models?**
> We acknowledge the reviewer’s observation and provide clarification.
>
> **1. Vanilla vs. PT-Full: Catastrophic forgetting dominates in small models.**
> In smaller models (e.g., 1.7B, 4B), continual pretraining without replay risks severe catastrophic forgetting: new domain data overwrites existing knowledge and abilities, degrading performance even when the knowledge is implicitly retained. Although the model acquires new domain knowledge, its general reasoning ability may degrade so substantially that it can not effectively leverage the learned knowledge to generate answers, leading to drops in domain performance. Limited model capacity exacerbates overfitting and forgetting, making Vanilla (direct domain tuning) surprisingly more effective than full-parameter tuning (PT-Full).
>
> **2. Vanilla vs. Replay: Replay data is hard to construct faithfully.**
> Replay’s success hinges on accurately mimicking the original pretraining distribution. However, for LLMs, the true pretraining corpus and its composition are unknown, making it practically infeasible to build an effective replay buffer. Heuristic replay mixes often fail to prevent forgetting, allowing Vanilla to outperform poorly calibrated replay strategies.
>
> **3. Consistency with literature and implications for our work.**
> These findings align with recent work (e.g., Huang et al.[1], Li et al.[2]), which shows that continual tuning (even with rehearsal) can hurt performance due to forgetting. Thus, Vanilla outperforming PT-Full and Replay is not anomalous but highlights a core challenge in continual pretraining: **forgetting often outweighs gains from new knowledge**. This underscores our key motivation: **Replay alone is insufficient**, and more principled mechanisms like ADEPT are needed to balance adaptation and retention.
>
> ## **Reference**
> [1] Mitigating Catastrophic Forgetting in Large Language Models with Self-Synthesized Rehearsal. ACL 2024.
> [2] Revisiting Catastrophic Forgetting in Large Language Model Tuning. EMNLP Findings 2024.

---

> ### Author Response · Authors · 2025-11-27
> **Gentle Follow‑Up on Our Discussion and Revisions**
>
> Dear Reviewer E9mT,
>
> We hope this message finds you well. We sincerely appreciate the time and expertise you have devoted to reviewing our manuscript. Your thoughtful feedback has been invaluable to us, and we are truly grateful for your constructive suggestions.
> In response to your comments, we have carefully revised and strengthened our manuscript. Specifically, we have:
> + Expanded our evaluations with a more diverse set of benchmarks across the mathematics, medicine, and general domains, further confirming ADEPT’s strong and consistent performance.
> + Added new continual-pretraining experiments in domains such as code generation and multilingual medicine to further demonstrate that the proposed method is truly effective.
> + Clarified the computational overhead of probing and confirmed that our reported time reflects the full end-to-end cost.
> + Extended ADEPT to supervised fine-tuning (SFT) and validated its effectiveness beyond continual pretraining.
> + Provided clearer explanations on the probe data construction and the definition of “Vanilla”.
> + Added analysis to contextualize why Vanilla may outperform PT-Full and Replay in small-model settings.
>
> We sincerely hope that these revisions have adequately addressed your concerns. Your insights have significantly improved the clarity and rigor of our work, and we would be deeply grateful for any further thoughts you might be willing to share.
> As the discussion phase is drawing to a close, we completely understand that your schedule may be demanding. If you could spare a moment to let us know whether our updates have sufficiently resolved your concerns, it would mean a great deal to us.
> Thank you again for your time, thoughtful evaluation, and invaluable guidance.
>
> Warmest regards,
>
> All Authors of Submission #2077

---

### Author Response · Authors · 2025-11-23
**A Humble Response to All Reviewers**

We would like to express our deepest gratitude to all reviewers for their thoughtful, constructive, and insightful comments. Your feedback has been invaluable in helping us improve the quality, clarity, and rigor of our work.

In response to your suggestions, we have conducted additional experiments and carefully revised the manuscript. To ensure clarity in the new results, all tables presenting quantitative comparisons now follow a consistent formatting convention: the highest value in each column is **bolded**, and the second-highest is *italicized*.

Furthermore, to enhance readability and organization, we have put the detailed experimental results, including sensitivity analyses, evaluations on additional benchmarks and domains (e.g., code, multilingual medical tasks), applicability to supervised fine-tuning, and layer-expansion strategies—into **Appendices M through S**. Within the updated manuscript, all **newly added content is highlighted in purple**, while **typographical corrections are marked in red** for easy reference.

We sincerely hope these revisions adequately address your concerns and appreciate the time and care you have devoted to reviewing our work. We look forward to your feedback.

---

### Author Response · Authors · 2025-12-02
**General Rebuttal Summary**

We sincerely thank all reviewers for their general positive assessments of our work and their insightful, constructive comments.

We are grateful that **all reviewers recognize our clear motivation grounded in functional specialization of LLMs, and the strong empirical gains of ADEPT in both domain performance and general-ability retention across model sizes and domains**. They also **highlight our paper’s clarity, experimental breadth, and informative ablations**. Reviewers gCgy and eXJC further emphasized ADEPT’s **practicality and ease of implementation**, with gCgy noting its **substantial parameter and training-time efficiency advantages**.

In response, we conducted additional experiments, added new analyses, and provided clarifications to strengthen the manuscript further. Major revisions include:

1. **Substantially Expanded Evaluation Benchmarks (Reviewers E9mT, gCgy, Manuscript Appendix O)**
We added **open-ended, orthogonal tasks** across math, medical, and general domains—including **reasoning, robustness, and safety benchmarks**. The results consistently show ADEPT’s robust gains beyond multiple-choice metrics.

2. **Additional Experiments on Code and Multilingual Domains (Reviewers gCgy, nRAZ, Manuscript Appendix P&Q)**
To test ADEPT under **challenging knowledge settings**, we applied it to **code** and **multilingual** CPT, where it maintained strong performance. It demonstrates ADEPT's **effectiveness and generalizability**.

3. **Applicability to SFT/RL (Reviewer E9mT, Manuscript Appendix N)**
We extended ADEPT to **medical SFT** and found that it integrates naturally beyond CPT, confirming full compatibility with later-stage fine-tuning pipelines.

4. **General-critical vs. Domain-critical Layer Expansion (Reviewer eXJC, Manuscript Appendix R)**
We analyzed why ADEPT expands **least general-critical layers**, proposed a **general-calibrated variant**, and verified through experiments the clear **trade-off between knowledge retention and domain injection**.

5. **MLP Importance Analysis & Zeroing Strategy (Reviewer eXJC, Manuscript Appendix S)**
Our analysis confirmed that zeroing the **MLP down projection** best balances stability and adaptation; notably, Reviewer eXJC stated our response **satisfactorily addressed their concerns** and they **plan to raise their score to acceptance**.

6. **Probing Overhead Analysis (Reviewer gCgy, Manuscript B.8)**
We quantified the overhead of layer probing and unit re-weighting, and provided detailed scaling behavior across model sizes.

7. **Autotuning Expanded Layer Count (Reviewer gCgy, Manuscript B.8)**
Based on timing measurements, we showed that **auto-selecting** the number of expanded layers is practical under budget constraints and offered actionable guidance.

8. **Probing Frequency Sensitivity (Reviewer nRAZ, Manuscript Appendix M)**
We evaluated how probing frequency affects both runtime and performance, clarifying ADEPT’s sensitivity in this dimension.

9. **Clarifications, Rationales, and Manuscript Improvements**
We also strengthened the manuscript by adding citation-backed explanations, revising ambiguous phrasing, fixing typos and notation, and improving overall clarity and flow.

Across all these revisions, we provide **extensive empirical evidence, deep analyses, and theoretical grounding that collectively address the reviewers’ concerns**. The new results consistently reinforce the core findings of our original submission.

Finally, we briefly summarize the discussion stage.

Both **Reviewer nRAZ** and **Reviewer eXJC** explicitly stated that they **appreciated our detailed responses and the additional experiments** provided during the discussion. Reviewer **eXJC further confirmed that our answers mostly resolved their concerns and will “raise my score to acceptance.”** We thank them for their engagement and the opportunity to present the new results.

While **Reviewers E9mT and gCgy** were unable to return due to time constraints and circumstances beyond their control, both had indicated **low confidence** in their initial assessments. Regrettably, we are unable to confirm whether our response has addressed their concerns. But importantly, **several key analyses overlapping with their earlier concerns (E9mT: Q1/Q3; gCgy: W1/Q1/W4)** were among those explicitly acknowledged as satisfactory by Reviewer eXJC and nRAZ. The remaining unreturned questions primarily involved method **generalization and robustness, which are precisely the aspects covered by our extensive new experiments, affirmed by Reviewer eXJC and nRAZ.**

We thank the reviewers, ACs, SACs and PCs again for their time and valuable feedback, and we remain available for any further questions or discussion.

Warmest regards,

All Authors of Submission #2077

---

### Meta-Review · Area_Chair_YNN3 · 2026-01-06

**Summary:**

The paper is praised for its clarity, strong motivation, and strong empirical results. Many concerns are also raised by reviewers and most of them were addressed in the rebuttal.

**Reviewer Concerns:**

The core idea of leveraging functional specialization in LLM layers and units to guide selective expansion and adaptive tuning is viewed as principled, practical, and well justified. Reviewers highlight the simplicity and implementability of ADEPT, including function-preserving expansion, probing-based importance estimation, and adaptive learning-rate control. Empirically, the method demonstrates strong and consistent improvements across multiple model sizes and domains (math and medicine), often outperforming competitive baselines while mitigating catastrophic forgetting. Ablations, pilot studies, and visual analyses further strengthen confidence in the design choices, and the work is seen as a meaningful contribution to continual/domain-adaptive pretraining.

The main concern is limited experimental scope, including the need for broader and more diverse benchmarks. Several methodological aspects need also deeper analysis, including computational overhead, sensitivity to probing data and hyperparameters, fixed expansion choices, and potential domain conflicts in layer selection. These were mostly addressed in the rebuttal.

**Reviewer Scores:**

The first reviewer who gave the score of 4 is likely to increase his/her score as the authors have conducted extensive experiment to address his/her concerns.

The other reviewers are positive about the paper.

---

### Decision · Program_Chairs · 2026-01-26

Accept (Poster)